# Low-Rank Optimal Transport through Factor Relaxation with Latent Coupling

**Peter Halmos**[1,*], **Xinhao Liu**[1,*], **Julian Gold**[2,*], and **Benjamin J. Raphael**[1]

[1]Department of Computer Science, Princeton University
[2]Center for Statistics and Machine Learning, Princeton University

## Abstract

Optimal transport (OT) is a general framework for finding a minimum-cost transport plan, or coupling, between probability distributions, and has many applications in machine learning. A key challenge in applying OT to massive datasets is the quadratic scaling of the coupling matrix with the size of the dataset. Forrow et al. (2019) introduced a factored coupling for the $k$-Wasserstein barycenter problem, which Scetbon et al. (2021) adapted to solve the primal low-rank OT problem. We derive an alternative parameterization of the low-rank problem based on the *latent coupling* (LC) factorization previously introduced by Lin et al. (2021) generalizing Forrow et al. (2019). The LC factorization has multiple advantages for low-rank OT including decoupling the problem into three OT problems and greater flexibility and interpretability. We leverage these advantages to derive a new algorithm *Factor Relaxation with Latent Coupling* (FRLC), which uses *coordinate* mirror descent to compute the LC factorization. FRLC handles multiple OT objectives (Wasserstein, Gromov-Wasserstein, Fused Gromov-Wasserstein), and marginal constraints (balanced, unbalanced, and semi-relaxed) with linear space complexity. We provide theoretical results on FRLC, and demonstrate superior performance on diverse applications – including graph clustering and spatial transcriptomics – while demonstrating its interpretability.

## 1 Introduction

Optimal transport (OT) is a powerful geometric framework for comparing probability distributions. OT problems seek a transport plan $P$ efficiently transforming one distribution ($a$) into another ($b$), subject to a ground cost $C$. The minimum cost yields a distance between $a$ and $b$, while the optimal transport plan reveals key structural similarities between the distributions. Owing to its versatility – different ground costs result in different ways to compare data – OT has found many applications in machine learning and beyond: from self-attention Tay et al. (2020); Sander et al. (2022); Geshkovski et al. (2023) and domain adaptation Courty et al. (2014); Solomon et al. (2015) to computational biology Schiebinger et al. (2019); Yang et al. (2020); Bunne et al. (2023); Liu et al. (2023).

This versatility is compounded by several variants using different forms of the objective function and/or constraints on the transport plan $P$. Wasserstein (W) OT Kantorovich (1942) compares distributions over the same space through the expected work of $P$, while Gromov-Wasserstein (GW) OT Mémoli (2011) compares distributions supported on distinct geometries through the expected metric distortion of $P$. Fused Gromov-Wasserstein (FGW) Vayer et al. (2020) OT is suited to structured data, taking a convex combination of the former two objectives. Independently, one can relax constraints on the *marginals* of $P$: in computational applications, $P$ is a matrix whose row-sum $P\mathbf{1}_m$ and column-sum $P^{\mathrm{T}}\mathbf{1}_n$ are called its left and right marginals. *Balanced* OT requires $P\mathbf{1}_m = a$ and $P^{\mathrm{T}}\mathbf{1}_n = b$. *Unbalanced* OT Frogner et al. (2015) replaces these constraints with penalties in the

transport cost, and is more robust to outliers. *Semi-relaxed* OT can be used to understand how one dataset embeds into another by imposing one hard constraint on either the left or right marginal, used for feature transfer Dong et al. (2023), and alignment of spatiotemporal data Halmos et al. (2024).

An important consideration in applying OT is the quadratic space of the transport plan. To address both the quadratic complexity and to provide robustness under sampling noise, Forrow et al. (2019) introduced another variant of OT, optimizing a $k$-Wasserstein Barycenter proxy for the rank-constrained Wasserstein objective. Their approach factors the transport plan through a small set of anchor points called hubs. Generalizing this approach, Scetbon et al. (2021) introduce the factorization $\boldsymbol{P} = \boldsymbol{Q} \operatorname{diag}(1/\boldsymbol{g}) \boldsymbol{R}^{\mathrm{T}}$ comprised of *sub-coupling* matrices $\boldsymbol{Q}$ and $\boldsymbol{R}$ sharing an *inner marginal* $\boldsymbol{g}$, meaning $\boldsymbol{Q}^{\mathrm{T}} \mathbf{1}_n = \boldsymbol{R}^{\mathrm{T}} \mathbf{1}_m = \boldsymbol{g}$. Building on this, Scetbon et al. (2021, 2022, 2023) derived algorithms to compute low-rank optimal transport plans for the primal OT problem with general costs, extending low-rank OT to GW and unbalanced problems using factored couplings.

Interestingly, a different factorization of $\boldsymbol{P}$ was proposed by Lin et al. (2021) in the context of $k$-Wasserstein barycenters. We call their factorization a *latent coupling* (LC) factorization, given by $\boldsymbol{P} = \boldsymbol{Q}\operatorname{diag}(1/\boldsymbol{g}_Q)\boldsymbol{T}\operatorname{diag}(1/\boldsymbol{g}_R)\boldsymbol{R}^{\mathrm{T}}$, with *two* inner marginals $\boldsymbol{g}_Q = \boldsymbol{Q}^{\mathrm{T}} \mathbf{1}_n$ and $\boldsymbol{g}_R = \boldsymbol{R}^{\mathrm{T}} \mathbf{1}_n$ and a general coupling $\boldsymbol{T}$. Lin et al. (2021) constrain the transport between $\boldsymbol{a}$ and $\boldsymbol{b}$ through two sets of learned anchor points, where the factorization is defined by three transport plans computed from three cost matrices between the points and their anchors. This objective differs from that of Forrow et al. (2019); Scetbon et al. (2021), who seek a minimal rank coupling with respect to a single, fixed cost $\boldsymbol{C}$. We observe that factored couplings of Forrow et al. (2019) correspond to LC factorizations with diagonal $\boldsymbol{T}$, suggesting the LC factorization of Lin et al. (2021) may provide an alternative parameterization of transport plans for the low-rank OT problem considered in Forrow et al. (2019); Scetbon et al. (2021). To our knowledge, this idea has not yet been explored.

**Contributions.** We present a new algorithm, Factor Relaxation with Latent Coupling (FRLC, with the informal mnemonic "frolic"), to compute a minimum cost low-rank transport plan using the LC factorization. Parameterizing low-rank transport plans with the LC factorization has a number of advantages. First, optimization of the low-rank OT objective decouples into three OT sub-problems on the LC factors $\boldsymbol{Q}, \boldsymbol{R}, \boldsymbol{T}$, leading to a simpler optimization algorithm. Second, this decoupling provides straightfoward extensions of FRLC to low-rank unbalanced and semi-relaxed OT; similar extensions for factored couplings required additional work Scetbon et al. (2023) beyond the balanced case. Third, the latent coupling $\boldsymbol{T}$ in the LC factorization provides additional flexibility to model transport between datasets with different numbers of clusters, and to model mass-splitting between these clusters, providing a high-level and interpretable description of $\boldsymbol{P}$ that differs from the factored couplings of Forrow et al. (2019). FRLC computes the LC factorization using a novel *coordinate* mirror descent scheme, alternating descent steps on variables $(\boldsymbol{Q}, \boldsymbol{R})$ and $\boldsymbol{T}$, inspired by the mirror descent approach of Scetbon et al. (2021). We call the descent step on $(\boldsymbol{Q}, \boldsymbol{R})$ *factor relaxation*, as the factors $\boldsymbol{Q}$ and $\boldsymbol{R}$ have relaxed inner marginals, allowing FRLC to be solved by OT sub-problems. FRLC handles multiple OT objectives (Wasserstein, Gromov-Wasserstein, Fused Gromov-Wasserstein), and marginal constraints (balanced, unbalanced, and semi-relaxed). We show FRLC performs better than existing state-of-the-art low-rank methods on a range of synthetic and real datasets, retaining the interpretability of Lin et al. (2021), and inheriting the broad applicability of Scetbon et al. (2021); Scetbon & Cuturi (2022); Scetbon et al. (2022, 2023).

## 2  Background

**Wasserstein OT.** Let $\{x_1, \ldots, x_n\}$ and $\{y_1, \ldots y_m\}$ be datasets in a metric space $\mathcal{X}$, and let $\Delta_d$ be the probability simplex of size $d$. Through probability vectors $\boldsymbol{a} \in \Delta_n$ and $\boldsymbol{b} \in \Delta_m$, each dataset is encoded as a probability measure: $\mu = \sum_{i=1}^{n} \boldsymbol{a}_i \delta_{x_i}$ and $\nu = \sum_{j=1}^{m} \boldsymbol{b}_j \delta_{y_j}$. Let

$$\Pi_{\boldsymbol{a}, \cdot} := \{\boldsymbol{P} \in \mathbb{R}_+^{n \times m} : \boldsymbol{P} \mathbf{1}_m = \boldsymbol{a}\}, \ \Pi_{\cdot, \boldsymbol{b}} := \{\boldsymbol{P} \in \mathbb{R}_+^{n \times m} : \boldsymbol{P}^{\mathrm{T}} \mathbf{1}_n = \boldsymbol{b}\}, \ \Pi_{\boldsymbol{a}, \boldsymbol{b}} := \Pi_{\boldsymbol{a}, \cdot} \cap \Pi_{\cdot, \boldsymbol{b}}.$$

Thus, $\Pi_{\boldsymbol{a}, \boldsymbol{b}}$ is the set of transport plans (probabilistic coupling matrices) with marginals $\boldsymbol{a}$ and $\boldsymbol{b}$. Given a cost function $c : \mathcal{X} \times \mathcal{X} \to \mathbb{R}_+$, define the cost matrix $\boldsymbol{C} \in \mathbb{R}_+^{n \times m}$ via $\boldsymbol{C}_{ij} = c(x_i, y_j)$. The Kantorovich formulation Kantorovich (1942) of discrete OT, also called the Wasserstein problem, seeks a transport plan $\boldsymbol{P}$ of minimal cost :

$$\mathrm{W}(\mu, \nu) := \min_{\boldsymbol{P} \in \Pi_{\boldsymbol{a}, \boldsymbol{b}}} \langle \boldsymbol{C}, \boldsymbol{P} \rangle_F. \tag{1}$$

**Gromov-Wasserstein OT.** In many applications, one wishes to compare datasets $\{x_1, \dots, x_n\} \subset \mathcal{X}$ and $\{y_1, \dots, y_m\} \subset \mathcal{Y}$ across distinct metric spaces $\mathcal{X}$ and $\mathcal{Y}$. The Gromov-Wasserstein (GW) objective Mémoli (2007, 2011) addresses the absence of a common metric or coordinate system through intra-domain cost functions $c_1 : \mathcal{X} \times \mathcal{X} \to \mathbb{R}_+$ and $c_2 : \mathcal{Y} \times \mathcal{Y} \to \mathbb{R}_+$, leading to intra-domain cost matrices $\boldsymbol{A}_{ik} = c_1(x_i, x_k)$ and $\boldsymbol{B}_{jl} = c_2(y_j, y_l)$. The GW objective function $\mathcal{Q}_{\boldsymbol{A},\boldsymbol{B}}(\boldsymbol{P}) := \sum_{i,j,k,l}(\boldsymbol{A}_{ik} - \boldsymbol{B}_{jl})^2 \boldsymbol{P}_{ij}\boldsymbol{P}_{kl}$ quantifies the expected metric distortion under $\boldsymbol{P}$, leading to the optimization problem:

$$\text{GW}(\mu, \nu) := \min_{\boldsymbol{P} \in \Pi_{\boldsymbol{a},\boldsymbol{b}}} \mathcal{Q}_{\boldsymbol{A},\boldsymbol{B}}(\boldsymbol{P}). \tag{2}$$

The Fused Gromov-Wasserstein (FGW) objective function Vayer et al. (2020) is a convex combination of the W and GW objectives, given as $\alpha \langle \boldsymbol{C}, \boldsymbol{P} \rangle_F + (1 - \alpha)\mathcal{Q}_{\boldsymbol{A},\boldsymbol{B}}(\boldsymbol{P})$, for hyperparameter $\alpha \in (0, 1)$.

**Relaxed marginal constraints.** *Balanced* OT (1) constrains $\boldsymbol{P}$ to lie in $\Pi_{\boldsymbol{a},\boldsymbol{b}}$. *Unbalanced* OT relaxes constraints $\boldsymbol{P}\mathbf{1}_m = \boldsymbol{a}$ and $\boldsymbol{P}^{\mathrm{T}}\mathbf{1}_n = \boldsymbol{b}$, replacing them with penalties in the form of KL divergences (or other divergences, see Chizat et al. (2018)):

$$\text{U-W}(\mu, \nu) := \min_{\boldsymbol{P} \in \mathbb{R}_+^{n \times m}} \langle \boldsymbol{C}, \boldsymbol{P} \rangle_F + \tau_L \text{KL}(\boldsymbol{P}\mathbf{1}_m \| \boldsymbol{a}) + \tau_R \text{KL}(\boldsymbol{P}^{\mathrm{T}}\mathbf{1}_n \| \boldsymbol{b}), \tag{3}$$

where $\tau_L, \tau_R > 0$ control the strength of each penalty. *Semi-relaxed* optimal transport relaxes exactly one of the hard constraints $\boldsymbol{P}\mathbf{1}_m = \boldsymbol{a}$ and $\boldsymbol{P}^{\mathrm{T}}\mathbf{1}_n = \boldsymbol{b}$ in the same manner. The semi-relaxed version of (1) obtained by relaxing only the "right" marginal constraint on $\boldsymbol{b}$ is:

$$\text{SR}^{\mathrm{R}}\text{-W}(\mu, \nu) := \min_{\boldsymbol{P} \in \Pi_{\boldsymbol{a},\cdot}} \langle \boldsymbol{C}, \boldsymbol{P} \rangle + \tau \text{KL}(\boldsymbol{P}^{\mathrm{T}}\mathbf{1}_n \| \boldsymbol{b}), \tag{4}$$

while it's "left" marginal counterpart $\text{SR}^{\mathrm{L}}\text{-W}(\mu, \nu)$ is defined analogously over $\boldsymbol{P} \in \Pi_{\cdot,\boldsymbol{b}}$, using penalty $\tau \text{KL}(\boldsymbol{P}\mathbf{1}_m \| \boldsymbol{a})$. Likewise, one can form semi-relaxed or unbalanced GW and FGW problems.

**Entropy regularization.** The seminal work Cuturi (2013b) introduced the Sinkhorn algorithm to solve an entropy regularized version of (1), $\text{W}_\epsilon(\mu, \nu) := \min_{\boldsymbol{P} \in \Pi_{\boldsymbol{a},\boldsymbol{b}}} \langle \boldsymbol{C}, \boldsymbol{P} \rangle_F - \epsilon H(\boldsymbol{P})$, massively improving the $O(n^3 \log n)$ time complexity of classical techniques Orlin (1997); Tarjan (1997). Above, $H$ is the entropy, $H(\boldsymbol{P}) = -\sum_{ij} \boldsymbol{P}_{ij}(\log \boldsymbol{P}_{ij} - 1)$, and $\epsilon > 0$ is the regularization strength.

**Low-rank regularization.** The nonnegative rank $\text{rk}_+(\boldsymbol{M})$ of matrix $\boldsymbol{M}$ is the least number of nonnegative rank-one matrices summing to $\boldsymbol{M}$. For $r \geq 1$, define

$$\Pi_{\boldsymbol{a},\cdot}(r) = \{\boldsymbol{P} \in \Pi_{\boldsymbol{a},\cdot} : \text{rk}_+(\boldsymbol{P}) \leq r\}, \quad \Pi_{\cdot,\boldsymbol{b}}(r) = \{\boldsymbol{P} \in \Pi_{\cdot,\boldsymbol{b}} : \text{rk}_+(\boldsymbol{P}) \leq r\}, \tag{5}$$

and let $\Pi_{\boldsymbol{a},\boldsymbol{b}}(r) = \Pi_{\boldsymbol{a},\cdot}(r) \cap \Pi_{\cdot,\boldsymbol{b}}(r)$. To estimate Wasserstein distances with greater stability and accuracy under sampling noise, Forrow et al. (2019) proposed a low-rank regularization on the coupling matrix, factoring the transport through a small set of anchor points. More explicitly, Scetbon et al. (2021) parameterized the set as $\Pi_{\boldsymbol{a},\boldsymbol{b}}(r)$ through the set $\text{FC}_{\boldsymbol{a},\boldsymbol{b}}(r)$ of *factored couplings*,

$$\text{FC}_{\boldsymbol{a},\boldsymbol{b}}(r) := \{(\boldsymbol{Q}, \boldsymbol{R}, \boldsymbol{g}) \in \mathbb{R}_+^{n \times r} \times \mathbb{R}_+^{m \times r} \times (\mathbb{R}_+^*)^r : \boldsymbol{Q} \in \Pi_{\boldsymbol{a},\boldsymbol{g}}, \boldsymbol{R} \in \Pi_{\boldsymbol{b},\boldsymbol{g}}\}.$$

The set $\text{FC}_{\boldsymbol{a},\boldsymbol{b}}(r)$ parameterizes $\Pi_{\boldsymbol{a},\boldsymbol{b}}(r)$ through $(\boldsymbol{Q}, \boldsymbol{R}, \boldsymbol{g}) \mapsto \boldsymbol{Q}\text{diag}(1/\boldsymbol{g})\boldsymbol{R}^{\mathrm{T}}$, as shown by Cohen & Rothblum (1993).

Scetbon et al. (2021) apply this factorization to solve the Wasserstein problem subject to $\boldsymbol{P} \in \Pi_{\boldsymbol{a},\boldsymbol{b}}(r)$ for general cost matrices:

$$\text{W}_r(\mu, \nu) := \min_{\boldsymbol{P} \in \Pi_{\boldsymbol{a},\boldsymbol{b}}(r)} \langle \boldsymbol{C}, \boldsymbol{P} \rangle_F \tag{6}$$

GW, unbalanced and semi-relaxed low-rank OT problems are defined as in (2), (3) and (4), replacing $\mathbb{R}_+^{n \times m}$, $\Pi_{\boldsymbol{a},\cdot}$, or $\Pi_{\cdot,\boldsymbol{b}}$ with rank-constrained counterparts (5). Scetbon & Cuturi (2022); Scetbon et al. (2022, 2023) developed a robust framework for solving all of these problems.

## 3 Factor Relaxation with Latent Coupling (FRLC) algorithm

### 3.1 Latent Coupling Factorization

We parameterize low-rank coupling matrices $\boldsymbol{P} \in \Pi_{\boldsymbol{a},\boldsymbol{b}}(r)$ using a factorization introduced in Lin et al. (2021), which we call the *latent coupling (LC) factorization* (Fig. 1). The key property of

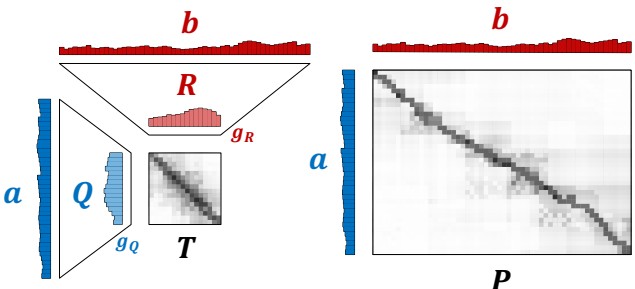

Figure 1: (Left) The LC factorization $P = Q\mathrm{diag}(1/g_Q)T\mathrm{diag}(1/g_R)R^{\mathrm{T}}$ of coupling matrix $P$ with outer marginals $a, b$, inner marginals $g_Q, g_R$, factors $Q, R$, and latent coupling $T$. (Right) Full-rank coupling matrix $P$.

this factorization is the presence of a coupling matrix $T$ linking two distinct inner marginals. For simplicity we describe this factorization using an $r$-dimensional latent space, but we also extend to non-square matrices linking two latent spaces of different dimensions, as demonstrated in the results.

**Definition 3.1** (Inner marginals). Given a factorization $P = QXR^{\mathrm{T}}$ of a coupling matrix $P \in \Pi_{a,b}(r)$, the *inner marginals* of $Q$ and $R$ are $g_Q := Q^{\mathrm{T}}\mathbf{1}_n$ and $g_R := R^{\mathrm{T}}\mathbf{1}_m$, respectively, where $g_Q, g_R \in \Delta_r$.

To distinguish the different marginals, we refer to $a$ and $b$ as *outer marginals*.

**Definition 3.2** (LC factorization). Given a coupling matrix $P \in \Pi_{a,b}(r)$, a *latent coupling (LC) factorization of $P$* is $P = Q\mathrm{diag}(1/g_Q)T\mathrm{diag}(1/g_R)R^{\mathrm{T}}$, where $g_Q$ and $g_R$ are the inner marginals of $Q$ and $R$, $Q \in \Pi_{a,\cdot}$, $R \in \Pi_{b,\cdot}$, and $T \in \Pi_{g_Q,g_R}$.

We call the factors $Q, R, T$ in an LC factorization *sub-couplings*. Let $\mathcal{R}_+ := \mathbb{R}_+^{n \times r} \times \mathbb{R}_+^{m \times r} \times \mathbb{R}_+^{r \times r}$. Given probability vectors $a \in \Delta_n$, $b \in \Delta_m$ and a positive integer rank $r$, let

$$\mathsf{LC}_{a,b}(r) := \{(Q, R, T) \in \mathcal{R}_+ : Q \in \Pi_{a,\cdot}, R \in \Pi_{b,\cdot}, T \in \Pi_{g_Q,g_R}\},$$

be the set of admissible sub-couplings for the LC factorization. Definition 3.2 gives the following map from $\mathsf{LC}_{a,b}(r)$ to $\Pi_{a,b}(r)$:

$$(Q, R, T) \mapsto Q\mathrm{diag}(1/g_Q)T\mathrm{diag}(1/g_R)R^{\mathrm{T}} =: P_{(Q,R,T)}. \tag{7}$$

Since this map is surjective, the set $\mathsf{LC}_{a,b}(r)$ parameterizes $\Pi_{a,b}(r)$. Surjectivity follows from the fact that $\mathsf{FC}_{a,b}(r)$ maps injectively into $\mathsf{LC}_{a,b}(r)$, through $(Q, R, g) \mapsto (Q, R, \mathrm{diag}(g))$, and $\mathsf{FC}_{a,b}(r)$ maps surjectively onto $\Pi_{a,b}(r)$ via $(Q, R, g) \mapsto Q\mathrm{diag}(1/g)R^{\mathrm{T}}$. Definition 3.1 and Definition 3.2 are readily extended in two directions: the case when the outer marginal constraints are relaxed such that $Q \in \mathbb{R}_+^{n \times m}$ or $R \in \mathbb{R}_+^{n \times m}$, while maintaining the constraint that $T \in \Pi_{g_Q,g_R}$; as well as the case of non-square $T$.

### 3.2 The Balanced FRLC Algorithm

We introduce an algorithm Factor Relaxation with Latent Coupling (FRLC), to compute a LC factorization of minimum cost. We first describe the FRLC algorithm for the balanced Wasserstein problem. Extensions to other and marginal constraints are discussed later. The FRLC objective function, for low-rank, balanced Wasserstein OT, is

$$\mathcal{L}_{\mathrm{LC}}(Q, R, T) := \langle C, P_{(Q,R,T)} \rangle_F, \tag{8}$$

where $P_{(Q,R,T)}$ is defined by (7). Since $\mathsf{LC}_{a,b}(r)$ parameterizes $\Pi_{a,b}(r)$, problem (8) is equivalent to low rank problem (6). The FRLC algorithm is built from projections onto convex sets, described by constraints on the outer marginals alone for $(Q, R)$ and by the inner marginals alone for $T$. Given $(Q, R, T) \in \mathsf{LC}_{a,b}(r)$, sub-couplings $Q$ and $R$ are constrained by:

$$\mathcal{C}_1(a) := \{(Q, R, T) \in \mathcal{R}_+ : Q\mathbf{1}_r = a\}, \quad \mathcal{C}_1(b) := \{(Q, R, T) \in \mathcal{R}_+ : R\mathbf{1}_r = b\}.$$

The convex sets constraining the latent coupling matrix $\boldsymbol{T}$ are

$$\mathcal{C}_2(\boldsymbol{g}_Q) := \{(\boldsymbol{Q}, \boldsymbol{R}, \boldsymbol{T}) \in \mathcal{R}_+ : \boldsymbol{T}\mathbf{1}_r = \boldsymbol{g}_Q\}, \quad \mathcal{C}_2(\boldsymbol{g}_R) := \{(\boldsymbol{Q}, \boldsymbol{R}, \boldsymbol{T}) \in \mathcal{R}_+ : \boldsymbol{T}^{\mathrm{T}}\mathbf{1}_r = \boldsymbol{g}_R\},$$

where $\boldsymbol{g}_Q = \boldsymbol{Q}^{\mathrm{T}}\mathbf{1}_n$ and $\boldsymbol{g}_R = \boldsymbol{R}^{\mathrm{T}}\mathbf{1}_m$ as per Definition 3.1. Writing $\mathcal{C}_1 = \mathcal{C}_1(\boldsymbol{a}) \cap \mathcal{C}_1(\boldsymbol{b})$ and $\mathcal{C}_2 = \mathcal{C}_2(\boldsymbol{g}_Q) \cap \mathcal{C}_2(\boldsymbol{g}_R)$, one has $\mathsf{LC}_{\boldsymbol{a},\boldsymbol{b}}(r) = \mathcal{C}_1 \cap \mathcal{C}_2$.

We use *coordinate* mirror descent to optimize (8), building on the mirror descent (MD) approach of Scetbon et al. (2021); Scetbon & Cuturi (2022); Scetbon et al. (2022, 2023) for the low-rank problem. First we take a descent step in the variables $(\boldsymbol{Q}, \boldsymbol{R})$ for a fixed $\boldsymbol{T}$, using KL penalties on their inner marginals. These "soft" constraints allow the joint optimization in $(\boldsymbol{Q}, \boldsymbol{R})$ to decouple into two semi-relaxed OT problems, one for each variable. We call this step *factor relaxation* as this allows $(\boldsymbol{Q}, \boldsymbol{R})$ to have relaxed inner marginals $\boldsymbol{g}_Q$ and $\boldsymbol{g}_R$. Next we take a descent step in the latent coupling variable $\boldsymbol{T}$, fixing the $\boldsymbol{Q}$ and $\boldsymbol{R}$, equivalent to solving a balanced OT problem. Thus, solving both coordinate descent steps corresponds to solving three OT problems.

We now provide further details on these coordinate descent steps, with the full algorithm given in Algorithm 1. Let $(\gamma_k)_{k=1}^N$ be a sequence of step sizes. As in Scetbon & Cuturi (2022), we choose $\ell^\infty$-normalization for the step-sizes. Our coordinate mirror descent in the factor relaxation step is:

$$(\boldsymbol{Q}_{k+1}, \boldsymbol{R}_{k+1}) \leftarrow \underset{(\boldsymbol{Q}, \boldsymbol{R}) \,:\, (\boldsymbol{Q}, \boldsymbol{R}, \boldsymbol{T}_k) \in \mathcal{C}_1}{\arg\min} \langle (\boldsymbol{Q}, \boldsymbol{R}), \nabla_{(\boldsymbol{Q}, \boldsymbol{R})}\mathcal{L}_{\mathrm{LC}} \rangle + \frac{1}{\gamma_k}\mathrm{KL}((\boldsymbol{Q}, \boldsymbol{R})\|(\boldsymbol{Q}_k, \boldsymbol{R}_k))$$
$$+ \tau\mathrm{KL}((\boldsymbol{Q}^{\mathrm{T}}\mathbf{1}_n, \boldsymbol{R}^{\mathrm{T}}\mathbf{1}_m)\|(\boldsymbol{Q}_k^{\mathrm{T}}\mathbf{1}_n, \boldsymbol{R}_k^{\mathrm{T}}\mathbf{1}_m))$$

The Sinkhorn kernels for the semi-relaxed OT problems arising from the factor relaxation step are:

$$\boldsymbol{K}_Q^{(k)} := \boldsymbol{Q}_k \odot \exp\big(-\gamma_k(\boldsymbol{C}\boldsymbol{R}_k\boldsymbol{X}_k^{\mathrm{T}} - \mathbf{1}_n\mathrm{diag}^{-1}((\boldsymbol{C}\boldsymbol{R}_k\boldsymbol{X}_k^{\mathrm{T}})^{\mathrm{T}}\boldsymbol{Q}_k\mathrm{diag}(1/\boldsymbol{g}_{Q_k}))^{\mathrm{T}})\big)$$
$$\boldsymbol{K}_R^{(k)} := \boldsymbol{R}_k \odot \exp\big(-\gamma_k(\boldsymbol{C}^{\mathrm{T}}\boldsymbol{Q}_k\boldsymbol{X}_k - \mathbf{1}_m\mathrm{diag}^{-1}(\mathrm{diag}(1/\boldsymbol{g}_{R_k})\boldsymbol{R}_k^{\mathrm{T}}\boldsymbol{C}^{\mathrm{T}}\boldsymbol{Q}_k\boldsymbol{X}_k)^{\mathrm{T}})\big),$$

introducing the shorthand $\boldsymbol{X} = \mathrm{diag}(1/\boldsymbol{g}_Q)\boldsymbol{T}\mathrm{diag}(1/\boldsymbol{g}_R)$ and where $\mathrm{diag}^{-1}(\cdot) : \mathbb{R}^{r \times r} \to \mathbb{R}^r$ denotes the matrix-to-vector extraction of the diagonal. This $\tau$-dependent regularization also allows us to show smoothness of the objective in Proposition E.5, from which the convergence guarantee Proposition 3.3 follows. We derive the semi-relaxed projection Algorithm 2 of the sub-couplings $\boldsymbol{Q}$ and $\boldsymbol{R}$ in Appendix G for completeness. We also show in Lemma A.1 that $\boldsymbol{g}_Q$ and $\boldsymbol{g}_R$ induced by the semi-relaxed projection are both feasible and locally optimal, not requiring separate optimization.

As $(\boldsymbol{Q}_{k+1}, \boldsymbol{R}_{k+1}, \boldsymbol{T}) \in \mathcal{C}_2$ if and only if $\boldsymbol{T} \in \Pi_{\boldsymbol{g}_{Q_{k+1}}, \boldsymbol{g}_{R_{k+1}}}$, after the factor relaxation step, we next take a coordinate MD step on the latent coupling $\boldsymbol{T}$:

$$\boldsymbol{T}_{k+1} \leftarrow \underset{\boldsymbol{T} \,:\, (\boldsymbol{Q}_{k+1}, \boldsymbol{R}_{k+1}, \boldsymbol{T}) \in \mathcal{C}_2}{\arg\min} \langle \boldsymbol{T}, \nabla_{\boldsymbol{T}}\mathcal{L}_{\mathrm{LC}} \rangle + \frac{1}{\gamma_k}\mathrm{KL}(\boldsymbol{T}\|\boldsymbol{T}_k). \tag{9}$$

This is equivalent to applying Sinkhorn (Algorithm 5) to $\boldsymbol{T}$ given $\boldsymbol{g}_Q$ and $\boldsymbol{g}_R$ with the kernel:

$$\boldsymbol{K}_T^{(k)} := \boldsymbol{T}_k \odot \exp\big(-\gamma_k\mathrm{diag}(1/\boldsymbol{g}_{Q_{k+1}})\boldsymbol{Q}_{k+1}^{\mathrm{T}}\boldsymbol{C}\boldsymbol{R}_{k+1}\mathrm{diag}(1/\boldsymbol{g}_{R_{k+1}})\big).$$

After the final iteration of the coordinate-MD scheme, $\boldsymbol{X} = \mathrm{diag}(1/\boldsymbol{g}_Q)\boldsymbol{T}\mathrm{diag}(1/\boldsymbol{g}_R)$ satisfies $\boldsymbol{X}\boldsymbol{g}_R = \mathbf{1}_r$ and $\boldsymbol{X}^{\mathrm{T}}\boldsymbol{g}_Q = \mathbf{1}_r$ as $\boldsymbol{T}$ is a coupling between $\boldsymbol{g}_Q$ and $\boldsymbol{g}_R$. Thus $\boldsymbol{P}_r = \boldsymbol{Q}\boldsymbol{X}\boldsymbol{R}^{\mathrm{T}} \in \Pi_{\boldsymbol{a},\boldsymbol{b}}$ and the iterates $(\boldsymbol{Q}_k, \boldsymbol{T}_k, \boldsymbol{R}_k)$ remain in the intersection of the constraint sets. Thus, in contrast to other approaches Scetbon et al. (2021); Lin et al. (2021); Forrow et al. (2019), we do not require Dykstra projections back into the intersection to maintain feasability. We note that our implementation of FRLC allows for a non-square latent coupling $\boldsymbol{T}$, providing greater interpretability in problem-specific applications. Above, we presented FRLC in the simplest case that $\boldsymbol{T}$ is square.

### 3.3 Initialization, convergence, and FRLC extensions

**Full-rank random initializations of the sub-coupling matrices.** We propose a new initialization of the sub-couplings $(\boldsymbol{Q}, \boldsymbol{R}, \boldsymbol{T})$ for the LC-factorization in Algorithm 6.This generates a full-rank initialization (Proposition F.1) in the set of rank-$r$ couplings $\Pi_{\boldsymbol{a},\boldsymbol{b}}(r)$ and is accomplished by applying Sinkhorn to random matrices. Our approach differs from Scetbon et al. (2021); Scetbon & Cuturi (2022) who use initializations for the diagonal factorization of Forrow et al. (2019), and are not applicable to a latent coupling that is non-diagonal, non-square, or with two distinct inner marginals.

---

**Algorithm 1** Balanced FRLC

---

Input $\boldsymbol{C}, r, \boldsymbol{a}, \boldsymbol{b}, \tau, \gamma, \delta, \varepsilon$

Initialize $\boldsymbol{g}_Q, \boldsymbol{g}_R = \frac{1}{r}\mathbf{1}_r$

$\boldsymbol{Q}_0, \boldsymbol{R}_0, \boldsymbol{T}_0 \leftarrow$ *Initialize-Couplings*$(\boldsymbol{a}, \boldsymbol{b}, \boldsymbol{g}_Q, \boldsymbol{g}_R)$   # Alg. 6

$\boldsymbol{X}_0 \leftarrow \mathrm{diag}(1/\boldsymbol{Q}_0^\mathrm{T}\mathbf{1}_n)\boldsymbol{T}_0\,\mathrm{diag}(1/\boldsymbol{R}_0^\mathrm{T}\mathbf{1}_m)$

**while** $\Delta((\boldsymbol{Q}_k, \boldsymbol{R}_k, \boldsymbol{T}_k), (\boldsymbol{Q}_{k-1}, \boldsymbol{R}_{k-1}, \boldsymbol{T}_{k-1})) > \varepsilon$ **do**   # $\triangle$ as in (10)

$\quad \nabla_{\boldsymbol{Q}} \leftarrow \boldsymbol{C}\boldsymbol{R}_k\boldsymbol{X}_k^\mathrm{T} - \mathbf{1}_n\mathrm{diag}^{-1}((\boldsymbol{C}\boldsymbol{R}_k\boldsymbol{X}_k^\mathrm{T})^\mathrm{T}\boldsymbol{Q}_k\mathrm{diag}(1/\boldsymbol{g}_Q))^\mathrm{T}$

$\quad \nabla_{\boldsymbol{R}} \leftarrow \boldsymbol{C}^\mathrm{T}\boldsymbol{Q}_k\boldsymbol{X}_k - \mathbf{1}_m\mathrm{diag}^{-1}(\mathrm{diag}(1/\boldsymbol{g}_R)\boldsymbol{R}_k^\mathrm{T}\boldsymbol{C}^\mathrm{T}\boldsymbol{Q}_k\boldsymbol{X}_k)^\mathrm{T}$

$\quad \gamma_k \leftarrow \gamma/\max\{\|\nabla_{\boldsymbol{Q}}\|_\infty, \|\nabla_{\boldsymbol{R}}\|_\infty\}$   # $\ell^\infty$-normalization of Scetbon & Cuturi (2022)

$\quad \boldsymbol{K}_{\boldsymbol{Q}}^{(k)}, \boldsymbol{K}_{\boldsymbol{R}}^{(k)} \leftarrow \boldsymbol{Q}_k \odot \exp(-\gamma_k\nabla_{\boldsymbol{Q}}), \boldsymbol{R}_k \odot \exp(-\gamma_k\nabla_{\boldsymbol{R}})$

$\quad \boldsymbol{Q}_k \leftarrow \mathrm{SR^R}\text{-}projection(\boldsymbol{K}_{\boldsymbol{Q}}^{(k)}, \gamma_k, \tau, \boldsymbol{a}, \boldsymbol{Q}_{k-1}^\mathrm{T}\mathbf{1}_n, \delta)$   # Semi-relaxed OT, Alg. 2

$\quad \boldsymbol{R}_k \leftarrow \mathrm{SR^R}\text{-}projection(\boldsymbol{K}_{\boldsymbol{R}}^{(k)}, \gamma_k, \tau, \boldsymbol{b}, \boldsymbol{R}_{k-1}^\mathrm{T}\mathbf{1}_m, \delta)$   # Semi-relaxed OT

$\quad \boldsymbol{g}_Q, \boldsymbol{g}_R \leftarrow \boldsymbol{Q}_k^\mathrm{T}\mathbf{1}_n, \boldsymbol{R}_k^\mathrm{T}\mathbf{1}_m$

$\quad \nabla_{\boldsymbol{T}} = \mathrm{diag}(1/\boldsymbol{g}_Q)\boldsymbol{Q}_k^\mathrm{T}\boldsymbol{C}\boldsymbol{R}_k\,\mathrm{diag}(1/\boldsymbol{g}_R)$

$\quad \gamma_{\boldsymbol{T}} = \gamma/\|\nabla_{\boldsymbol{T}}\|_\infty$   # $\ell^\infty$-normalization

$\quad \boldsymbol{K}_{\boldsymbol{T}}^{(k)} \leftarrow \boldsymbol{T}_k \odot \exp(-\gamma_{\boldsymbol{T}}\nabla_{\boldsymbol{T}})$

$\quad \boldsymbol{T}_k \leftarrow Sinkhorn(\boldsymbol{K}_{\boldsymbol{T}}^{(k)}, \boldsymbol{g}_R, \boldsymbol{g}_Q, \delta)$   # Balanced OT, Alg. 5

$\quad \boldsymbol{X}_k \leftarrow \mathrm{diag}(1/\boldsymbol{g}_Q)\boldsymbol{T}\,\mathrm{diag}(1/\boldsymbol{g}_R)$

**end while**

Return $\boldsymbol{P}_r = \boldsymbol{Q}\boldsymbol{X}\boldsymbol{R}^\mathrm{T}$

---

**Convergence analysis of FRLC.** As objective (8) is non-convex, it is important to have convergence guarantees. Our convergence criterion $\Delta(\cdot, \cdot)$ is defined in (10). To prove convergence we require a lower bound on the entries of $\boldsymbol{g}_Q$ and $\boldsymbol{g}_R$. Previous works introduce a lower-bound vector $\boldsymbol{\alpha} \leq \boldsymbol{g}$ enforced element-wise for stability and smoothness Scetbon et al. (2021). In FRLC the use of semi-relaxed projections naturally enforces a lower-bound. In Appendix E.5, we show that for any $\delta \in (0, \frac{1}{r})$, the FRLC algorithm's $\tau$-weighted regularization on the inner marginals can guarantee a uniform lower-bound of $\delta$ on the entries: for sufficiently large $\tau$ and $\tilde{O}(m^2/\epsilon)$ iterations for the sub-coupling Pham et al. (2020), one guarantees a lower bound of $\delta$ on $\boldsymbol{g}_R$ and $\boldsymbol{g}_Q$. This allows us to show objective smoothness in Proposition E.5. Previous work on low-rank optimal transport Scetbon et al. (2021) use the non-asymptotic convergence criterion of Ghadimi et al. (2014). Following existing works Dang & Lan (2015) establishing convergence rates of coordinate mirror-descent for smooth objectives, we show in Proposition 3.3 this criterion may be extended to coordinate-MD by adapting the block-descent lemma of Beck & Tetruashvili (2013).

**Proposition 3.3.** *Suppose one has $f \in C^1(\mathcal{X}, \mathbb{R})$ with block-coordinate Lipschitz gradient and block smoothness constants $(L_i)_{i=1}^p$, and a function $h \in C(\mathcal{X}, \mathbb{R})$ which is $\alpha$-strongly convex. For $\Phi = f + h$, suppose one performs coordinate mirror descent on $\Phi$ minimized over a product of closed convex sets $\mathcal{X} = \prod_{i=1}^p \mathcal{X}_i$. Let the sub-iterates with respect to the $i$-th block update be $\{\mathbf{x}_k^i\}_{i=0}^p$ where $\mathbf{x}_k := \mathbf{x}_k^0$ for $k \in [N]$ outer iterations. Then one has:*

$$\min_k \Delta(\mathbf{x}_k, \mathbf{x}_{k-1}) \leq \frac{D^2 L}{N(\alpha^2/2L)} = \frac{2D^2 L^2}{N\alpha^2},$$

*where $D$ is (36), $L$ is the global smoothness constant, stepsizes $\gamma_{k,i} := \alpha/L$, and convergence criterion $\Delta(\mathbf{x}_k, \mathbf{x}_{k-1})$ is given in (35).*

Specialized to the LC-parametrization, the criterion $\Delta_k(\boldsymbol{x}_k, \boldsymbol{x}_{k+1})$ is:

$$\Delta_k(\boldsymbol{x}_k, \boldsymbol{x}_{k+1}) := \frac{1}{\gamma_k^2}\left[\|\boldsymbol{Q}_{k+1} - \boldsymbol{Q}_k\|_F^2 + \|\boldsymbol{R}_{k+1} - \boldsymbol{R}_k\|_F^2 + \|\boldsymbol{T}_{k+1} - \boldsymbol{T}_k\|_F^2\right] \tag{10}$$

for $\boldsymbol{x}_k = (\boldsymbol{Q}_k, \boldsymbol{R}_k, \boldsymbol{T}_k)$. We show through Propositions 3.3, E.5 the following result:

**Proposition 3.4.** *The FRLC algorithm with step-sizes $\gamma_k = \alpha/L$ and iterates $\boldsymbol{x}_k = (\boldsymbol{Q}_k, \boldsymbol{R}_k, \boldsymbol{T}_k)$ has non-asymptotic stationary convergence in the criterion $\Delta(\cdot, \cdot)$ with:*

$$\min_{k \in 1, .., N-1} \Delta_k(\boldsymbol{x}_k, \boldsymbol{x}_{k+1}) \leq 2D^2 L^2/N\alpha^2$$

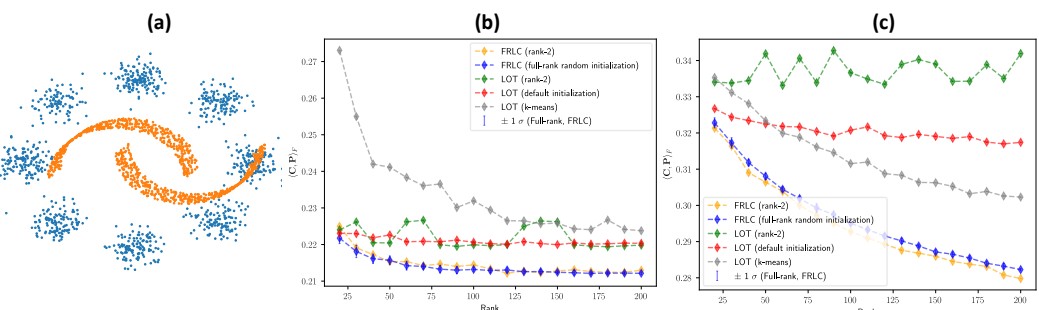

Figure 2: (a) Simulated dataset containing points from two moons (orange) and eight Gaussians (blue). (b) Transport cost $\langle C, P \rangle_F$ achieved by FRLC and LOT Scetbon et al. (2021) for the balanced Wasserstein problem on the dataset in (a) for different ranks and initializations. FLRC full rank (blue curve) is average over 10 random initializations. (c) Results on the 10D mixture of Gaussians dataset.

*Where $N$ is the number of iterations, $D$ the optimality-gap as in (36), and $L = \max_{i \in \{1,...,3\}}(L_i)$ the global smoothness for $L_i = \mathrm{poly}(\|C\|_F, n, m, r, \delta)$ the block-wise smoothness constants.*

The proof of Proposition 3.4 follows directly from our extension of the non-asymptotic criterion with the block-descent result Proposition 3.3 and the proof that this lemma holds in FRLC Proposition E.5. We also mention two improvements to other low rank approximation results in literature. In Proposition F.2 we show that one can *analytically* solve for the block-optimal $g$ for the factorization of Scetbon et al. (2021), and we improve the bound on the low-rank approximation error in Proposition E.7.

**FRLC for other marginal constraints and objectives.** The balanced FRLC algorithm can be extended simply to other marginal constraints owing to the decoupling of the coordinate MD scheme. In particular, by using either the semi-relaxed projections (Algorithm 2) or fully-relaxed (unbalanced) projections (Algorithm 3) on sub-couplings $Q$ and $R$, one can solve the balanced problem, the problem with the left or right marginal relaxed, or the unbalanced problem. As such, *all* variants of marginal constraints can be handled by a single algorithm, given in Algorithm 4.

We also extend the FRLC algorithm to the Gromov-Wasserstein problem. This consists of computing a GW-specific gradient with the appropriate marginal constraints applied to simplify their form, and re-computing Sinkhorn kernels as exponentiations of these gradients. The matrix form of the quadratic GW objective is $1_m^{\mathrm{T}} P^{\mathrm{T}} A^{\odot 2} P 1_m + 1_n^{\mathrm{T}} P B^{\odot 2} P^{\mathrm{T}} 1_n - 2\langle APB, P \rangle$, where $\odot$ denotes the Hadamard (entrywise) product. Then the GW-specific Sinkhorn kernels are

$$K_Q^{(k)} \leftarrow \exp\left(2\gamma_k(2AQX R^{\mathrm{T}} BR X^{\mathrm{T}} - A^{\odot 2} Q 1_r 1_r^{\mathrm{T}})\right),$$

$$K_R^{(k)} \leftarrow \exp\left(2\gamma_k(2BR X^{\mathrm{T}} Q^{\mathrm{T}} AQX - B^{\odot 2} R 1_r 1_r^{\mathrm{T}})\right),$$

$$K_T^{(k)} \leftarrow \exp(4\gamma_k \,\mathrm{diag}(g_Q^{-1}) Q^{\mathrm{T}} AQX R^{\mathrm{T}} BR \,\mathrm{diag}(g_R^{-1})).$$

In Algorithm 4, one can solve the GW-problem by using the kernels above. Here, we present the kernels omitting a rank-1 perturbation, which is given in Appendix D. From the Wasserstein and GW gradients, the FGW gradient is easily taken as a convex combination of the two. In this work, we primarily focus on the LC-factorization for the rank $r$ Wasserstein problem (6).

## 4 Experimental Results

We compare FRLC to existing low-rank and full-rank optimal transport algorithms on several datasets: simulated datasets previously used in Tong et al. (2023) and Scetbon et al. (2021); a massive spatial-transcriptomics dataset Chen et al. (2022); and a graph partitioning task Chowdhury & Needham (2021). Further details of each experiment (e.g. pre-processing, validation) are in Appendices K, L, and M. In the section below, LOT refers to the works of Scetbon et al. (2021, 2023, 2022) and Latent OT refers to Lin et al. (2021).

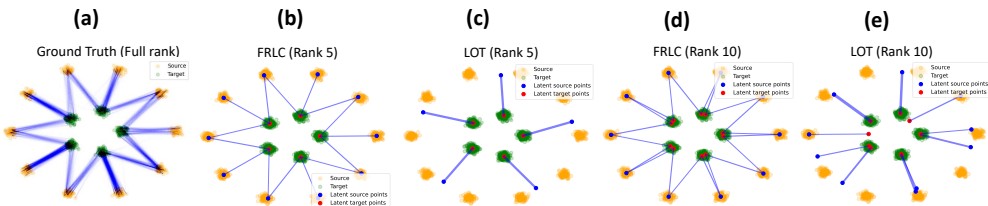

Figure 3: LC-projections of couplings of Gaussians centered on the $5^{\text{th}}$-roots of unity (green) and $10^{\text{th}}$ roots of unity (yellow). (a) Ground-truth full-rank coupling. (b) Non-square rank-5 latent-coupling of FRLC (c) LC-projection barycenters aligned with rank-5 diagonal coupling of LOT Scetbon et al. (2021). (d) Square rank-10 latent coupling of FRLC. (e) Rank-10 diagonal coupling of LOT .

## 4.1 Evaluation of Low-rank Approximations for Balanced OT on Synthetic Data

We first compare the balanced OT version of FRLC with the the low-rank balanced OT algorithm LOT of Scetbon et al. (2021) on a synthetic dataset following Tong et al. (2023). The dataset consists of $m = 1000$ points from two moons and $n = 1000$ points sampled from eight 2D Gaussian densities (Fig. 2a). We solve the Wasserstein problem (1) with cost matrix $C$ computed using the Euclidean distance. The full-rank coupling matrix $P$ has rank 1000, and we compute both FRLC and LOT solutions with rank between 20 and 200. For each rank, we initialize FRLC adapting the deterministic rank-2 initialization proposed in Scetbon et al. (2021) and the random initialization of Alg. 6. We initialize LOT using the rank-2 initialization and two other options in `ott-jax` Cuturi et al. (2022).

We find that FRLC obtains lower transport cost $\langle C, P \rangle_F$ with increasing rank (Fig. 2b) and consistently achieves lower transport cost than LOT across all ranks and all initializations. Specifically, starting both methods at the same rank-2 initialization, FRLC consistently achieves a lower cost than LOT for all ranks. Additionally, we observe smooth convergence of FRLC for both rank-2 initialization and the full-rank random initialization of Alg. 6 (Fig. 5).

We also evaluate FRLC and LOT on two datasets of Gaussian mixtures, one in 2-dimensions and one in 10-dimensions, each with $n = m = 5,000$ points from two mixtures of Gaussians, following Scetbon et al. (2021), with further details in Appendix K. We observe the same trend as the previous simulation for both datasets (Fig. 2c, Fig. 7), with FRLC achieving lower transport costs than LOT across all ranks and all initializations. In addition FRLC has half the runtime of LOT (CPU) – including the setup time of FRLC but excluding the setup time of LOT in `ott-jax` – on datasets of $n = m = 1000$ points from all three datasets with rank $r = 100$ (Table 2). At the same time FRLC achieves lower primal cost $\langle C, P \rangle_F$ with tighter marginals $\|P \mathbf{1}_n - a\|_2$ and $\|P^T \mathbf{1}_m - b\|_2$. Lin et al. (2021) only solves a proxy for the rank-constrained Wasserstein problem, and thus is not the focus of our comparisons. Nevertheless, we verify that on all synthetic experiments that FRLC achieves significantly lower primal OT cost than Latent OT (Table 5).

## 4.2 Interpretation of the Latent Coupling and LC-Projection

We demonstrate the intepretability of the latent coupling $T$ in the LC factorization. In both the LC factorization and factored couplings, the sub-couplings $Q$ and $R$ each have associated barycentric projection operators which coarse-grain input datasets $\mathbf{Z}^{(1)}, \mathbf{Z}^{(2)}$. In particular, the LC projection is defined from the LC factorization as follows.

**Definition 4.1** (LC-Projection). Let $Q \operatorname{diag}(1/g_Q) T \operatorname{diag}(1/g_R) R^{\mathrm{T}}$ be an LC factorization of of a coupling matrix $P \in \Pi_{a,b}(r)$ computed from datasets $\mathbf{Z}^{(1)} \in \mathbb{R}^{n \times d}, \mathbf{Z}^{(2)} \in \mathbb{R}^{m \times d}$, with $T \in \mathbb{R}_+^{r_1 \times r_2}$. The *LC-projections* $\mathbf{Y}^{(1)}$ and $\mathbf{Y}^{(2)}$ of $\mathbf{Z}^{(1)}$ and $\mathbf{Z}^{(2)}$ are $\mathbf{Y}^{(1)} := \operatorname{diag}(1/g_Q) Q^{\mathrm{T}} \mathbf{Z}^{(1)}$, and $\mathbf{Y}^{(2)} := \operatorname{diag}(1/g_R) R^{\mathrm{T}} \mathbf{Z}^{(2)}$.

By interpreting any factored coupling $(Q, R, g)$ as an LC factorization $(Q, R, \operatorname{diag}(g))$, Definition 4.1 describes the barycentric projections for both factorizations. We compare the projections of the coupling computed by FRLC to those of LOT Scetbon et al. (2021) on a dataset containing 1000 samples from 2D-Gaussians centered at the $5^{\text{th}}$-roots of unity and 1000 samples from 2D Gaussians

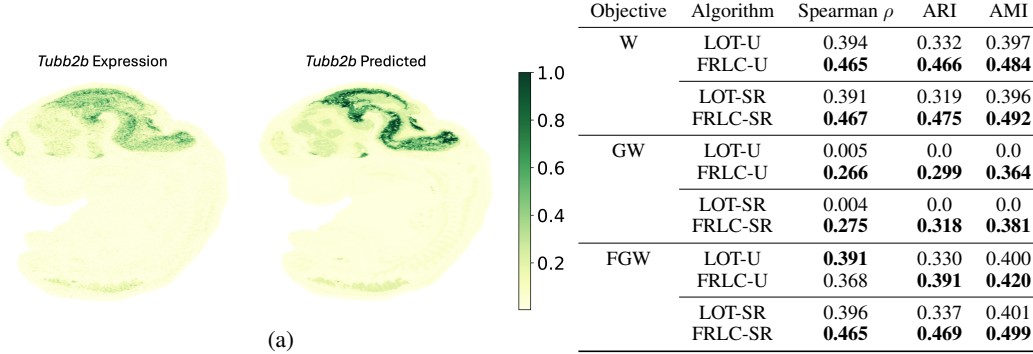

| Objective | Algorithm | Spearman $\rho$ | ARI | AMI |
|---|---|---|---|---|
| W | LOT-U | 0.394 | 0.332 | 0.397 |
|  | FRLC-U | **0.465** | **0.466** | **0.484** |
|  | LOT-SR | 0.391 | 0.319 | 0.396 |
|  | FRLC-SR | **0.467** | **0.475** | **0.492** |
| GW | LOT-U | 0.005 | 0.0 | 0.0 |
|  | FRLC-U | **0.266** | **0.299** | **0.364** |
|  | LOT-SR | 0.004 | 0.0 | 0.0 |
|  | FRLC-SR | **0.275** | **0.318** | **0.381** |
| FGW | LOT-U | **0.391** | 0.330 | 0.400 |
|  | FRLC-U | 0.368 | **0.391** | **0.420** |
|  | LOT-SR | 0.396 | 0.337 | 0.401 |
|  | FRLC-SR | **0.465** | **0.469** | **0.499** |

(a)

Figure 4: (a) Brain marker gene *Tubb2b* expression and FRLC prediction. (b) Comparison of the low-rank unbalanced (LOT-U) algorithm of Scetbon et al. (2023) and FRLC on aligning spatial transcriptomics data. Bold indicates top performing method for each metric on each objective.

centered at the $10^{\text{th}}$-roots of unity (the latter scaled by a factor of two, Fig. 3a). In both cases, the latent coupling $T$ or $\text{diag}(g)$ is visualized as a transport between barycenters. We run FRLC and LOT with ranks $r = 5$ and $r = 10$ to match the number of target and source clusters. In the rank-5 case, FRLC uses a *non-square* latent coupling $T \in \mathbb{R}_+^{10 \times 5}$ which correctly captures the coupling between clusters (Fig. 3(b)), while the LOT rank-5 projection computes barycenters that are outside of the clusters (Fig. 3c). A similar result is observed for square rank-10 latent couplings computed by FRLC (Fig. 3d) and LOT (Fig. 3e) demonstrating that the LOT barycenters in Fig. 3b) are not an artifact of using the lowest rank. We observe similar results on other simulated datasets (Fig. 11).

### 4.3 Evaluation on Spatial Transcriptomics Alignment

We compare FRLC and the algorithm (LOT-U) of Scetbon et al. (2023) (which solves unbalanced low-rank Wasserstein, GW, and FGW problems) on the task of computing an alignment between cells from different time points during mouse embryonic development. Specifically, we compute an alignment between a spatial transcriptomics (ST) dataset of an E11.5 stage mouse embryo and an E12.5 stage mouse embryo Chen et al. (2022). Optimal transport is a popular approach to align single-cell Schiebinger et al. (2019) and spatial trancriptomics datasets Zeira et al. (2022); Liu et al. (2023); Klein et al. (2023). In single-cell transcriptomics, one measures a gene expression vector for each cell, and in spatial transcriptomics one additionally measures the 2D location of each cell. The cost matrix $C$ describes the difference between gene expression vectors and intra-domain cost matrices $A$ and $B$ are derived from the 2D coordinates within each slice. Therefore, OT problems of W, GW, and FGW objectives can be solved and the coupling matrix represents the cell-cell alignment (Appendix M). However, computation of a full-rank OT solution is not feasible in our large-scale dataset: the E11.5 slice has about 30,000 cells while the E12.5 slice has about 50,000 cells.

We evaluate the alignments by assessing performance on two prediction tasks from Scetbon et al. (2023): (1) a *gene expression prediction* task where we predict the expression of a gene in E12.5 from expression of the gene in E11.5 using the alignment; (2) a *cell type prediction task* where we predict the cell types of E12.5 from the cell type clustering of E11.5 (Appendix M). We evaluate the accuracy of the gene expression prediction task through the Spearman correlation $\rho$ between the predicted expression and the ground truth expression of 10 test marker genes. We evaluate the accuracy of the cell type prediction task by computing the Adjusted Rand Index (ARI) and Adjusted Mutual Information (AMI) between the predicted cell types and the cell types derived in the original publication Chen et al. (2022). Being a comparison between different objectives, this relies on downstream metrics. For completeness, we validate the efficacy of FRLC on directly minimizing the balanced Wasserstein cost $\langle C, P \rangle_F$ against Scetbon et al. (2021) in Figure 8.

For a direct comparison, we use FRLC to solve the same unbalanced problems (denoted FRLC-U). We perform an extensive grid search (Appendix M.3) to pick the best hyperparameters (including rank $\ll 30,000$) for all algorithms. Scetbon et al. (2023) previously showed that unbalanced FGW algorithm has the best performance on ST alignment. We find that unbalanced FRLC achieves comparable or better results than the previous state-of-the-art unbalanced low-rank method on all three objectives (Table 4). We also solve a semi-relaxed version of each problem motivated by the

| Method | Factorization | Cost | Variables | Algorithm | Sub-routine for coupling |
|---|---|---|---|---|---|
| Factored Coupling Forrow et al. (2019) | Factored coupling | $k$-Wasserstein barycenter | Anchors & sub-couplings | Lloyd-type | Dykstra's |
| Latent OT Lin et al. (2021) | Latent coupling | Extension of $k$-Wasserstein barycenter | Anchors & sub-couplings | Lloyd-type | Dykstra's |
| LOT Scetbon et al. (2021) | Factored coupling | Primal OT cost | Sub-couplings & inner marginal | Mirror-descent | Dykstra's |
| FRLC (this work) | Latent coupling | Primal OT cost | Sub-couplings | Coordinate mirror-descent | OT |

Table 1: Comparing aspects of low-rank OT methods. Factorization indicates the structure of the inner matrix.

observation that all cells from E12.5 have an ancestor, but not all cells from E11.5 have the same number of descendants due to cell growth and death. Thus the former marginal is tight, and the latter relaxed Halmos et al. (2024). We run both semi-relaxed FRLC (FRLC-SR) and a setting of LOT-U that recovers the semi-relaxed problem (LOT-SR). Semi-relaxed FRLC achieves the best results on all three metrics by a large margin (Table 4). As one example, the expression of *Tubb2b*, a mouse brain marker gene, agreeing with the expression predicted from the semi-relaxed alignment of FRLC (Fig. 4a).

### 4.4 Additional Experiments

We evaluate FRLC on an unsupervised graph partitioning problem Chowdhury & Needham (2021) on four real-world graph datasets Yang & Leskovec (2012); Yin et al. (2017); Banerjee et al. (2013). We benchmark the performance of the semi-relaxed and GW settings of FRLC against (1) GWL Xu et al. (2019), solving a balanced GW problem; (2) SpecGWL Chowdhury & Needham (2021) using the heat kernel on the graph Laplacian as the cost matrix. We find FRLC achieves the better clustering performance than GWL and SpecGWL on 9/12 and 11/12 of the datasets (Table 3 and Appendix L).

## 5 Discussion

We provide comparison of existing low-rank solvers in Table 1. The FRLC algorithm has a number of advantages, including (1) coarsening a full-rank plan $P$ to non-diagonal latent coupling $T$; (2) minimizing the primal OT problem for general cost $C$ rather than a barycenteric problem; (3) optimizing only sub-couplings; and (4) using Sinkhorn alone as the sub-routine for low-rank OT. While we argue these are substantial advantages, FRLC has limitations which warrant follow-up work. In particular, three key limitations of our work, common to the existing low-rank OT algorithms, are: (1) selecting values of the latent coupling ranks; (2) strengthening the convergence criterion; (3) addressing sensitivity to the initialization from non-convexity of the objective. A limitation specific to our work is the selection of the $\tau$ hyperparameter controlling the smoothness of the trajectory. These and other limitations are discussed in Section N of the Appendix. Another direction for further investigation is to better understand what structure LC factorizations capture when the optimal plan is known to have full rank, e.g. when the Monge map exists, as has been explored by Liu et al. (2021).

## 6 Conclusion

We introduce FRLC, an algorithm to compute low-rank optimal transport plan from the latent coupling (LC) factorization. FRLC handles different OT objective costs and relaxations of the marginal constraints. Moreover, the LC factorization provides an interpretable coarse-graining of the full transport plan and its marginals through the mapping $(P, a, b) \rightarrow (T, g_Q, g_R)$. We demonstrate the superior performance of FRLC compared to state-of-the-art low-rank methods on real and synthetic datasets.

## Acknowledgments and Disclosure of Funding

This work is supported by NCI grant U24CA248453 to B.J.R. J.G. gratefully acknowledges support from the Schmidt DataX Fund at Princeton University made possible through a major gift from the Schmidt Futures Foundation.

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

# A  Low-rank optimal transport

## A.1  Low-rank factorizations

**The set of low-rank couplings.**    Given $M \in \mathbb{R}_+^{n \times m}$, the *nonnegative rank* of $M$ is the least number of nonnegative, rank-1 matrices that sum to $M$:

$$\mathrm{rk}_+(M) = \min_{r \geq 1} \left\{ M = \sum_{i=1}^{r} M_i, \text{ such that } \mathrm{rk}(M_i) = 1 \text{ and } M_i \geq 0 \text{ for all } i \right\}.$$

Let $a \in \Delta_n, b \in \Delta_m$ be probability vectors, and let $\Pi_{a,b}(r)$ denote the set of rank-$r$ coupling matrices with marginals $a$ and $b$:

$$\Pi_{a,b}(r) = \{ P \in \mathbb{R}_+^{n \times m} : P^{\mathrm{T}} \mathbf{1}_m = a, \ P \mathbf{1}_n = b, \ \mathrm{rk}_+(P) \leq r \}.$$

To optimize any cost over $\Pi_{a,b}(r)$, one requires a parameterization of this set.

**Factored couplings.**    The *factored coupling* parameterization of $\Pi_{a,b}(r)$ introduced in Forrow et al. (2019), and used by Scetbon et al. (2021); Scetbon & Cuturi (2022); Scetbon et al. (2022, 2023) is

$$\mathsf{FC}_{a,b}(r) := \{ (Q, R, g) \in \mathbb{R}_+^{n \times r} \times \mathbb{R}_+^{m \times r} \times (\mathbb{R}_+^*)^r : Q \in \Pi_{a,g}, R \in \Pi_{b,g} \}.$$

Cohen & Rothblum (1993) show that any $P \in \Pi_{a,b}(r)$ may be decomposed as $P = Q \mathrm{diag}(1/g) R^{\mathrm{T}}$ for some triple $(Q, R, g) \in \mathsf{FC}$. Thus, for cost matrix $C \in \mathbb{R}^{n \times m}$, the general low-rank optimal transport problem is equivalent to an optimization over factored couplings:

$$\min_{P \in \Pi_{a,b}(r)} \langle C, P \rangle_F = \min_{(Q,R,g) \in \mathsf{FC}_{a,b}(r)} \langle C, Q \mathrm{diag}(1/g) R^{\mathrm{T}} \rangle_F. \tag{11}$$

**Latent coupling factorization.**    The *latent coupling* parameterization of $\Pi_{a,b}(r)$ introduced in Lin et al. (2021), and used in the present work is

$$\mathsf{LC}_{a,b}(r) := \{ (Q, R, T) \in \mathbb{R}_+^{n \times r} \times \mathbb{R}_+^{m \times r} \times \mathbb{R}_+^{r \times r} : Q \in \Pi_{a,\cdot}, R \in \Pi_{b,\cdot}, T \in \Pi_{g_Q, g_R} \},$$

where $g_Q, g_R$ are the inner marginals of $Q$ and $R$.

**Latent coupling diagonalization.** The LC-factorization recovers the factorization of Forrow et al. (2019) as a sub-case. While the diagonal factorization of previous works cannot be directly converted to the LC-factorization, the LC-factorization can easily recover the diagonal factorization. In particular, taking $Q' \leftarrow Q \, \mathrm{diag}(1/g_Q) T$ one can refactor

$$P_r = Q \, \mathrm{diag}(1/g_Q) T \, \mathrm{diag}(1/g_R) R^{\mathrm{T}} = Q' \, \mathrm{diag}(1/g_R) R^{\mathrm{T}}$$

or alternatively taking $R' = R \, \mathrm{diag}(1/g_R) T^{\mathrm{T}}$ may refactor as

$$P_r = Q \, \mathrm{diag}(1/g_Q) T \, \mathrm{diag}(1/g_R) R^{\mathrm{T}} = Q \, \mathrm{diag}(1/g_Q) (R')^{\mathrm{T}}$$

So that instead of returning $(Q, R, T)$ one may alternatively return $(Q, R, T) \rightarrow (Q', R, \mathrm{diag}(g_R))$ or $(Q, R, T) \rightarrow (Q, R', \mathrm{diag}(g_Q))$ to recover the Forrow et al. (2019) factorization. An example of this diagonal-conversion is offered in Figure 12.

## A.2  Balanced low-rank optimal transport

**The FRLC optimization problem**    Our optimization problem is over the variables $(Q, R, T)$ and defined as follows:

$$\min_{(Q,R,T) \in \mathsf{LC}_{a,b}(r)} \mathcal{L}_{\mathrm{LC}}(Q, R, T), \tag{12}$$

where our objective function $\mathcal{L}_{\mathrm{LC}}$ is

$$\mathcal{L}_{\mathrm{LC}}(Q, R, T) = \langle C, Q(\mathrm{diag}(1/Q^{\mathrm{T}} \mathbf{1}_n)) T(\mathrm{diag}(1/R^{\mathrm{T}} \mathbf{1}_m)) R^{\mathrm{T}} \rangle \tag{13}$$

Given $(Q, R, T) \in \mathsf{LC}_{a,b}(r)$, sub-couplings $Q$ and $R$ are constrained by:

$$\mathcal{C}_1(a) := \{ (Q, R, T) \in \mathcal{R}_+ : Q \mathbf{1}_r = a \}, \quad \mathcal{C}_1(b) := \{ (Q, R, T) \in \mathcal{R}_+ : R \mathbf{1}_r = b \},$$

while the convex sets constraining the latent coupling matrix $\boldsymbol{T}$ are

$$\mathcal{C}_2(\boldsymbol{g}_Q) := \{(\boldsymbol{Q}, \boldsymbol{R}, \boldsymbol{T}) \in \mathcal{R}_+ : \boldsymbol{T}\mathbf{1}_r = \boldsymbol{g}_Q\}, \quad \mathcal{C}_2(\boldsymbol{g}_R) := \{(\boldsymbol{Q}, \boldsymbol{R}, \boldsymbol{T}) \in \mathcal{R}_+ : \boldsymbol{T}^{\mathrm{T}}\mathbf{1}_r = \boldsymbol{g}_R\},$$

where $\boldsymbol{g}_Q = \boldsymbol{Q}^{\mathrm{T}}\mathbf{1}_n$ and $\boldsymbol{g}_R = \boldsymbol{R}^{\mathrm{T}}\mathbf{1}_m$ as per Definition 3.1. Under these definitions, $\mathsf{LC}_{\boldsymbol{a},\boldsymbol{b}}(r) = \mathcal{C}_1 \cap \mathcal{C}_2$, where

$$\mathcal{C}_1 = \mathcal{C}_1(\boldsymbol{a}) \cap \mathcal{C}_1(\boldsymbol{b}) \quad \text{and} \quad \mathcal{C}_2 = \mathcal{C}_2(\boldsymbol{g}_Q) \cap \mathcal{C}_2(\boldsymbol{g}_R) \tag{14}$$

To solve (12), we separate the variables into two "blocks" of variables, $(\boldsymbol{Q}, \boldsymbol{R})$ and $\boldsymbol{T}$, and perform two block updates per iteration, as follows. Let $(\gamma_k)_{k \geq 0}$ be a positive sequence of stepsizes. Suppose we have $(\boldsymbol{Q}_k, \boldsymbol{R}_k, \boldsymbol{T}_k) \in \mathcal{C}$. We update the first variable block $(\boldsymbol{Q}, \boldsymbol{R})$ by taking a locally optimal (mirror descent) update step, while the second variable block $\boldsymbol{T}$ is held fixed:

$$(\boldsymbol{Q}_{k+1}, \boldsymbol{R}_{k+1}) \leftarrow \underset{(\boldsymbol{Q},\boldsymbol{R}):(\boldsymbol{Q},\boldsymbol{R},\boldsymbol{T}_k)\in\mathcal{C}_1}{\arg\min} \langle(\boldsymbol{Q},\boldsymbol{R}), \nabla_{(\boldsymbol{Q},\boldsymbol{R})}\mathcal{L}_{\mathrm{LC}}\rangle + \frac{1}{\gamma_k}\mathrm{KL}((\boldsymbol{Q},\boldsymbol{R})\|(\boldsymbol{Q}_k,\boldsymbol{R}_k)) \tag{15}$$

Here, we slightly abused the notation by putting $(\boldsymbol{Q}, \boldsymbol{R})$ inside an inner product. The triple $(\boldsymbol{Q}_{k+1}, \boldsymbol{R}_{k+1}, \boldsymbol{T}_k)$ produced by this update lies in $\mathcal{C}_1$. Next, we update $\boldsymbol{T}$, the second variable block, by taking another locally optimal (mirror descent) step, while the first variable block is held fixed.

$$\boldsymbol{T}_{k+1} \leftarrow \underset{\boldsymbol{T}:(\boldsymbol{Q}_{k+1},\boldsymbol{R}_{k+1},\boldsymbol{T})\in\mathcal{C}_2}{\arg\min} \langle\boldsymbol{T}, \nabla_{\boldsymbol{T}}\mathcal{L}_{\mathrm{LC}}\rangle + \frac{1}{\gamma_k}\mathrm{KL}(\boldsymbol{T}\|\boldsymbol{T}_k). \tag{16}$$

By construction, the triple $(\boldsymbol{Q}_{k+1}, \boldsymbol{R}_{k+1}, \boldsymbol{T}_{k+1}) \in \mathcal{C}_2$. However, because the set $\mathcal{C}_1$ only constrains the first variable block $\boldsymbol{Q}_{k+1}, \boldsymbol{R}_{k+1}$, we have that $(\boldsymbol{Q}_{k+1}, \boldsymbol{R}_{k+1}, \boldsymbol{T}_{k+1}) \in \mathcal{C}_2$, and hence $(\boldsymbol{Q}_{k+1}, \boldsymbol{R}_{k+1}, \boldsymbol{T}_{k+1}) \in \mathcal{C}$. Thus, each iteration produces a feasible triple $(\boldsymbol{Q}_{k+1}, \boldsymbol{R}_{k+1}, \boldsymbol{T}_{k+1})$ through a pair of locally optimal block updates.

**The LOT optimization problem Scetbon et al. (2021)** For comparison, recall the optimization problem that is solved in the LOT framework:

$$\min_{(\boldsymbol{Q},\boldsymbol{R},\boldsymbol{g})\in\widetilde{\mathcal{C}}} \mathcal{L}_{\mathrm{LOT}}, \tag{17}$$

where the objective function $\mathcal{L}_{\mathrm{LOT}}$ is

$$\mathcal{L}_{\mathrm{LOT}} := \langle\boldsymbol{C}, \boldsymbol{Q}\mathrm{diag}(1/\boldsymbol{g})\boldsymbol{R}^{\mathrm{T}}\rangle \tag{18}$$

and where $\widetilde{\mathcal{C}} = \widetilde{\mathcal{C}}_1 \cap \widetilde{\mathcal{C}}_2$ with:

$$\widetilde{\mathcal{C}}_1 := \{(\boldsymbol{Q}, \boldsymbol{R}, \boldsymbol{g}) \in \mathbb{R}_+^{n\times r} \times \mathbb{R}_+^{m\times r} \times (\mathbb{R}_+^*)^r : \boldsymbol{Q}\mathbf{1}_r = \boldsymbol{a}, \boldsymbol{R}\mathbf{1}_r = \boldsymbol{b}\}$$

$$\widetilde{\mathcal{C}}_2 := \{(\boldsymbol{Q}, \boldsymbol{R}, \boldsymbol{g}) \in \mathbb{R}_+^{n\times r} \times \mathbb{R}_+^{m\times r} \times \mathbb{R}_+^r : \boldsymbol{Q}^{\mathrm{T}}\mathbf{1}_n = \boldsymbol{g} = \boldsymbol{R}^{\mathrm{T}}\mathbf{1}_m\}.$$

Here, there are also three optimization variables $(\boldsymbol{Q}, \boldsymbol{R}, \boldsymbol{g})$, but they are updated *together* in each iteration of LOT. That is, given feasible $(\boldsymbol{Q}_k, \boldsymbol{R}_k, \boldsymbol{g}_k) \in \mathcal{C}$, an iteration of LOT updates this triple via

$$(\boldsymbol{Q}_{k+1}, \boldsymbol{R}_{k+1}, \boldsymbol{g}_{k+1}) := \underset{(\boldsymbol{Q},\boldsymbol{R},\boldsymbol{g})\in\widetilde{\mathcal{C}}}{\arg\min} \langle(\boldsymbol{Q},\boldsymbol{R},\boldsymbol{g}), \nabla_{(\boldsymbol{Q},\boldsymbol{R},\boldsymbol{g})}\mathcal{L}_{\mathrm{LOT}}\rangle + \frac{1}{\gamma_k}\mathrm{KL}((\boldsymbol{Q},\boldsymbol{R},\boldsymbol{g})\|(\boldsymbol{Q}_k,\boldsymbol{R}_k,\boldsymbol{g}_k)),$$

where $(\gamma_k)_{k\geq 0}$ is again a positive sequence of stepsizes. Scetbon et al. (2021) then compute the unconstrained argmin across all variables to yield a set of unconstrained kernels $(\boldsymbol{K}_Q, \boldsymbol{K}_R, \boldsymbol{k}_g)$, using Dykstra to jointly project the unconstrained update onto the intersection $\widetilde{\mathcal{C}}$ of the constraint sets.

**OT subroutine in FRLC** To see why we do not need Dykstra in the FRLC scheme, observe that the update (15) of variables $(\boldsymbol{Q}, \boldsymbol{R})$ can be equivalently expressed as

$$(\boldsymbol{Q}_{k+1}, \boldsymbol{R}_{k+1}) \leftarrow \underset{\boldsymbol{Q},\boldsymbol{R}:(\boldsymbol{Q},\boldsymbol{R},\boldsymbol{T}_k)\in\mathcal{C}_1}{\arg\min} \langle(\boldsymbol{Q},\boldsymbol{R}), \nabla_{(\boldsymbol{Q},\boldsymbol{R})}\mathcal{L}_{\mathrm{LC}}\rangle + \frac{1}{\gamma_k}\mathrm{KL}((\boldsymbol{Q},\boldsymbol{R})\|(\boldsymbol{Q}_k,\boldsymbol{R}_k)),$$

Thus, even though the pair $(\boldsymbol{Q}, \boldsymbol{R})$ is being updated at this step, solving for $(\boldsymbol{Q}_{k+1}, \boldsymbol{R}_{k+1})$ above is equivalent to updating each individually because $\boldsymbol{Q}$ and $\boldsymbol{R}$ do not share an inner marginal:

$$\boldsymbol{Q}_{k+1} \leftarrow \underset{\boldsymbol{Q}:\boldsymbol{Q}\mathbf{1}_r=\boldsymbol{a}}{\arg\min} \langle\boldsymbol{Q}, \nabla_{\boldsymbol{Q}}\mathcal{L}_{\mathrm{LC}}\rangle + \frac{1}{\gamma_k}\mathrm{KL}(\boldsymbol{Q}\|\boldsymbol{Q}_k) \tag{19}$$

$$R_{k+1} \leftarrow \underset{R \,:\, R\mathbf{1}_r = b}{\arg\min} \langle R, \nabla_R \mathcal{L}_{\text{LC}} \rangle + \frac{1}{\gamma_k} \text{KL}(R \| R_k) \tag{20}$$

We choose to add the regularization $\tau \text{KL}((Q^{\text{T}}\mathbf{1}_n, R^{\text{T}}\mathbf{1}_m) \| (Q_k^{\text{T}}\mathbf{1}_n, R_k^{\text{T}}\mathbf{1}_m))$ to turn each update here into an entropy-regularized semi-relaxed optimal transport problem, and to ensure $\beta$-smoothness:

$$Q_{k+1} \leftarrow \underset{Q \,:\, Q\mathbf{1}_r = a}{\arg\min} \langle Q, \nabla_Q \mathcal{L}_{\text{LC}} \rangle + \frac{1}{\gamma_k} \text{KL}(Q \| Q_k) + \tau \text{KL}(Q^{\text{T}}\mathbf{1}_n \| Q_k^{\text{T}}\mathbf{1}_n) \tag{21}$$

$$R_{k+1} \leftarrow \underset{R \,:\, R\mathbf{1}_r = b}{\arg\min} \langle R, \nabla_R \mathcal{L}_{\text{LC}} \rangle + \frac{1}{\gamma_k} \text{KL}(R \| R_k) + \tau \text{KL}(R^{\text{T}}\mathbf{1}_m \| R_k^{\text{T}}\mathbf{1}_m) \tag{22}$$

After updating $Q$ and $R$, the update on $T$ then follows a similar form:

$$T_{k+1} \leftarrow \underset{T \,:\, (Q_{k+1}, R_{k+1}, T) \in \mathcal{C}_2}{\arg\min} \langle T, \nabla_T \mathcal{L}_{\text{LC}} \rangle + \frac{1}{\gamma_k} \text{KL}(T \| T_k), \tag{23}$$

leading to balanced constraints on $T$ of the form $T^{\text{T}}\mathbf{1}_r = Q_{k+1}^{\text{T}}\mathbf{1}_n$ and $T\mathbf{1}_r = R_{k+1}^{\text{T}}\mathbf{1}_m$, allowing the problem to be solved by Sinkhorn.

Importantly, there are no constraints in $\mathcal{C}_1$ involving both $Q$ and $R$, which is what allows the optimization to split in this way. If there were such constraints, we would have needed to use Dykstra to update the pair $(Q, R)$. Because our update scheme is equivalent to the three updates of individual variables given in (21), (22), (23), we can solve for each update using optimal transport. As $Q$ and $R$ are not required to match the inner marginals exactly, the OT problems associated to $Q$ and $R$ are semi-relaxed by construction.

The separation of our block updates into a step where $(Q, R, T_k) \in \mathcal{C}_1$ and $(Q_{k+1}, R_{k+1}, T) \in \mathcal{C}_2$ allows us to entirely remove the optimization over inner marginals $g_Q$ and $g_R$, as done in all previous works on low-rank optimal transport which optimize $g$ explicitly as both a variable and a constraint of the optimization Scetbon et al. (2021, 2023); Scetbon & Cuturi (2022). If one were to introduce an extended loss in the style of previous works which adds $g_Q$ and $g_R$ as variables in the form $\mathcal{H}(Q, R, T, g_Q, g_R) = \langle Q\text{diag}(1/g_Q)T\text{diag}(1/g_R)R^{\text{T}}, C \rangle_F$, one observes an equivalence to simply taking a semi-relaxed projection.

**Lemma A.1.** *Define the function* $\mathcal{H}(Q, R, T, g_Q, g_R) = \langle C, Q\text{diag}(1/g_Q)T\text{diag}(1/g_R)R^{\text{T}} \rangle_F$ *and let* $\mathcal{L}_{\text{LC}}(Q, R, T)$ *be as in* (13). *One has the following equivalence:*

$$\underset{g_R \in \Delta_r, g_Q \in \Delta_r, Q \in \Pi_{a, g_Q}, R \in \Pi_{b, g_R}}{\min} \mathcal{H}(Q, R, T_k, g_Q, g_R) = \underset{(Q, R, T_k) \in \mathcal{C}_1}{\min} \mathcal{L}_{\text{LC}}(Q, R, T_k) \tag{24}$$

*Thus the semi-relaxed projections yield locally optimal inner marginals.*

*Proof.* To see why this is true, notice that, so long as the outer marginals are tightly satisfied $Q\mathbf{1}_r = a$ and $R\mathbf{1}_r = b$ for $Q \geq \mathbf{0}_{n \times r}, R \geq \mathbf{0}_{m \times r}$, one has

$$\sum_i g_{R,i} = \sum_i \langle R_{i,.}^{\text{T}}, \mathbf{1}_m \rangle = \sum_i \langle R_{.,i}, \mathbf{1}_m \rangle = \sum_\ell \sum_i R_{\ell,i} = \sum_\ell b_\ell = 1$$

and

$$\sum_i g_{Q,i} = \sum_i \langle Q_{i,.}^{\text{T}}, \mathbf{1}_n \rangle = \sum_i \langle Q_{.,i}, \mathbf{1}_m \rangle = \sum_\ell \sum_i Q_{\ell,i} = \sum_\ell a_\ell = 1$$

Therefore, the inner marginals $g_Q^* = (Q^*)^{\text{T}}\mathbf{1}_n$ and $g_R^* = (R^*)^{\text{T}}\mathbf{1}_m$ induced by the optimal $Q^*, R^*$ of the optimization problem on the right hand side satisfy the constraints $g_R^* \in \Delta_r, g_Q^* \in \Delta_r$ on the left hand side, so the two minimums coincide. □

This implies that an extra optimization for $g_Q$ and $g_R$ is unnecessary in a coordinate update which alternates $(Q, R)$ and $T$. If we did not perform a block-update, in the form of standard MD in the objective described above, we would encounter some difficulty. In particular $g_Q$ and $g_R$ would optimized with the constraint $g_Q \in \Delta_r$ and $g_R \in \Delta_r$, and would concurrently constrain all of the other variables as $Q^{\text{T}}\mathbf{1}_n = g_Q, T\mathbf{1}_r = g_Q$, and $R^{\text{T}}\mathbf{1}_m = g_R, T^{\text{T}}\mathbf{1}_r = g_R$.

# B  Block-Coordinate steps for the OT sub-problems

We use a latent non-diagonal coupling instead of an inner diagonal coupling $\operatorname{diag}(\boldsymbol{g})$ of the form of Forrow et al. (2019). This allows us to loosen the constraint that the inner marginals have to be joined by a common coupling $\boldsymbol{Q}^{\mathrm{T}}\mathbf{1}_n = \boldsymbol{R}^{\mathrm{T}}\mathbf{1}_m = \boldsymbol{g}$. The fundamental advantage of this choice is that we can decouple the convex-optimization problem for $(\boldsymbol{Q}, \boldsymbol{R}, \boldsymbol{T})$ entirely. One can simply solve for the optimal $\boldsymbol{Q}$ and $\boldsymbol{R}$ *independently*, yield the associated inner marginals for each $\boldsymbol{Q}^{\mathrm{T}}\mathbf{1}_n = \boldsymbol{g_Q}$ and $\boldsymbol{R}^{\mathrm{T}}\mathbf{1}_m = \boldsymbol{g_R}$, and then find the optimal $\boldsymbol{T}$ which links the two. This link is provided by the aforementioned form of the problem, where:

$$\boldsymbol{P} = \boldsymbol{QXR}^{\mathrm{T}}$$

For $\boldsymbol{Q}, \boldsymbol{R}$ in either the appropriate set of couplings or a relaxation thereof (which we will describe shortly). $\boldsymbol{X}$ is related to $\boldsymbol{T}$ by:

$$\boldsymbol{X} = \operatorname{diag}(1/\boldsymbol{g_Q})\boldsymbol{T}\operatorname{diag}(1/\boldsymbol{g_R})$$

And, $\boldsymbol{T} \in \Pi_{\boldsymbol{g_Q},\boldsymbol{g_R}}$ consistently for all cases. As the semi-relaxed case is intermediate between fully-relaxed and balanced, it has ideas which generalize to both directly. As such, we use it as the leading example again. As in Scetbon et al. (2021), we take proximal-steps of the form:

$$\min_{\boldsymbol{\zeta}}\langle\nabla\mathcal{L}(\boldsymbol{\zeta})\,|_{\boldsymbol{\zeta}_k}, \boldsymbol{\zeta}\rangle_F + \frac{1}{\gamma_k}\mathrm{KL}(\boldsymbol{\zeta}\|\boldsymbol{K}^{(k)})$$

Where these steps are now in block-wise fashion on $(\boldsymbol{Q}, \boldsymbol{R})$ and $\boldsymbol{T}$, rather than joint. One may identify for each sub-factor in $(\boldsymbol{Q}, \boldsymbol{R})$ and $\boldsymbol{T}$ a linearized gradient as before, which yields a set of objectives which each solve an independent optimal-transport for the sub-factors. In particular, we have that:

$$\langle\boldsymbol{QXR}^{\mathrm{T}}, \boldsymbol{C}\rangle_F = \mathrm{Tr}\left[\boldsymbol{QXR}^{\mathrm{T}}\boldsymbol{C}^{\mathrm{T}}\right] = \langle\boldsymbol{CRX}^{\mathrm{T}}, \boldsymbol{Q}\rangle_F$$
$$\langle\boldsymbol{QXR}^{\mathrm{T}}, \boldsymbol{C}\rangle_F = \mathrm{Tr}\left[\boldsymbol{QXR}^{\mathrm{T}}\boldsymbol{C}^{\mathrm{T}}\right] = \langle\boldsymbol{C}^{\mathrm{T}}\boldsymbol{QX}, \boldsymbol{R}\rangle_F$$
$$\langle\boldsymbol{QXR}^{\mathrm{T}}, \boldsymbol{C}\rangle_F = \mathrm{Tr}\left[\boldsymbol{R}^{\mathrm{T}}\boldsymbol{C}^{\mathrm{T}}\boldsymbol{QX}\right] = \langle\boldsymbol{Q}^{\mathrm{T}}\boldsymbol{CR}, \boldsymbol{X}\rangle_F$$

A linearization in the left-slot of the inner product as $\langle\boldsymbol{Q}, \boldsymbol{CRX}(\boldsymbol{Q}_k)^{\mathrm{T}}\rangle := \langle\boldsymbol{Q}, \boldsymbol{CRX}^{\mathrm{T}}\rangle$ or $\langle\boldsymbol{C}^{\mathrm{T}}\boldsymbol{QX}(\boldsymbol{R}_k), \boldsymbol{R}\rangle_F := \langle\boldsymbol{C}^{\mathrm{T}}\boldsymbol{QX}, \boldsymbol{R}\rangle_F$ is common practice for quadratic problems. In this case, the directional derivative of $\boldsymbol{Q}$ and $\boldsymbol{R}$ in the matrix-direction $\boldsymbol{V}$ are respectively:

$$D\langle\boldsymbol{CRX}^{\mathrm{T}}, \boldsymbol{Q}\rangle_F \circ (\boldsymbol{V}) = \langle\boldsymbol{CRX}^{\mathrm{T}}, \boldsymbol{V}\rangle_F \implies \nabla_{\boldsymbol{Q}}\mathcal{L} = \boldsymbol{CRX}^{\mathrm{T}}$$
$$D\langle\boldsymbol{C}^{\mathrm{T}}\boldsymbol{QX}, \boldsymbol{R}\rangle_F \circ (\boldsymbol{V}) = \langle\boldsymbol{C}^{\mathrm{T}}\boldsymbol{QX}, \boldsymbol{V}\rangle_F \implies \nabla_{\boldsymbol{R}}\mathcal{L} = \boldsymbol{C}^{\mathrm{T}}\boldsymbol{QX}$$

Without this linearization assumption on $\boldsymbol{X}$, the full gradient may be evaluated as well. In particular, we note that for $\operatorname{diag}^{-1}(\cdot)$ the matrix-to-vector extraction of the diagonal, the directional derivative on $\boldsymbol{Q}$ is:

$$D\langle\boldsymbol{C}, \boldsymbol{QXR}^{\mathrm{T}}\rangle_F \circ \boldsymbol{V} = \langle\boldsymbol{CRX}^{\mathrm{T}}, \boldsymbol{V}\rangle_F + \langle\boldsymbol{C}, \boldsymbol{QDX}^{\mathrm{T}} \circ (\boldsymbol{V})\boldsymbol{R}^{\mathrm{T}}\rangle_F$$
$$= \langle\boldsymbol{CRX}^{\mathrm{T}} - \mathbf{1}_n\operatorname{diag}^{-1}((\boldsymbol{CRX}^{\mathrm{T}})^{\mathrm{T}}\boldsymbol{Q}\operatorname{diag}(1/\boldsymbol{g_Q}))^{\mathrm{T}}, \boldsymbol{V}\rangle_F$$

Thus, without the linearization assumption one may use product rule on $\boldsymbol{X}$ as an implicit function of $\boldsymbol{Q}$ (resp. $\boldsymbol{R}$) to take the total derivative:

$$\nabla_{\boldsymbol{Q}}\mathcal{L}_{\mathrm{FRLC}} = \boldsymbol{CRX}^{\mathrm{T}} - \mathbf{1}_n\operatorname{diag}^{-1}((\boldsymbol{CRX}^{\mathrm{T}})^{\mathrm{T}}\boldsymbol{Q}\operatorname{diag}(1/\boldsymbol{g_Q}))^{\mathrm{T}}$$

Likewise for $\boldsymbol{R}$,

$$D\langle\boldsymbol{C}, \boldsymbol{QXR}^{\mathrm{T}}\rangle_F \circ \boldsymbol{V} = \langle\boldsymbol{C}^{\mathrm{T}}\boldsymbol{QX}, \boldsymbol{V}\rangle_F + \langle\boldsymbol{C}, \boldsymbol{QDX} \circ (\boldsymbol{V})\boldsymbol{R}^{\mathrm{T}}\rangle_F$$
$$= \langle\boldsymbol{C}^{\mathrm{T}}\boldsymbol{QX} - \mathbf{1}_m\operatorname{diag}^{-1}(\operatorname{diag}(1/\boldsymbol{g_R})\boldsymbol{R}^{\mathrm{T}}\boldsymbol{C}^{\mathrm{T}}\boldsymbol{QX})^{\mathrm{T}}, \boldsymbol{V}\rangle_F,$$

and so

$$\nabla_{\boldsymbol{R}}\mathcal{L}_{\mathrm{FRLC}} = \boldsymbol{C}^{\mathrm{T}}\boldsymbol{QX} - \mathbf{1}_m\operatorname{diag}^{-1}(\operatorname{diag}(1/\boldsymbol{g_R})\boldsymbol{R}^{\mathrm{T}}\boldsymbol{C}^{\mathrm{T}}\boldsymbol{QX})^{\mathrm{T}}.$$

Both the W and GW problem have these rank-one perturbations of the gradient from the derivative with respect to $\boldsymbol{X}$.

Lastly, owing to the block-coordinate updates which fix $(\boldsymbol{Q},\boldsymbol{R})$ preceding the update on $\boldsymbol{T}$, the derivative on $\boldsymbol{T}$ follows directly by chain rule on $\boldsymbol{X}$:

$$D\langle\boldsymbol{Q}^{\mathrm{T}}\boldsymbol{C}\boldsymbol{R},\boldsymbol{X}\rangle_F \circ (\boldsymbol{V}) = \langle\boldsymbol{Q}^{\mathrm{T}}\boldsymbol{C}\boldsymbol{R},\boldsymbol{V}\rangle_F \implies \nabla_{\boldsymbol{X}}\mathcal{L} = \boldsymbol{Q}^{\mathrm{T}}\boldsymbol{C}\boldsymbol{R}$$
$$\nabla_{\boldsymbol{T}}\mathcal{L} = \mathrm{diag}(1/\boldsymbol{g_Q})\boldsymbol{Q}^{\mathrm{T}}\boldsymbol{C}\boldsymbol{R}\,\mathrm{diag}(1/\boldsymbol{g_R})$$

Let $(\gamma_k)$ be a sequence of step sizes and consider the first-order conditions required for the proximal step as before. From these we have the updated proximal-step updates:

$$\boldsymbol{K}_{\boldsymbol{Q}}^{(k)} \leftarrow \min_{\boldsymbol{Q}}\langle\boldsymbol{C}\boldsymbol{R}\boldsymbol{X}^{\mathrm{T}},\boldsymbol{Q}\rangle_F + \frac{1}{\gamma_k}\mathrm{KL}(\boldsymbol{Q}_k\|\boldsymbol{K}_{\boldsymbol{Q}}^{(k)})$$

$$\boldsymbol{K}_{\boldsymbol{R}}^{(k)} \leftarrow \min_{\boldsymbol{R}}\langle\boldsymbol{C}^{\mathrm{T}}\boldsymbol{Q}\boldsymbol{X},\boldsymbol{R}\rangle_F + \frac{1}{\gamma_k}\mathrm{KL}(\boldsymbol{R}_k\|\boldsymbol{K}_{\boldsymbol{R}}^{(k)})$$

$$\boldsymbol{K}_{\boldsymbol{T}}^{(k)} \leftarrow \min_{\boldsymbol{T}}\langle\boldsymbol{Q}^{\mathrm{T}}\boldsymbol{C}\boldsymbol{R},\boldsymbol{X}\rangle_F + \frac{1}{\gamma_k}\mathrm{KL}(\boldsymbol{T}_k\|\boldsymbol{K}_{\boldsymbol{T}}^{(k)}),$$

with kernels $\boldsymbol{K}_{\zeta_j}^{(k)}$, for $j=1,2,3$ given by

$$\boldsymbol{K}_{\boldsymbol{Q}}^{(k)} := \boldsymbol{Q}_k \odot \exp(-\gamma_k\boldsymbol{C}\boldsymbol{R}_k\boldsymbol{X}_k^{\mathrm{T}})$$
$$\boldsymbol{K}_{\boldsymbol{R}}^{(k)} := \boldsymbol{R}_k \odot \exp(-\gamma_k\boldsymbol{C}^{\mathrm{T}}\boldsymbol{Q}_k\boldsymbol{X}_k)$$
$$\boldsymbol{K}_{\boldsymbol{T}}^{(k)} := \boldsymbol{T}_k \odot \exp(-\gamma_k\mathrm{diag}(\boldsymbol{g}_Q^{-1})\boldsymbol{Q}^{\mathrm{T}}\boldsymbol{C}\boldsymbol{R}\,\mathrm{diag}(\boldsymbol{g}_R^{-1}))$$

Or, dropping the linearization assumption on $(\boldsymbol{Q},\boldsymbol{R})$ the updates are: $\boldsymbol{K}_{\zeta_j}^{(k)}$, for $j=1,2$ given by

$$\boldsymbol{K}_{\boldsymbol{Q}}^{(k)} := \boldsymbol{Q}_k \odot \exp(-\gamma_k(\boldsymbol{C}\boldsymbol{R}_k\boldsymbol{X}_k^{\mathrm{T}} - \boldsymbol{1}_n\mathrm{diag}^{-1}((\boldsymbol{C}\boldsymbol{R}_k\boldsymbol{X}_k^{\mathrm{T}})^{\mathrm{T}}\boldsymbol{Q}_k\mathrm{diag}(1/\boldsymbol{g}_{Q_k}))^{\mathrm{T}}))$$

$$\boldsymbol{K}_{\boldsymbol{R}}^{(k)} := \boldsymbol{R}_k \odot \exp(-\gamma_k(\boldsymbol{C}^{\mathrm{T}}\boldsymbol{Q}_k\boldsymbol{X}_k - \boldsymbol{1}_m\mathrm{diag}^{-1}(\mathrm{diag}(1/\boldsymbol{g}_R)\boldsymbol{R}_k^{\mathrm{T}}\boldsymbol{C}^{\mathrm{T}}\boldsymbol{Q}_k\boldsymbol{X}_k)^{\mathrm{T}}))$$

The first projection is onto the set that satisfies the marginal constraint $\boldsymbol{R}\boldsymbol{1}_r = \boldsymbol{b}$. In particular, following the discussion above, one has the coordinate-MD step:

$$\min_{(\boldsymbol{Q},\boldsymbol{R},\boldsymbol{T})} \quad \frac{1}{\gamma_k}\mathrm{KL}\left((\boldsymbol{Q},\boldsymbol{R},\boldsymbol{T})\,\|\,(\boldsymbol{K}_{\boldsymbol{Q}},\boldsymbol{K}_{\boldsymbol{R}},\boldsymbol{K}_{\boldsymbol{T}})\right) + \tau\mathrm{KL}(\boldsymbol{Q}\boldsymbol{1}_r\|\boldsymbol{a}) \tag{25}$$
$$s.t. \quad \boldsymbol{R}\boldsymbol{1}_r = \boldsymbol{b}$$

As before, there is no difference from the previous case, where one takes the unconstrained projection $\boldsymbol{R} = \mathrm{diag}(\boldsymbol{b}/\boldsymbol{K}_{\boldsymbol{R}}\boldsymbol{1}_r)\boldsymbol{K}_{\boldsymbol{R}}$. To generalize this, we also consider adding a soft-constraint on the inner marginal of $\boldsymbol{R}$ to be near that of the previous iteration. In particular, we consider the problem:

$$\min_{(\boldsymbol{Q},\boldsymbol{R},\boldsymbol{T})} \quad \frac{1}{\gamma_k}\mathrm{KL}((\boldsymbol{Q},\boldsymbol{R},\boldsymbol{T})\|(\boldsymbol{K}_{\boldsymbol{Q}},\boldsymbol{K}_{\boldsymbol{R}},\boldsymbol{K}_{\boldsymbol{T}}))$$
$$+ \tau\mathrm{KL}(\boldsymbol{Q}\boldsymbol{1}_r\|\boldsymbol{a}) + \tau\mathrm{KL}(\boldsymbol{R}^{\mathrm{T}}\boldsymbol{1}_m\|\boldsymbol{g}_R^{(k-1)} \equiv \boldsymbol{R}_{k-1}^{\mathrm{T}}\boldsymbol{1}_m) \tag{26}$$
$$s.t. \quad \boldsymbol{R}\boldsymbol{1}_r = \boldsymbol{b}$$

Which yields the relaxed solution of $\boldsymbol{R} = \mathrm{SR}^{\mathrm{R}}\text{-}\textit{projection}(\boldsymbol{K}_{\boldsymbol{R}},\gamma_k,\tau,\boldsymbol{b},\boldsymbol{g}_R^{(k-1)})$ which generalizes the original projection and is equivalent to it for $\tau = 0$. This regularization is essential, as it ensures $\beta$-smoothness of the objective. For $\boldsymbol{Q}$, as the constraint on $\boldsymbol{g}$ is fully relaxed, the Lagrange multiplier $\boldsymbol{\lambda}_1 = \boldsymbol{0}$ entirely, such that the problem 40 now becomes fully-unconstrained:

$$\inf_{\boldsymbol{Q}}\left(\frac{1}{\gamma_k}\mathrm{KL}(\boldsymbol{Q}\|\boldsymbol{K}_{\boldsymbol{Q}}) + \tau\mathrm{KL}(\boldsymbol{Q}\boldsymbol{1}_r\|\boldsymbol{a})\right) \tag{27}$$

To generalize this solution, we again consider adding a soft-regularization on the inner marginal of $\boldsymbol{Q}$, where we consider the alternate problem:

$$\inf_{\boldsymbol{Q}}\left(\frac{1}{\gamma_k}\mathrm{KL}(\boldsymbol{Q}\|\boldsymbol{K}_{\boldsymbol{Q}}) + \tau\mathrm{KL}(\boldsymbol{Q}\boldsymbol{1}_r\|\boldsymbol{a}) + \tau\mathrm{KL}(\boldsymbol{Q}^{\mathrm{T}}\boldsymbol{1}_n\|\boldsymbol{g}_{\boldsymbol{Q}}^{(k-1)} \equiv \boldsymbol{Q}_{k-1}^{\mathrm{T}}\boldsymbol{1}_n)\right) \tag{28}$$

Which trivially recovers the original for $\tau = 0$. This form has a solution given by an unbalanced optimal transport with kernel $\boldsymbol{K_Q}$. We see these two, convex problems in sequence give *independent* optimal solutions for $\boldsymbol{Q}$ and $\boldsymbol{R}$ as the two matrices are not required to share an inner marginal. The last step is to link them via $\boldsymbol{T}$, corresponding to the projection of $(\boldsymbol{Q}_k, \boldsymbol{R}_k, \boldsymbol{T})$ onto the set of valid rank-$r$ couplings $\min_{(\boldsymbol{Q}_k, \boldsymbol{R}_k, \boldsymbol{T}) \in C} \mathcal{L}_{\mathrm{LC}}(\boldsymbol{Q}_k, \boldsymbol{R}_k, \boldsymbol{T}) := \min_{\boldsymbol{T} \in \Pi(\boldsymbol{g}_Q, \boldsymbol{g}_R)} \mathcal{L}_{\mathrm{LC}}(\boldsymbol{Q}_k, \boldsymbol{R}_k, \boldsymbol{T})$. We verify in C that if $\boldsymbol{T} \in \Pi_{\boldsymbol{g}_Q = \boldsymbol{Q}^{\mathrm{T}} \mathbf{1}_n, \boldsymbol{g}_R = \boldsymbol{R}^{\mathrm{T}} \mathbf{1}_m}$ then $\boldsymbol{P} \in \Pi_{\boldsymbol{a}, \boldsymbol{b}}$. As such, one does not require any projection onto the intersection of convex sets, as done in Scetbon et al. (2021, 2023) via the Dykstra projection algorithm Dykstra (1983). Alternating a coordinate-MD step in $(\boldsymbol{Q}, \boldsymbol{R})$ and a step on $\boldsymbol{T}$, one not only minimizes the objective in an alternating fashion but remains in the feasible set without the need for projection algorithms beyond Sinkhorn. As such, the final linking step is done via:

$$
\begin{aligned}
\min_{\boldsymbol{T}} \quad & \frac{1}{\gamma_k} \mathrm{KL}(\boldsymbol{T} \| \boldsymbol{K_T}) \\
s.t. \quad & \boldsymbol{T} \mathbf{1}_r = \boldsymbol{g}_Q, \boldsymbol{T}^{\mathrm{T}} \mathbf{1}_r = \boldsymbol{g}_R
\end{aligned}
\tag{29}
$$

This formulation amounts to solving a *balanced* Sinkhorn problem on $\boldsymbol{T}$ with respect to the proximal step kernel matrix.

---

**Algorithm 2**  SR$^{\mathrm{R}}$-projection  *(semi-relaxed OT, right marginal relaxed)*

---

Input $\boldsymbol{K}, \gamma, \tau, \boldsymbol{a}, \boldsymbol{b}, \delta$
$\boldsymbol{u} \leftarrow \mathbf{1}_n$
$\boldsymbol{v} \leftarrow \mathbf{1}_r$
**repeat**
  $\tilde{\boldsymbol{u}} \leftarrow \boldsymbol{u}$
  $\tilde{\boldsymbol{v}} \leftarrow \boldsymbol{v}$
  $\boldsymbol{u} \leftarrow (\boldsymbol{a}/\boldsymbol{K}\boldsymbol{v})$
  $\boldsymbol{v} \leftarrow (\boldsymbol{b}/\boldsymbol{K}^{\mathrm{T}}\boldsymbol{u})^{\tau/(\tau+\gamma^{-1})}$
**until** $\gamma^{-1} \max\{\|\log \tilde{\boldsymbol{u}}/\boldsymbol{u}\|_\infty, \|\log \tilde{\boldsymbol{v}}/\boldsymbol{v}\|_\infty\} < \delta$
return $\mathrm{diag}(\boldsymbol{u}) \boldsymbol{K} \, \mathrm{diag}(\boldsymbol{v})$

---

**Algorithm 3**  U-projection  *(unbalanced OT)*

---

Input $\boldsymbol{K}, \gamma, \tau, \boldsymbol{a}, \boldsymbol{b}, \delta$
$\boldsymbol{u} \leftarrow \mathbf{1}_n$
$\boldsymbol{v} \leftarrow \mathbf{1}_r$
**repeat**
  $\tilde{\boldsymbol{u}} \leftarrow \boldsymbol{u}$
  $\tilde{\boldsymbol{v}} \leftarrow \boldsymbol{v}$
  $\boldsymbol{u} \leftarrow (\boldsymbol{a}/\boldsymbol{K}\boldsymbol{v})^{\tau/(\tau+\gamma^{-1})}$
  $\boldsymbol{v} \leftarrow (\boldsymbol{b}/\boldsymbol{K}^{\mathrm{T}}\boldsymbol{u})^{\tau/(\tau+\gamma^{-1})}$
**until** $\gamma^{-1} \max\{\|\log \tilde{\boldsymbol{u}}/\boldsymbol{u}\|_\infty, \|\log \tilde{\boldsymbol{v}}/\boldsymbol{v}\|_\infty\} < \delta$
return $\mathrm{diag}(\boldsymbol{u}) \boldsymbol{K} \, \mathrm{diag}(\boldsymbol{v})$

---

# C  The Latent Coupling Matrix

To solve the balanced form (and generalize the principle to the relaxed problems), we consider an alternative parametrization of the inner matrix. In particular, previous works Scetbon et al. (2021, 2023) consider $\mathrm{diag}(1/\boldsymbol{g})$ to be the inner matrix, with marginals $\boldsymbol{Q}^{\mathrm{T}} \mathbf{1}_n = \boldsymbol{R}^{\mathrm{T}} \mathbf{1}_m = \boldsymbol{g}$ to ensure that the outer conditions $\boldsymbol{Q} \mathbf{1}_r = \boldsymbol{a}$ and $\boldsymbol{R} \mathbf{1}_r = \boldsymbol{b}$ hold. We instead relax the constraint that $\boldsymbol{Q}^{\mathrm{T}} \mathbf{1}_n$ and $\boldsymbol{R}^{\mathrm{T}} \mathbf{1}_m$ are equal, by allowing $\boldsymbol{Q}^{\mathrm{T}} \mathbf{1}_n = \boldsymbol{g}_Q$ and $\boldsymbol{R}^{\mathrm{T}} \mathbf{1}_m = \boldsymbol{g}_R$ to vary arbitrarily, and considering a non-diagonal inner matrix $\boldsymbol{X} \in \mathbb{R}^{r \times r}$ in the place of $\mathrm{diag}(1/\boldsymbol{g})$ where we have the conditions:

$$
\boldsymbol{X} \boldsymbol{g}_R = \boldsymbol{X} \boldsymbol{R}^{\mathrm{T}} \mathbf{1}_m = \mathbf{1}_r
$$

And

$$
\boldsymbol{X}^{\mathrm{T}} \boldsymbol{g}_Q = \boldsymbol{X} \boldsymbol{Q}^{\mathrm{T}} \mathbf{1}_n = \mathbf{1}_r
$$

Thus, if one considers the coupling formed as $\boldsymbol{P}_r = \boldsymbol{QXR}^{\mathrm{T}}$, one maintains that $\boldsymbol{P}_r \in \Pi_{\boldsymbol{a},\boldsymbol{b}}$ from the condition of $\mathcal{C}_1$, defined in the balanced case as in (14). We clearly have that:

$$\boldsymbol{P}_r \boldsymbol{1}_m = \boldsymbol{QXR}^{\mathrm{T}} \boldsymbol{1}_m = \boldsymbol{Q}\boldsymbol{1}_r = \boldsymbol{a}$$

And:

$$\boldsymbol{P}_r^{\mathrm{T}} \boldsymbol{1}_n = \boldsymbol{RX}^{\mathrm{T}} \boldsymbol{Q}\boldsymbol{1}_n = \boldsymbol{R}\boldsymbol{1}_r = \boldsymbol{b}$$

We first consider two approaches to optimizing for such a $\boldsymbol{X}$, given that it is not a coupling. First, we consider the appropriate proximal step for $\boldsymbol{X}$

$$\min_{\boldsymbol{\zeta}} \langle \nabla \mathcal{L}(\boldsymbol{\zeta}) \mid_{\boldsymbol{\xi}_k}, \boldsymbol{\zeta} \rangle_F + \frac{1}{\gamma_k} \mathrm{KL}(\boldsymbol{\zeta} \| \boldsymbol{K}^{(k)})$$

We first note that the gradient of our loss with respect to $\boldsymbol{X}$ is given as $\boldsymbol{Q}^{\mathrm{T}} \boldsymbol{C} \boldsymbol{R}$. If one supposes that $\boldsymbol{X}$ is invertible with $\boldsymbol{X}^{-1} = \boldsymbol{T}$, we have that for any such $\boldsymbol{T}$:

$$\boldsymbol{X}^{-1} \boldsymbol{1}_r = \boldsymbol{T}\boldsymbol{1}_r = \boldsymbol{g}_R$$

And

$$\boldsymbol{X}^{-T} \boldsymbol{1}_r = \boldsymbol{T}^{\mathrm{T}} \boldsymbol{1}_r = \boldsymbol{g}_Q$$

This implies that this inverse matrix $\boldsymbol{T}$ is a coupling such that $\boldsymbol{T} \in \Pi_{\boldsymbol{g}_R,\boldsymbol{g}_Q}$, which also suggests one might be able to update it using Sinkhorn. In fact, being a density in $\mathbb{R}_+^{r \times r}$ it represents a transition matrix between the latent $r$-dimensional variables. In particular, writing the proximal step in full, we have:

$$\min_{\boldsymbol{T}} \langle \boldsymbol{Q}^{\mathrm{T}} \boldsymbol{C} \boldsymbol{R}, \boldsymbol{T}^{-1} \rangle_F + \frac{1}{\gamma_k} \mathrm{KL}(\boldsymbol{T}_k \| \boldsymbol{K}_{\boldsymbol{T}}^{(k)})$$

Noting the derivative $D(\boldsymbol{X}^{-1}) \circ \boldsymbol{V} = -\boldsymbol{X}^{-1} \boldsymbol{V} \boldsymbol{X}^{-1}$, we have from the first-order condition that:

$$-\boldsymbol{T}^{-T} \boldsymbol{Q}^{\mathrm{T}} \boldsymbol{C} \boldsymbol{R} \boldsymbol{T}^{-T} + \frac{1}{\gamma_k} \log \left[ \frac{\boldsymbol{T}_k}{\boldsymbol{K}_{\boldsymbol{T}}^{(k)}} \right] = \boldsymbol{0}$$

This implies the kernel matrix update:

$$\boldsymbol{K}_{\boldsymbol{T}}^{(k)} = \boldsymbol{T}_k \odot \exp\{+\gamma_k \boldsymbol{T}^{-T} \boldsymbol{Q}^{\mathrm{T}} \boldsymbol{C} \boldsymbol{R} \boldsymbol{T}^{-T}\}$$

Where one then takes the Sinkhorn projection J onto the set $\Pi_{\boldsymbol{g}_R,\boldsymbol{g}_Q}$ as $\mathcal{P}_{\Pi_{\boldsymbol{g}_R,\boldsymbol{g}_Q}}(\boldsymbol{K}_{\boldsymbol{T}}^{(k)})$ using the Sinkhorn algorithm Cuturi (2013b). However, a more stable and inversion-free update exists which ensures $\boldsymbol{X}$ remains positive by a diagonal re-scaling in the form introduced by Lin et al. (2021). In particular, if one takes $\boldsymbol{X} = \mathrm{diag}(1/\boldsymbol{g}_Q)\boldsymbol{T} \,\mathrm{diag}(1/\boldsymbol{g}_R)$, then

$$\boldsymbol{X}\boldsymbol{g}_R = \mathrm{diag}(1/\boldsymbol{g}_Q)\boldsymbol{T} \,\mathrm{diag}(1/\boldsymbol{g}_R)\boldsymbol{g}_R = \mathrm{diag}(1/\boldsymbol{g}_Q)\boldsymbol{T}\boldsymbol{1}_r = \boldsymbol{1}_r$$

and likewise

$$\boldsymbol{X}^{\mathrm{T}} \boldsymbol{g}_Q = \mathrm{diag}(1/\boldsymbol{g}_R)\boldsymbol{T}^{\mathrm{T}} \,\mathrm{diag}(1/\boldsymbol{g}_Q)\boldsymbol{g}_Q = \mathrm{diag}(1/\boldsymbol{g}_R)\boldsymbol{T}^{\mathrm{T}}\boldsymbol{1}_r = \boldsymbol{1}_r$$

so that $\boldsymbol{X}$ necessarily satisfies $\boldsymbol{X}\boldsymbol{g}_R = \boldsymbol{1}_r$ and $\boldsymbol{X}^{\mathrm{T}} \boldsymbol{g}_Q = \boldsymbol{1}_r$. Thus $\boldsymbol{T}\boldsymbol{1}_r = \boldsymbol{g}_Q$ and $\boldsymbol{T}^{\mathrm{T}} \boldsymbol{1}_r = \boldsymbol{g}_R$ and $\boldsymbol{T} \in \Pi_{\boldsymbol{g}_Q,\boldsymbol{g}_R}$. With analogous reasoning to before, one has a step for the coupling $\boldsymbol{T}$ in the form:

$$\min_{\boldsymbol{T}} \langle \boldsymbol{Q}^{\mathrm{T}} \boldsymbol{C} \boldsymbol{R}, \mathrm{diag}(1/\boldsymbol{g}_Q)\boldsymbol{T} \,\mathrm{diag}(1/\boldsymbol{g}_R) \rangle_F + \frac{1}{\gamma_k} \mathrm{KL}(\boldsymbol{T}_k \| \boldsymbol{K}_{\boldsymbol{T}}^{(k)})$$

Which yields the kernel matrix:

$$\boldsymbol{K}_{\boldsymbol{T}}^{(k)} = \boldsymbol{T}_k \odot \exp\{-\gamma_k \,\mathrm{diag}(\boldsymbol{g}_Q)^{-1} \boldsymbol{Q}^{\mathrm{T}} \boldsymbol{C} \boldsymbol{R} \,\mathrm{diag}(\boldsymbol{g}_R)^{-1}\}$$

Which is likewise projected onto $\Pi_{\boldsymbol{g}_Q,\boldsymbol{g}_R}$ using the Sinkhorn algorithm. From this $\boldsymbol{T} \in \Pi_{\boldsymbol{g}_Q,\boldsymbol{g}_R}$, one takes $\boldsymbol{X} = \mathrm{diag}(1/\boldsymbol{g}_Q)\boldsymbol{T} \,\mathrm{diag}(1/\boldsymbol{g}_R)$ as the inner matrix that corresponds the unequal marginals $\boldsymbol{g}_Q$ and $\boldsymbol{g}_R$ which ensuring $\boldsymbol{P}_r \in \Pi_{\boldsymbol{a},\boldsymbol{b}}$.

**Algorithm 4** FRLC    *(General marginal constraint low-rank optimal transport)*

Input $\boldsymbol{C}, r, r_2, \boldsymbol{a}, \boldsymbol{b}, \tau, \tau_2, \gamma, \delta, \varepsilon$
Initialize $\boldsymbol{g}_Q, \boldsymbol{g}_R = \frac{1}{r}\mathbf{1}_r, \frac{1}{r_2}\mathbf{1}_{r_2}$
$\boldsymbol{Q}_0, \boldsymbol{R}_0, \boldsymbol{T}_0 \leftarrow$ *Initialize-Couplings*$(\boldsymbol{a}, \boldsymbol{b}, \boldsymbol{g}_Q, \boldsymbol{g}_R)$
**if** $r = r_2$ **then**
    $\boldsymbol{X}_0 \leftarrow \boldsymbol{T}_0^{-1}$    # Invertible case
**else**
    $\boldsymbol{X}_0 \leftarrow \mathrm{diag}(1/\boldsymbol{Q}_0^{\mathrm{T}}\mathbf{1}_n)\boldsymbol{T}_0\,\mathrm{diag}(1/\boldsymbol{R}_0^{\mathrm{T}}\mathbf{1}_m)$    # General case
**end if**
**while** $\Delta((\boldsymbol{Q}_k, \boldsymbol{R}_k, \boldsymbol{T}_k), (\boldsymbol{Q}_{k-1}, \boldsymbol{R}_{k-1}, \boldsymbol{T}_{k-1})) > \varepsilon$ **do**
    $\nabla_{\boldsymbol{Q}} \leftarrow \boldsymbol{C}\boldsymbol{R}_k\boldsymbol{X}_k^{\mathrm{T}} - \mathbf{1}_n\mathrm{diag}^{-1}((\boldsymbol{C}\boldsymbol{R}_k\boldsymbol{X}_k^{\mathrm{T}})^{\mathrm{T}}\boldsymbol{Q}_k\mathrm{diag}(1/\boldsymbol{g}_Q))^{\mathrm{T}}$
    $\nabla_{\boldsymbol{R}} \leftarrow \boldsymbol{C}^{\mathrm{T}}\boldsymbol{Q}_k\boldsymbol{X}_k - \mathbf{1}_m\mathrm{diag}^{-1}(\mathrm{diag}(1/\boldsymbol{g}_R)\boldsymbol{R}_k^{\mathrm{T}}\boldsymbol{C}^{\mathrm{T}}\boldsymbol{Q}_k\boldsymbol{X}_k)^{\mathrm{T}}$
    $\gamma_k \leftarrow \gamma/\max\{\|\nabla_{\boldsymbol{Q}}\|_\infty, \|\nabla_{\boldsymbol{R}}\|_\infty\}$    # $\ell^\infty$-normalization of Scetbon & Cuturi (2022)
    $\boldsymbol{K}_{\boldsymbol{Q}}^{(k)}, \boldsymbol{K}_{\boldsymbol{R}}^{(k)} \leftarrow \boldsymbol{Q}_k \odot \exp(-\gamma_k\nabla_{\boldsymbol{Q}}), \boldsymbol{R}_k \odot \exp(-\gamma_k\nabla_{\boldsymbol{R}})$
    **if** Balanced **then**
        $\boldsymbol{Q}_k \leftarrow \mathrm{SR}^{\mathrm{R}}$-*projection*$(\boldsymbol{K}_{\boldsymbol{Q}}^{(k)}, \gamma_k, \tau, \boldsymbol{a}, \boldsymbol{Q}_{k-1}^{\mathrm{T}}\mathbf{1}_n, \delta)$    # Semi-relaxed OT
        $\boldsymbol{R}_k \leftarrow \mathrm{SR}^{\mathrm{R}}$-*projection*$(\boldsymbol{K}_{\boldsymbol{R}}^{(k)}, \gamma_k, \tau, \boldsymbol{b}, \boldsymbol{R}_{k-1}^{\mathrm{T}}\mathbf{1}_m, \delta)$    # Semi-relaxed OT
    **else if** Unbalanced **then**
        $\boldsymbol{Q}_k \leftarrow \mathrm{U}$-*projection*$(\boldsymbol{K}_{\boldsymbol{Q}}^{(k)}, \gamma_k, \tau, \boldsymbol{a}, \boldsymbol{Q}_{k-1}^{\mathrm{T}}\mathbf{1}_n, \delta)$    # Unbalanced OT
        $\boldsymbol{R}_k \leftarrow \mathrm{U}$-*projection*$(\boldsymbol{K}_{\boldsymbol{R}}^{(k)}, \gamma_k, \tau_2, \boldsymbol{b}, \boldsymbol{R}_{k-1}^{\mathrm{T}}\mathbf{1}_m, \delta)$    # Unbalanced OT
    **else if** Semi-Relaxed Left **then**
        $\boldsymbol{Q}_k \leftarrow \mathrm{U}$-*projection*$(\boldsymbol{K}_{\boldsymbol{Q}}^{(k)}, \gamma_k, \tau, \boldsymbol{a}, \boldsymbol{Q}_{k-1}^{\mathrm{T}}\mathbf{1}_n, \delta)$    # Unbalanced OT
        $\boldsymbol{R}_k \leftarrow \mathrm{SR}^{\mathrm{R}}$-*projection*$(\boldsymbol{K}_{\boldsymbol{R}}^{(k)}, \gamma_k, \tau, \boldsymbol{b}, \boldsymbol{R}_{k-1}^{\mathrm{T}}\mathbf{1}_m, \delta)$    # Semi-relaxed OT
    **else if** Semi-Relaxed Right **then**
        $\boldsymbol{Q}_k \leftarrow \mathrm{SR}^{\mathrm{R}}$-*projection*$(\boldsymbol{K}_{\boldsymbol{Q}}^{(k)}, \gamma_k, \tau, \boldsymbol{a}, \boldsymbol{Q}_{k-1}^{\mathrm{T}}\mathbf{1}_n, \delta)$    # Semi-relaxed OT
        $\boldsymbol{R}_k \leftarrow \mathrm{U}$-*projection*$(\boldsymbol{K}_{\boldsymbol{R}}^{(k)}, \gamma_k, \tau, \boldsymbol{b}, \boldsymbol{R}_{k-1}^{\mathrm{T}}\mathbf{1}_m, \delta)$    # Unbalanced OT
    **end if**
    $\boldsymbol{g}_Q, \boldsymbol{g}_R \leftarrow \boldsymbol{Q}_k^{\mathrm{T}}\mathbf{1}_n, \boldsymbol{R}_k^{\mathrm{T}}\mathbf{1}_m$
    $\nabla_{\boldsymbol{T}} = \mathrm{diag}(\boldsymbol{g}_Q)^{-1}\boldsymbol{Q}_k^{\mathrm{T}}\boldsymbol{C}\boldsymbol{R}_k\,\mathrm{diag}(\boldsymbol{g}_R)^{-1}$
    $\gamma_{\boldsymbol{T}} = \gamma/\|\nabla_{\boldsymbol{T}}\|_\infty$    # $\ell^\infty$-normalization
    $\boldsymbol{K}_{\boldsymbol{T}}^{(k)} \leftarrow \boldsymbol{T}_k \odot \exp\{-\gamma_{\boldsymbol{T}}\nabla_{\boldsymbol{T}}\}$
    $\boldsymbol{T}_k \leftarrow$ *Sinkhorn*$(\boldsymbol{K}_{\boldsymbol{T}}^{(k)}, \boldsymbol{g}_R, \boldsymbol{g}_Q, \delta)$    # Balanced OT
    $\boldsymbol{X}_k \leftarrow \mathrm{diag}(1/\boldsymbol{g}_Q)\boldsymbol{T}\,\mathrm{diag}(1/\boldsymbol{g}_R)$
**end while**
Return $\boldsymbol{P}_r = \boldsymbol{Q}\boldsymbol{X}\boldsymbol{R}^{\mathrm{T}}$

---

**Algorithm 5** Sinkhorn Algorithm    *(Cuturi (2013b), balanced OT)*

Input $\boldsymbol{K}, \boldsymbol{a}, \boldsymbol{b}, \delta$
$\boldsymbol{u} \leftarrow \mathbf{1}_n$
$\boldsymbol{v} \leftarrow \mathbf{1}_m$
**while** $\|\mathrm{diag}(\boldsymbol{u})\boldsymbol{K}\boldsymbol{v} - \boldsymbol{a}\|_1 + \|\mathrm{diag}(\boldsymbol{v})\boldsymbol{K}^{\mathrm{T}}\boldsymbol{u} - \boldsymbol{b}\|_1 > \delta$ **do**
    $\boldsymbol{u}^{(l+1)} \leftarrow \boldsymbol{a}/\boldsymbol{K}\boldsymbol{v}^{(l)}$
    $\boldsymbol{v}^{(l+1)} \leftarrow \boldsymbol{b}/\boldsymbol{K}^{\mathrm{T}}\boldsymbol{u}^{(l+1)}$
**end while**
Return $\mathrm{diag}(\boldsymbol{u})\boldsymbol{K}\,\mathrm{diag}(\boldsymbol{v})$

---

# D    Gromov-Wasserstein (GW)

As defined in, the general Gromov-Wasserstein problem concerns a minimization of the energy:

$$\mathcal{Q}_{\boldsymbol{A},\boldsymbol{B}}(\boldsymbol{P}) = \sum_{i,j,k,l} (\boldsymbol{A}_{ik} - \boldsymbol{B}_{jl})^2 \boldsymbol{P}_{ij}\boldsymbol{P}_{kl} \qquad (30)$$

Where the minimization is over the set of all couplings $\mathbf{\Pi}_{a,b}$:

$$\mathrm{GW}(\mu, \nu) := \min_{P \in \Pi_{a,b}} \mathcal{Q}_{A,B}(P) \tag{31}$$

We consider extending the semi-relaxed framework to the GW problem under the low-rank restriction on $\boldsymbol{P}$. This extension has thus far been considered for the balanced and unbalanced case in two previous works Scetbon et al. (2023, 2022). In both of these works, the algorithms for the Wasserstein case extend trivially to the Gromov-Wasserstein (GW) problem. In particular, each kernel with variable $\boldsymbol{\zeta}$ has an update of the form $\boldsymbol{K} \leftarrow \boldsymbol{\zeta} \odot \exp\left(-\gamma_k \nabla_{\boldsymbol{\zeta}} \mathcal{L}(\boldsymbol{\zeta})\right)$ for $\mathcal{L}(\boldsymbol{\zeta})$ heretofore taken to be the Wasserstein loss $\langle \boldsymbol{P}(\boldsymbol{\zeta}), \boldsymbol{C} \rangle_F$ where the coupling $\boldsymbol{P} = \boldsymbol{QXR}^{\mathrm{T}}$ is interpreted as a function of the low-rank sub-factor variables $\boldsymbol{\zeta} \in \{\boldsymbol{Q}, \boldsymbol{R}, \boldsymbol{X}\}$. Taking $\mathcal{L} := \mathcal{Q}_{A,B}(\boldsymbol{P}(\boldsymbol{\zeta}))$ one can simply extend the gradient through the GW-loss and directly use it in place of the Wasserstein gradient in the update $\boldsymbol{K} \leftarrow \boldsymbol{\zeta} \odot \exp\left(-\gamma_k \nabla_{\boldsymbol{\zeta}} \mathcal{L}(\boldsymbol{\zeta})\right)$ for $\mathcal{L}(\boldsymbol{\zeta})$ of each algorithm. The matrix-form of the GW-cost is expressed as:

$$\mathcal{Q}_{A,B}(\boldsymbol{P}) = \mathbf{1}_m^{\mathrm{T}} \boldsymbol{P}^{\mathrm{T}} \boldsymbol{A}^{\odot 2} \boldsymbol{P} \mathbf{1}_m + \mathbf{1}_n^{\mathrm{T}} \boldsymbol{P} \boldsymbol{B}^{\odot 2} \boldsymbol{P}^{\mathrm{T}} \mathbf{1}_n - 2\langle \boldsymbol{APB}, \boldsymbol{P} \rangle$$

Which, using the constraints of $\mathcal{C}_2$ reduces the cost as a function of $\boldsymbol{Q}, \boldsymbol{R}, \boldsymbol{X}$ to:

$$\mathcal{Q}_{A,B}(\boldsymbol{Q}, \boldsymbol{R}, \boldsymbol{X}) = \mathbf{1}_r^{\mathrm{T}} \boldsymbol{Q}^{\mathrm{T}} \boldsymbol{A}^{\odot 2} \boldsymbol{Q} \mathbf{1}_r + \mathbf{1}_r^{\mathrm{T}} \boldsymbol{R}^{\mathrm{T}} \boldsymbol{B}^{\odot 2} \boldsymbol{R} \mathbf{1}_r - 2\langle \boldsymbol{QXR}^{\mathrm{T}}, \boldsymbol{AQXR}^{\mathrm{T}} \boldsymbol{B} \rangle_F$$

$$\nabla_{\boldsymbol{Q}} \mathcal{Q}_{A,B}(\boldsymbol{Q}, \boldsymbol{R}, \boldsymbol{X}) = 2\boldsymbol{A}^{\odot 2} \boldsymbol{Q} \mathbf{1}_r \mathbf{1}_r^{\mathrm{T}} - 4\boldsymbol{AQXR}^{\mathrm{T}} \boldsymbol{BRX}^{\mathrm{T}}$$

Which is proportional to $\nabla_{\boldsymbol{Q}} \mathcal{Q}_{A,B}(\boldsymbol{Q}, \boldsymbol{R}, \boldsymbol{X}) \propto -4\boldsymbol{AQXR}^{\mathrm{T}} \boldsymbol{BRX}^{\mathrm{T}}$ for the balanced and right-marginal semi-relaxed case. And:

$$\nabla_{\boldsymbol{R}} \mathcal{Q}_{A,B}(\boldsymbol{Q}, \boldsymbol{R}, \boldsymbol{X}) = 2\boldsymbol{B}^{\odot 2} \boldsymbol{R} \mathbf{1}_r \mathbf{1}_r^{\mathrm{T}} - 4\boldsymbol{BRX}^{\mathrm{T}} \boldsymbol{Q}^{\mathrm{T}} \boldsymbol{AQX}$$

Which likewise can be reduced in proportionality to $\nabla_{\boldsymbol{R}} \mathcal{Q}_{A,B}(\boldsymbol{Q}, \boldsymbol{R}, \boldsymbol{g}) \propto -4\boldsymbol{BRX}^{\mathrm{T}} \boldsymbol{Q}^{\mathrm{T}} \boldsymbol{AQX}$ in the balanced and left-marginal semi-relaxed case. The gradients, as presented above, assume a linearization in $\boldsymbol{X} \leftarrow \boldsymbol{X}_k$. If one does not make this assumption and takes $\boldsymbol{X}(\boldsymbol{Q}, \boldsymbol{R}) = \mathrm{diag}(1/\boldsymbol{Q}^T \mathbf{1}_n) \boldsymbol{T} \mathrm{diag}(1/\boldsymbol{R}^T \mathbf{1}_m)$, a rank-one perturbation must be added to the $\boldsymbol{Q}$ and $\boldsymbol{R}$ gradient of the form:

$$\nabla_{\boldsymbol{Q}}^{(2)} = 4\mathbf{1}_n \mathrm{diag}^{-1}(\boldsymbol{XR}^{\mathrm{T}} \boldsymbol{B}(\boldsymbol{QXR}^{\mathrm{T}})^{\mathrm{T}} \boldsymbol{AQ} \mathrm{diag}(1/\boldsymbol{g_Q}))^{\mathrm{T}}$$

$$\nabla_{\boldsymbol{R}}^{(2)} = 4\mathbf{1}_m \mathrm{diag}^{-1}(\boldsymbol{X}^T \boldsymbol{Q}^T \boldsymbol{AQXR}^{\mathrm{T}} \boldsymbol{BR} \mathrm{diag}(1/\boldsymbol{g_R}))^{\mathrm{T}}$$

Analogously, for the gradient on $\boldsymbol{T}$ one can simply take the gradient with respect to $\boldsymbol{X}$, and subsequently $\boldsymbol{T}$ as $\boldsymbol{X}(\boldsymbol{T}) = \mathrm{diag}(\boldsymbol{g_Q})^{-1} \boldsymbol{T} \mathrm{diag}(\boldsymbol{g_R})^{-1}$. The gradient with respect to $\boldsymbol{X}$ is given as:

$$\nabla_{\boldsymbol{X}} \mathcal{Q}_{A,B}(\boldsymbol{Q}, \boldsymbol{R}, \boldsymbol{X}) = -4\boldsymbol{Q}^{\mathrm{T}} \boldsymbol{AQXR}^{\mathrm{T}} \boldsymbol{BR}$$

And thus, the directional derivative with respect to $\boldsymbol{X}$ in the direction $\boldsymbol{V_X} = \mathrm{diag}(\boldsymbol{g_Q})^{-1} \boldsymbol{V_T} \mathrm{diag}(\boldsymbol{g_R})^{-1}$ and thus by the chain rule $\boldsymbol{V_T}$ is:

$$D\mathcal{Q}_{A,B}(\boldsymbol{Q}, \boldsymbol{R}, \boldsymbol{X}) \circ (\boldsymbol{V_X}) = -4\langle \boldsymbol{Q}^{\mathrm{T}} \boldsymbol{AQXR}^{\mathrm{T}} \boldsymbol{BR}, \boldsymbol{V_X} \rangle_F$$
$$= -4\langle \boldsymbol{Q}^{\mathrm{T}} \boldsymbol{AQXR}^{\mathrm{T}} \boldsymbol{BR}, \mathrm{diag}(1/\boldsymbol{g_Q}) \boldsymbol{V_T} \mathrm{diag}(1/\boldsymbol{g_R}) \rangle_F$$

So that the gradient with respect to the coupling matrix $\boldsymbol{T}$ is given as:

$$\nabla_{\boldsymbol{T}} \mathcal{Q}_{A,B}(\boldsymbol{Q}, \boldsymbol{R}, \boldsymbol{T}) = -4 \mathrm{diag}(1/\boldsymbol{g_Q}) \boldsymbol{Q}^{\mathrm{T}} \boldsymbol{AQXR}^{\mathrm{T}} \boldsymbol{BR} \mathrm{diag}(1/\boldsymbol{g_R})$$

## E  Convergence Analysis and Other Proofs

### E.1  Convergence and Smoothness of the Objective

We show in Proposition 3.3 that directly applying the block-descent lemma of Beck & Tetruashvili (2013) to the template of Ghadimi et al.'s proof Ghadimi et al. (2014) is sufficient to show the non-asymptotic stationary convergence of a coordinate mirror descent procedure in Ghadimi's criterion. The non-asymptotic guarantee of Ghadimi et al. (2014) follows directly in the case of coordinate mirror descent using Lemma E.1 and Lemma E.2 below. For completeness, we define all notation used, describe the coordinate mirror descent algorithm in general, and discuss a few relevant preliminaries.

Suppose that the vector of $n$ variables $\mathbf{x} \in \mathbb{R}^n$ is partitioned into $p$ blocks, $\mathbf{x} = (\mathbf{x}(1), \ldots, \mathbf{x}(p))$, where $\mathbf{x}(i) \in \mathbb{R}^{n_i}$. Here, $n_1, \ldots, n_p$ are positive integers summing to $n$. Following the notation of Beck & Tetruashvili (2013); Nesterov (2012), we define matrices $\mathbf{U}_i \in \mathbb{R}^{n \times n_i}$ such that $\mathbf{x}(i) = \mathbf{U}_i^{\mathrm{T}} \mathbf{x}$ for all $i = 1, \ldots, p$. This also implies $\mathbf{x} = \sum_{i=1}^{p} \mathbf{U}_i \mathbf{x}(i)$. This allows us to define the vector of partial derivatives corresponding to each block of variables $\mathbf{x}(i)$:

$$\nabla_i f(\mathbf{x}) := \mathbf{U}_i^{\mathrm{T}} \nabla f(\mathbf{x}).$$

In Beck & Tetruashvili (2013), the gradient of $f$ is assumed to be block-coordinate-wise Lipschitz, with $L_i$ the smoothness constant associated to the $i$-th block of variables: for all $\mathbf{h}_i \in \mathbb{R}^{n_i}$, one has

$$\|\nabla_i f(\mathbf{x} + \mathbf{U}_i \mathbf{h}_i) - \nabla_i f(\mathbf{x})\| \leq L_i \|\mathbf{h}_i\|, \tag{32}$$

and for such functions we denote by $L := \max_i L_i$ the (global) smoothness constant of $\nabla f$. To be clear, a *smoothness constant* associated to $f$ is a Lipschitz constant of its gradient.

**Lemma E.1** (Block descent lemma, Beck & Tetruashvili (2013), Lemma 3.2.). *Suppose $f \in C^1(\mathbb{R}^n, \mathbb{R})$ is a continuously differentiable function over $\mathbb{R}^n$ whose gradient is block-coordinatewise Lipschitz* (32) *for $L_i$ the smoothness constant associated to the $i$-th block of variables $\mathbf{x}(i)$. Let $\mathbf{u}, \mathbf{v}$ be two vectors differing only in the $i$-th block: there exists $\mathbf{h}_i \in \mathbb{R}^{n_i}$ such that $\mathbf{v} - \mathbf{u} = \mathbf{U}_i \mathbf{h}_i$. Then,*

$$f(\mathbf{v}) \leq f(\mathbf{u}) + \langle \nabla f(\mathbf{u}), \mathbf{v} - \mathbf{u} \rangle + \frac{L_i}{2} \|\mathbf{u} - \mathbf{v}\|^2. \tag{33}$$

Lemma E.1 is central to adapting the proof of Theorem 1 of Ghadimi et al. (2014) to our case. Their Theorem 1 concerns the non-asymptotic convergence of mirror descent for objectives of the form

$$\min_{\mathbf{x} \in \mathcal{X}} f(\mathbf{x}) + h(\mathbf{x}),$$

where $\mathcal{X}$ is a closed, convex subset of $\mathbb{R}^n$, $f \in C^1(\mathcal{X}, \mathbb{R})$ is a possibly non-convex objective, and where $h$ is an $\alpha$-strongly convex function. Using notation similar to Ghadimi et al. (2014), we write $\Phi(\mathbf{x}) = f(\mathbf{x}) + h(\mathbf{x})$. Additionally, Ghadimi et al. (2014) assume $\nabla f$ is $L$-Lipschitz for some $L > 0$.

To unify our assumptions on $f$, we suppose that $\nabla f \in C^1(\mathcal{X}, \mathbb{R})$ is block-coordinate Lipschitz (32) with block Lipschitz constants $(L_i)_{i=1}^{p}$, and that $\mathcal{X}$ itself decomposes as a product $\mathcal{X} = \prod_{i=1}^{p} \mathcal{X}_i$, where each $\mathcal{X}_i$ is a closed convex set constraining the block variables $\mathbf{x}(i)$.

The proof of Ghadimi relies on $\beta$-smoothness of the objective in all variables. We show that component-wise smoothness in each block is sufficient to achieve an analogous convergence result for a coordinate mirror descent. To provide context for Proposition 3.3, we now describe in general (1) mirror descent, and (2) block-coordinate mirror descent.

We again follow the notation of Ghadimi et al. (2014). A function $\omega : \mathcal{X} \rightarrow \mathbb{R}$ is a *distance generating function* with modulus $\alpha > 0$, with respect to the Euclidean norm $\|\cdot\|$, if $\omega$ is continuously differentiable and strongly convex, so that

$$\langle \mathbf{x} - \mathbf{z}, \nabla \omega(\mathbf{x}) - \nabla \omega(\mathbf{z}) \rangle \geq \alpha \|\mathbf{x} - \mathbf{z}\|^2, \quad \text{for all } \mathbf{x}, \mathbf{z} \in \mathcal{X}.$$

The *prox-function* (or Bregman divergence) associated with $\omega$ is then

$$V(\mathbf{x}, \mathbf{z}) = \omega(\mathbf{x}) - \omega(\mathbf{z}) - \langle \nabla \omega(\mathbf{z}), \mathbf{x} - \mathbf{z} \rangle,$$

and from this prox-function, $\gamma > 0$, and some $\mathbf{g} \in \mathbb{R}^n$, we define the *generalized projection*

$$\mathbf{x}^+ := \underset{\mathbf{u} \in \mathcal{X}}{\arg\min} \left( \langle \mathbf{g}, \mathbf{u} \rangle + \frac{1}{\gamma} V(\mathbf{u}, \mathbf{x}) + h(\mathbf{u}) \right). \tag{34}$$

To describe the projected gradient descent algorithm (which coincides with mirror descent in our case of interest), we first define the *generalized projected gradient of $\Phi$ at $\mathbf{x}$*:

$$P_{\mathcal{X}}(\mathbf{x}, \mathbf{g}, \gamma) := \frac{1}{\gamma}(\mathbf{x} - \mathbf{x}^+).$$

The *mirror descent (MD) algorithm* is as follows: given initial point $\mathbf{x}_0 \in \mathcal{X}$, a total number of iterations $N$, and positive stepsizes $(\gamma_k)_{k=1}^{N}$, at step $k$, the $(k+1)$-st iterate is computed via

$$\mathbf{x}_{k+1} \leftarrow \underset{\mathbf{u} \in \mathcal{X}}{\arg\min} \left( \langle \nabla f(\mathbf{x}_k), \mathbf{u} \rangle + \frac{1}{\gamma_k} V(\mathbf{u}, \mathbf{x}_k) + h(\mathbf{u}) \right).$$

Among all iterates $\mathbf{x}_k$, the MD algorithm outputs the one at which the generalized projected gradient is of least norm. Concretely, $\mathbf{x}_R$ is the output of the MD algorithm, where

$$R := \arg\min_{k=0,\ldots,N} \|\mathbf{g}_{\mathcal{X},k}\|^2,$$

and where $\mathbf{g}_{\mathcal{X},k}$ is

$$\mathbf{g}_{\mathcal{X},k} := P_{\mathcal{X}}(\mathbf{x}_k, \nabla f(\mathbf{x}_k), \gamma_k).$$

Having described the MD algorithm, let us consider a block-coordinate variant; we assume $\mathbf{x}$ admits the block-coordinate structure described above. To simplify the presentation, we suppose that in a given iteration $k$, the block variables are updated sequentially from $i = 1, \ldots, p$. This leads to doubly-indexed iterates $(\mathbf{x}_k^i)$ with $k = 1, \ldots, N$ indexing each full iteration through all variables, and $i = 0, 1, \ldots, p$ indexing the sub-iterations which update one block of variables at a time.

The *coordinate mirror descent (CMD) algorithm* takes as input an initial point $\mathbf{x}_0 \in \mathcal{X}$, a number of iterations $N$, and a sequence of positive stepsizes $(\gamma_{k,i})_{k=1,i=1}^{N,p}$. We set $\mathbf{x}_0^0 = \mathbf{x}_0$, and for $k = 0, \ldots, N - 1$ and $i = 1, \ldots, p$, we compute $\mathbf{x}_k^i$ from $\mathbf{x}_k^{i-1}$ as follows:

$$\mathbf{x}_k^i(i) \leftarrow \arg\min_{\mathbf{u}_i \in \mathcal{X}_i} \left( \langle \nabla_i f(\mathbf{x}_k^{i-1}), \mathbf{u}_i \rangle + \frac{1}{\gamma_{k,i}} V_i(\mathbf{u}_i, \mathbf{U}_i^{\mathrm{T}} \mathbf{x}_k^{i-1}) + h_i(\mathbf{u}_i) \right)$$
$$\mathbf{x}_k^i(j) \leftarrow \mathbf{x}_k^{i-1}(j) \quad \text{for } j \neq i$$

Here, we have assumed that $V$ can be written as a composite function of the block variables,

$$V(\mathbf{x}, \mathbf{z}) = \sum_{i=1}^{p} V_i(\mathbf{x}(i), \mathbf{z}(i)),$$

as is the case for the KL divergence. We also have assumed that $h$ has this composite structure (as with entropy):

$$h(\mathbf{x}) = \sum_{i=1}^{p} h_i(\mathbf{x}(i)).$$

Lastly, for $k = 0, \ldots, N - 1$, we set $\mathbf{x}_{k+1}^0 := \mathbf{x}_k^p$. We define $\mathbf{g}_{\mathcal{X},k} := (\mathbf{g}_{\mathcal{X},k,1}, \ldots, \mathbf{g}_{\mathcal{X},k,p})$ to be the collection of block-wise differences, where by definition

$$\mathbf{g}_{\mathcal{X},k,i} = P_{\mathcal{X}_i}(\mathbf{x}_k^{i-1}, \nabla_i f(\mathbf{x}_k^{i-1}), \gamma_{k,i}) = \frac{1}{\gamma_{k,i}}\left( \mathbf{x}_k^{i-1} - \mathbf{x}_k^i \right) = \mathbf{U}_i^{\mathrm{T}} \mathbf{g}_{\mathcal{X},k}.$$

The convergence criterion $\Delta$ we use in Proposition 3.3 is the one used by Ghadimi et al. (2014) summed across blocks. In particular, the CMD algorithm returns iterate $\mathbf{x}_R$, where

$$R := \arg\min_{k=0,\ldots,N} \Delta(\mathbf{x}_k, \mathbf{x}_{k-1}),$$

$$\Delta(\mathbf{x}_k, \mathbf{x}_{k-1}) := \|\mathbf{g}_{\mathcal{X},k}\|^2 = \sum_{i=1}^{p} \|\mathbf{g}_{\mathcal{X},k,i}\|^2. \tag{35}$$

For $\Phi = f + h$ as above ($f$ has global smoothness constant $L$), let $\Phi^* = \arg\min_{\mathbf{x} \in \mathcal{X}} \Phi(\mathbf{x})$, and define

$$D := \left( \frac{\Phi(x_0) - \Phi^*}{L} \right)^{1/2}. \tag{36}$$

The other lemma used in the proof of Proposition 3.3 is as follows.

**Lemma E.2** (Ghadimi et al. (2014), Lemma 1). *Let* $\mathbf{x}^+ = \arg\min_{\mathbf{u} \in \mathcal{X}} \{ \langle \mathbf{g}, \mathbf{u} \rangle + \frac{1}{\gamma} V(\mathbf{u}, \mathbf{x}) + h(\mathbf{u}) \}$ *and* $P_{\mathcal{X}}(\mathbf{x}, \mathbf{g}, \gamma) = \frac{1}{\gamma}(\mathbf{x} - \mathbf{x}^+)$. *Then for all* $\mathbf{x} \in \mathbb{R}^n$, *all* $\mathbf{g} \in \mathbb{R}^n$, *and* $\gamma > 0$, *one has:*

$$\langle \mathbf{g}, P_{\mathcal{X}}(\mathbf{x}, \mathbf{g}, \gamma) \rangle \geq \alpha \|P_{\mathcal{X}}(\mathbf{x}, \mathbf{g}, \gamma)\|^2 + \frac{1}{\gamma}(h(\mathbf{x}^+) - h(\mathbf{x})).$$

**Proposition E.3** (Proposition 3.3). *Suppose one has $f \in C^1(\mathcal{X}, \mathbb{R})$ whose gradient is block-coordinate Lipschitz, with block smoothness constants $(L_i)_{i=1}^p$, and a function $h \in C(\mathcal{X}, \mathbb{R})$ which is $\alpha$-strongly convex. For $\Phi = f + h$, suppose one performs a coordinate mirror descent on $\Phi$ minimized over a product of closed convex sets $\mathcal{X} = \prod_{i=1}^p \mathcal{X}_i$. Let the sub-iterates with respect to the $i$-th block update be $\{\mathbf{x}_k^i\}_{i=0}^p$ where $\mathbf{x}_k := \mathbf{x}_k^0$ for $k \in [N]$ outer iterations. Then one has:*

$$\min_k \Delta(\mathbf{x}_k, \mathbf{x}_{k-1}) \leq \frac{D^2 L}{N(\alpha^2/2L)} = \frac{2D^2 L^2}{N\alpha^2},$$

*where $D$ is (36), $L$ is the global smoothness constant of $f$, and convergence criterion $\Delta(\mathbf{x}_k, \mathbf{x}_{k-1})$ is given in (35). Above, the stepsizes $\gamma_{k,i}$ in the coordinate mirror descent are $\gamma_{k,i} := \alpha/L$.*

*Proof.* As $f$ satisfies the hypotheses of the block descent lemma, Lemma E.1, we apply (33) to obtain:

$$f(\mathbf{x}_k^i) \leq f(\mathbf{x}_k^{i-1}) + \langle \nabla_i f(\mathbf{x}_k^{i-1}), \mathbf{x}_k^i - \mathbf{x}_k^{i-1} \rangle + \frac{L_i}{2} \|\mathbf{x}_k^i - \mathbf{x}_k^{i-1}\|^2.$$

Noting the definition $\mathbf{g}_{\mathcal{X},k,i} = \frac{1}{\gamma_{k,i}} \left( \mathbf{x}_k^{i-1} - \mathbf{x}_k^i \right)$, one has

$$f(\mathbf{x}_k^i) \leq f(\mathbf{x}_k^{i-1}) - \gamma_{k,i} \langle \nabla_i f(\mathbf{x}_k^{i-1}), \mathbf{g}_{\mathcal{X},k,i} \rangle + \frac{L_i}{2} \gamma_{k,i}^2 \|\mathbf{g}_{\mathcal{X},k,i}\|^2.$$

Lemma 1 of Ghadimi et al. (2014) (stated as Lemma E.2 above) applies identically through block-wise optimality on $\mathcal{X}_i$ because $\mathbf{g}_{\mathcal{X},k,i} = P_{\mathcal{X}_i}(\mathbf{x}_k^{i-1}, \nabla_i f(\mathbf{x}_k^{i-1}), \gamma_{k,i})$. Thus for any value $\nabla_i f(\mathbf{x}_k^{i-1})$ takes,

$$f(\mathbf{x}_k^i) \leq f(\mathbf{x}_k^{i-1}) - \left[ \alpha\gamma_{k,i} \|\mathbf{g}_{\mathcal{X},k,i}\|^2 + h(\mathbf{x}_k^i) - h(\mathbf{x}_k^{i-1}) \right] + \frac{L_i}{2} \gamma_{k,i}^2 \|\mathbf{g}_{\mathcal{X},k,i}\|^2,$$

and thus,

$$f(\mathbf{x}_k^i) + h(\mathbf{x}_k^i) \leq f(\mathbf{x}_k^{i-1}) + h(\mathbf{x}_k^{i-1}) - \left[ \alpha\gamma_{k,i} - \frac{L_i}{2}\gamma_{k,i}^2 \right] \|g_{\mathcal{X},k,i}\|^2.$$

The right-hand side above only becomes larger, taking $L_i = L$ to be the global smoothness constant. Introducing a sum over sub-iterates and total iterates, one has

$$\sum_{k,i}^{N,p} f(\mathbf{x}_k^i) + h(\mathbf{x}_k^i) \leq \sum_{k,i}^{N,p} f(\mathbf{x}_k^{i-1}) + h(\mathbf{x}_k^{i-1}) - \sum_{k,i}^{N,p} \left[ \alpha\gamma_{k,i} - \frac{L}{2}\gamma_{k,i}^2 \right] \|\mathbf{g}_{\mathcal{X},k,i}\|^2,$$

$$\sum_{k,i}^{N,p} \Phi(\mathbf{x}_k^i) \leq \sum_{k,i}^{N,p} \Phi(\mathbf{x}_k^{i-1}) - \sum_{k,i}^{N,p} \left[ \alpha\gamma_{k,i} - \frac{L}{2}\gamma_{k,i}^2 \right] \|\mathbf{g}_{\mathcal{X},k,i}\|^2.$$

Noting the end-point condition $\Phi(\mathbf{x}_k^p) = \Phi(\mathbf{x}_{k+1}^0)$, one may cancel all intermediate terms:

$$\Phi^* \leq \Phi(\mathbf{x}_N) \leq \Phi(\mathbf{x}_0) - \sum_{k,i}^{N,p} \left[ \alpha\gamma_{k,i} - \frac{L}{2}\gamma_{k,i}^2 \right] \|\mathbf{g}_{\mathcal{X},k,i}\|^2$$

Thus one finds the upper bound in terms of $\Phi(\mathbf{x}_0)$ and the minimum value $\Phi^* = \min_{\mathbf{x} \in \mathcal{X}} \Phi(\mathbf{x})$ of $\Phi$:

$$\sum_k^N \sum_i^p \left[ \alpha\gamma_{k,i} - \frac{L}{2}\gamma_{k,i}^2 \right] \|\mathbf{g}_{\mathcal{X},k,i}\|^2 \leq \Phi(\mathbf{x}_0) - \Phi^*. \tag{37}$$

Taking $\gamma_{k,i} = \alpha/L$ as in Ghadimi et al. (2014), the bracketed term directly above becomes $\alpha^2/2L$, and one has:

$$\sum_k^N \left[ \frac{\alpha^2}{2L} \right] \left( \min_k \Delta(\mathbf{x}_k, \mathbf{x}_{k-1}) \right) = \sum_k^N \left[ \frac{\alpha^2}{2L} \right] \left( \min_k \sum_i^p \|\mathbf{g}_{\mathcal{X},k,i}\|^2 \right)$$

$$\leq \sum_k^N \left[ \frac{\alpha^2}{2L} \right] \sum_i^p \|\mathbf{g}_{\mathcal{X},k,i}\|^2$$

$$\leq \Phi(\mathbf{x}_0) - \Phi^* = D^2 L,$$

where $D$ is as in (36), and where we used (37) to obtain the last line. Thus,

$$\min_k \Delta(\mathbf{x}_k, \mathbf{x}_{k-1}) \leq \frac{D^2 L}{N(\alpha^2/2L)} = \frac{2D^2 L^2}{N\alpha^2},$$

completing the proof. $\qquad\square$

**Definition E.4** (Relative smoothness). Let $\beta > 0$ and let $g \in \mathcal{C}^1(\mathbb{R}^n, \mathbb{R})$ be continuously differentiable. Additionally, let $\omega$ be a distance generating function with associated prox-function $V$. The function $g$ is $\beta$-*smooth relative to* $\omega$ if the following holds:

$$g(\boldsymbol{y}) \leq g(\boldsymbol{x}) + \langle \nabla \omega(\boldsymbol{x}), \boldsymbol{x} - \boldsymbol{y} \rangle + \beta V(\boldsymbol{y}, \boldsymbol{x})$$

**Proposition E.5.** *Let $\epsilon > 0$ be a predefined error tolerance, $r > 0$ a small rank parameter, $\delta \in (0, \frac{1}{r})$ a lower-bound parameter, and $N$ the number of inner iterations required for the semi-relaxed projection to converge for each iteration $k$ as $\|\boldsymbol{Q}^{\mathrm{T}}\mathbf{1}_n - \boldsymbol{g}_{\boldsymbol{Q}}^{(k-1)}\|_2 < \epsilon = \frac{1}{N}\left(\frac{1}{r} - \delta\right)$ for $\boldsymbol{g}_{\boldsymbol{Q}}^{(0)} = \frac{1}{r}\mathbf{1}_r$. Then, the FRLC objective*

$$\mathcal{L}_{\mathrm{LC}}(\boldsymbol{Q}, \boldsymbol{R}, \boldsymbol{T}) = \langle \boldsymbol{Q}\operatorname{diag}(1/\boldsymbol{Q}^{\mathrm{T}}\mathbf{1}_n)\boldsymbol{T}\operatorname{diag}(1/\boldsymbol{R}^{\mathrm{T}}\mathbf{1}_m)\boldsymbol{R}^{\mathrm{T}}, \boldsymbol{C}\rangle_F$$

*is component-wise smooth with respect to the variables $\boldsymbol{Q}, \boldsymbol{R}, \boldsymbol{T}$ with smoothness constants $\{\beta_i\}_{i=1}^3$ where $\beta_i = \operatorname{poly}(\|\boldsymbol{C}\|_F, n, m, r, \delta)$.*

*Proof.* The addition of a pair of regularizations on the inner marginals $\tau\mathrm{KL}(\boldsymbol{Q}^{\mathrm{T}}\mathbf{1}_n\|\boldsymbol{Q}_k^{\mathrm{T}}\mathbf{1}_n)$ and $\tau\mathrm{KL}(\boldsymbol{R}^{\mathrm{T}}\mathbf{1}_m\|\boldsymbol{R}_k^{\mathrm{T}}\mathbf{1}_m)$ in 21 and 22 ensures that we may use standard results on relaxed optimal-transport which bound how far the marginal $\boldsymbol{Q}_k^{\mathrm{T}}\mathbf{1}_n$ deviates across iterations.

In particular, for all $\epsilon > 0$ there exists $\tau$ and $N$ (number of iterations) sufficiently large, so that $\|\boldsymbol{R}^{\mathrm{T}}\mathbf{1}_m - \boldsymbol{g}_R\|_2^2 < \epsilon$ and $\|\boldsymbol{Q}^{\mathrm{T}}\mathbf{1}_n - \boldsymbol{g}_Q\|_2^2 < \epsilon$. In particular, Pham et al. (2020) shows that one can attain convergence to any $\epsilon$ for the unbalanced problem in $N = \tilde{O}(m^2/\epsilon)$ iterations (hiding logarithmic and $\tau$-factors) for each sub-problem solving for $\boldsymbol{R}_k$ in Algorithm 4 and analogously $N = \tilde{O}(n^2/\epsilon)$ for $\boldsymbol{Q}$. Under the uniform initialization of $\boldsymbol{g}_R$, and after $N$ iterations, we have:

$$\|\boldsymbol{g}_R^{(0)} - \boldsymbol{g}_R^{(N)}\|_2 = \left\|\frac{1}{r}\mathbf{1}_r - \boldsymbol{g}_R^{(N)}\right\|_2 \leq \sum_{k=1}^N \|\boldsymbol{g}_R^{(k)} - \boldsymbol{g}_R^{(k-1)}\|_2.$$

With sufficiently large $\tau$ and $N = \tilde{O}(m^2/\epsilon)$ sub-iterations, one can guarantee that:

$$\|\boldsymbol{g}_R^{(k)} - \boldsymbol{g}_R^{(k-1)}\|_2 < \epsilon = \frac{1}{N}\left(\frac{1}{r} - \delta\right).$$

This implies, for all iterations of the algorithm and all indices $i$, that

$$(\boldsymbol{g}_{R_k})_i > \delta, \quad \text{and analogously,} \quad (\boldsymbol{g}_{Q_k})_i > \delta. \tag{38}$$

Thus, by adding the regularization on the inner marginal, one may guarantee a lower-bound on the entries of $\boldsymbol{g}_R$ and $\boldsymbol{g}_Q$. This is essential for demonstrating smoothness of the objective.

First, we consider smoothness in $\boldsymbol{Q}$. We note that the gradient in $\boldsymbol{Q}$ splits into two terms:

$$\begin{aligned}
\nabla_{\boldsymbol{Q}}\mathcal{L}_{\mathrm{LC}}(\boldsymbol{Q}, \boldsymbol{R}, \boldsymbol{T}) &= \nabla_{\boldsymbol{Q}}^{(A)}\mathcal{L}_{\mathrm{LC}} + \nabla_{\boldsymbol{Q}}^{(B)}\mathcal{L}_{\mathrm{LC}} \\
&= \boldsymbol{C}\boldsymbol{R}\boldsymbol{X}^{\mathrm{T}} - \mathbf{1}_n\operatorname{diag}^{-1}((\boldsymbol{C}\boldsymbol{R}\boldsymbol{X}^{\mathrm{T}})^{\mathrm{T}}\boldsymbol{Q}\operatorname{diag}(1/\boldsymbol{g}_Q))^{\mathrm{T}} \\
&= \nabla_{\boldsymbol{Q}}^{(A)}\mathcal{L}_{\mathrm{LC}} - \mathbf{1}_n\operatorname{diag}^{-1}((\nabla_{\boldsymbol{Q}}^{(A)}\mathcal{L}_{\mathrm{LC}})^{\mathrm{T}}\boldsymbol{Q}\operatorname{diag}(1/\boldsymbol{g}_Q))^{\mathrm{T}}
\end{aligned}$$

Where

$$\begin{aligned}
\|\nabla_{\boldsymbol{Q}}&\mathcal{L}_{\mathrm{LC}}(\boldsymbol{Q}_{k+1}, \boldsymbol{R}_k, \boldsymbol{T}_k) - \nabla_{\boldsymbol{Q}}\mathcal{L}_{\mathrm{LC}}(\boldsymbol{Q}_k, \boldsymbol{R}_k, \boldsymbol{T}_k)\|_F \\
&\leq \|\nabla_{\boldsymbol{Q}}^{(A)}\mathcal{L}_{\mathrm{LC}}(\boldsymbol{Q}_{k+1}, \boldsymbol{R}_k, \boldsymbol{T}_k) - \nabla_{\boldsymbol{Q}}^{(A)}\mathcal{L}_{\mathrm{LC}}(\boldsymbol{Q}_k, \boldsymbol{R}_k, \boldsymbol{T}_k)\|_F \\
&\quad + \|\nabla_{\boldsymbol{Q}}^{(B)}\mathcal{L}_{\mathrm{LC}}(\boldsymbol{Q}_{k+1}, \boldsymbol{R}_k, \boldsymbol{T}_k) - \nabla_{\boldsymbol{Q}}^{(B)}\mathcal{L}_{\mathrm{LC}}(\boldsymbol{Q}_k, \boldsymbol{R}_k, \boldsymbol{T}_k)\|_F.
\end{aligned} \tag{39}$$

Starting with the first term on the right side of (39), one has:

$$\|\nabla_{\boldsymbol{Q}}^{(A)}\mathcal{L}_{\mathrm{LC}}(\boldsymbol{Q}_{k+1},\boldsymbol{R}_k,\boldsymbol{T}_k) - \nabla_{\boldsymbol{Q}}^{(A)}\mathcal{L}_{\mathrm{LC}}(\boldsymbol{Q}_k,\boldsymbol{R}_k,\boldsymbol{T}_k)\|_F$$

$$= \|\boldsymbol{C}\boldsymbol{R}_k(\mathrm{diag}(1/\boldsymbol{g}_{Q_k})\boldsymbol{T}_k\mathrm{diag}(1/\boldsymbol{g}_{R_k}))^{\mathrm{T}} - \boldsymbol{C}\boldsymbol{R}_k(\mathrm{diag}(1/\boldsymbol{g}_{Q_{k-1}})\boldsymbol{T}_k\mathrm{diag}(1/\boldsymbol{g}_{R_k}))^{\mathrm{T}}\|_F$$

$$\le \|\mathrm{diag}(1/\boldsymbol{g}_{R_k})\|_F\|\boldsymbol{C}\|_F\|\boldsymbol{R}_k\|_F\|\boldsymbol{T}_k\|_F\|\mathrm{diag}(1/\boldsymbol{g}_{Q_k}) - \mathrm{diag}(1/\boldsymbol{g}_{Q_{k-1}})\|_F.$$

Note that $\|\boldsymbol{R}_k\|_F^2 = \sum_{i,j}(\boldsymbol{R}_k)_{i,j}^2 < \sum_{i,j}(\boldsymbol{R}_k)_{i,j} = 1$, as $\boldsymbol{R}_k$ has marginals which sum to one. The same bound holds for $\|\boldsymbol{T}_k\|_F^2$, which is also a coupling. Invoking the lower-bound (38) of $\delta$ on the entries of the inner marginals, and continuing from the above display,

$$\|\nabla_{\boldsymbol{Q}}^{(A)}\mathcal{L}_{\mathrm{LC}}(\boldsymbol{Q}_{k+1},\boldsymbol{R}_k,\boldsymbol{T}_k) - \nabla_{\boldsymbol{Q}}^{(A)}\mathcal{L}_{\mathrm{LC}}(\boldsymbol{Q}_k,\boldsymbol{R}_k,\boldsymbol{T}_k)\|_F$$

$$\le \frac{\|\boldsymbol{C}\|_F}{\delta}\|\mathrm{diag}(1/\boldsymbol{g}_{Q_k}) - \mathrm{diag}(1/\boldsymbol{g}_{Q_{k-1}})\|_F$$

$$= \frac{\|\boldsymbol{C}\|_F}{\delta}\|\mathrm{diag}(1/\boldsymbol{g}_{Q_{k-1}})\mathrm{diag}(1/\boldsymbol{g}_{Q_k})(\mathrm{diag}(\boldsymbol{g}_{Q_{k-1}}) - \mathrm{diag}(\boldsymbol{g}_{Q_k}))\|_F$$

$$\le \frac{\|\boldsymbol{C}\|_F}{\delta^3}\|\mathrm{diag}(\boldsymbol{g}_{Q_{k-1}}) - \mathrm{diag}(\boldsymbol{g}_{Q_k})\|_F.$$

To further bound the right-hand side above, consider:

$$\|\mathrm{diag}(\boldsymbol{g}_{Q_k}) - \mathrm{diag}(\boldsymbol{g}_{Q_{k-1}})\|_F^2 = \|\boldsymbol{Q}_k^{\mathrm{T}}\mathbf{1}_n - \boldsymbol{Q}_{k-1}^{\mathrm{T}}\mathbf{1}_n\|_2^2 = \sum_{i=1}^{r}\left(\sum_{j=1}^{r}(\boldsymbol{Q}_k)_{i,j} - (\boldsymbol{Q}_{k-1})_{i,j}\right)^2$$

While can easily be upper-bounded by an application of Jensen's inequality as

$$= \sum_{i=1}^{r}r^2\left(\sum_{j=1}^{r}\frac{1}{r}\left((\boldsymbol{Q}_k)_{i,j} - (\boldsymbol{Q}_{k-1})_{i,j}\right)\right)^2$$

$$\le \sum_{i=1}^{r}r^2\left(\sum_{j=1}^{r}\frac{1}{r}\left((\boldsymbol{Q}_k)_{i,j} - (\boldsymbol{Q}_{k-1})_{i,j}\right)^2\right)$$

$$= r\sum_{i=1}^{r}\sum_{j=1}^{r}\left((\boldsymbol{Q}_k)_{i,j} - (\boldsymbol{Q}_{k-1})_{i,j}\right)^2$$

$$= r\|\boldsymbol{Q}_k - \boldsymbol{Q}_{k-1}\|_F^2.$$

Likewise, we have that $\|\mathrm{diag}(\boldsymbol{g}_{R_k}) - \mathrm{diag}(\boldsymbol{g}_{R_{k-1}})\|_F^2 \le r\|\boldsymbol{R}_k - \boldsymbol{R}_{k-1}\|_F^2$. Thus, it holds that

$$\frac{\|\boldsymbol{C}\|_F}{\delta^3}\|\boldsymbol{g}_{Q_k} - \boldsymbol{g}_{Q_{k-1}}\|_2 \le \frac{\|\boldsymbol{C}\|_F\sqrt{r}}{\delta^3}\|\boldsymbol{Q}_k - \boldsymbol{Q}_{k-1}\|_F.$$

Next, we focus on the $\nabla_{\boldsymbol{Q}}^{(B)}$ term. Observe that

$$\|\mathbf{1}_n\mathrm{diag}^{-1}\mathbf{X}\|_F^2 = \mathrm{Tr}(\mathbf{1}_n\mathrm{diag}^{-1}\mathbf{X})^{\mathrm{T}}(\mathbf{1}_n\mathrm{diag}^{-1}\mathbf{X})$$

$$= n\|\mathrm{diag}^{-1}\mathbf{X}\|_2^2 \le n\|\mathbf{X}\|_F^2.$$

Thus:

$$\|\nabla_{\boldsymbol{Q}_{k+1}}^{(B)} - \nabla_{\boldsymbol{Q}_k}^{(B)}\|_F \le \sqrt{n}\|(\nabla_{\boldsymbol{Q}_{k+1}}^{(A)})^{\mathrm{T}}\boldsymbol{Q}_{k+1}\mathrm{diag}(1/\boldsymbol{g}_{Q_{k+1}}) - (\nabla_{\boldsymbol{Q}_k}^{(A)})^{\mathrm{T}}\boldsymbol{Q}_k\mathrm{diag}(1/\boldsymbol{g}_{Q_k})\|_F$$

Adding and subtracting terms in the norm and applying triangle inequality:

$$\sqrt{n}\|(\nabla_{\boldsymbol{Q}_{k+1}}^{(A)})^{\mathrm{T}}\boldsymbol{Q}_{k+1}\mathrm{diag}(1/\boldsymbol{g}_{Q_{k+1}}) - (\nabla_{\boldsymbol{Q}_k}^{(A)})^{\mathrm{T}}\boldsymbol{Q}_{k+1}\mathrm{diag}(1/\boldsymbol{g}_{Q_{k+1}})$$

$$+ (\nabla_{\boldsymbol{Q}_k}^{(A)})^{\mathrm{T}}\boldsymbol{Q}_{k+1}\mathrm{diag}(1/\boldsymbol{g}_{Q_{k+1}}) - (\nabla_{\boldsymbol{Q}_k}^{(A)})^{\mathrm{T}}\boldsymbol{Q}_k\mathrm{diag}(1/\boldsymbol{g}_{Q_k})\|_F$$

$$\le \sqrt{n}\|\nabla_{\boldsymbol{Q}_{k+1}}^{(A)} - \nabla_{\boldsymbol{Q}_k}^{(A)}\|_F\|\boldsymbol{Q}_{k+1}\|_F\|\mathrm{diag}(1/\boldsymbol{g}_{Q_{k+1}})\|_F$$

$$+ \sqrt{n}\|\nabla_{\boldsymbol{Q}_k}^{(A)}\|_F\|(\boldsymbol{Q}_{k+1}\mathrm{diag}(1/\boldsymbol{g}_{Q_{k+1}}) - \boldsymbol{Q}_k\mathrm{diag}(1/\boldsymbol{g}_{Q_k})\|_F$$

Invoking the lower-bound on the marginal, continuing from the above display,

$$\leq \frac{\sqrt{n}}{\delta}\|\nabla^{(A)}_{\boldsymbol{Q}_{k+1}} - \nabla^{(A)}_{\boldsymbol{Q}_k}\|_F$$

$$+ \sqrt{n}\|\nabla^{(A)}_{\boldsymbol{Q}_k}\|_F\|(\boldsymbol{Q}_{k+1}\mathrm{diag}(1/\boldsymbol{g}_{\boldsymbol{Q}_{k+1}}) - \boldsymbol{Q}_k\mathrm{diag}(1/\boldsymbol{g}_{\boldsymbol{Q}_k}))\|_F$$

Let us consider $\|\nabla^{(A)}_{\boldsymbol{Q}_k}\|_F \leq \|\boldsymbol{X}_k\|_F\|\boldsymbol{C}\|_F\|\boldsymbol{R}_k\|_F \leq \|\boldsymbol{X}_k\|_F\|\boldsymbol{C}\|_F$. We have that:

$$\|\boldsymbol{X}_k\|_F^2 = \sum_{i,j} \frac{1}{\boldsymbol{g}^2_{(\boldsymbol{Q}_k)_i}}\boldsymbol{T}^2_{ij}\frac{1}{\boldsymbol{g}^2_{(\boldsymbol{R}_k)_j}}$$

Where we always have that $\boldsymbol{T}_{ij} \leq \boldsymbol{g}_{\boldsymbol{Q}i}$ and $\boldsymbol{T}_{ij} \leq \boldsymbol{g}_{\boldsymbol{R}j}$ by definition of $\boldsymbol{T}$ as a coupling. As such:

$$\leq \sum_{ij} \frac{1}{\boldsymbol{g}^2_{\boldsymbol{Q}i}}(\boldsymbol{g}_{\boldsymbol{Q}i}\boldsymbol{g}_{\boldsymbol{R}j})\frac{1}{\boldsymbol{g}^2_{\boldsymbol{R}j}} = \sum_{ij} \frac{1}{\boldsymbol{g}_{\boldsymbol{Q}i}}\frac{1}{\boldsymbol{g}_{\boldsymbol{R}j}} = \left\langle \boldsymbol{g}_{\boldsymbol{Q}}^{-1}\boldsymbol{g}_{\boldsymbol{R}}^{-T}, \boldsymbol{1}_m\boldsymbol{1}_r^{\mathrm{T}} \right\rangle_F \leq \frac{mr}{\delta^2}$$

Thus $\|\nabla^{(A)}_{\boldsymbol{Q}_k}\|_F \leq \frac{\sqrt{mr}}{\delta}\|\boldsymbol{C}\|_F$, and the bound above reduces to

$$\leq \frac{\sqrt{n}}{\delta}\|\nabla^{(A)}_{\boldsymbol{Q}_{k+1}} - \nabla^{(A)}_{\boldsymbol{Q}_k}\|_F$$

$$+ \frac{\sqrt{nmr}}{\delta}\|\boldsymbol{C}\|_F\|(\boldsymbol{Q}_{k+1}\mathrm{diag}(1/\boldsymbol{g}_{\boldsymbol{Q}_{k+1}}) - \boldsymbol{Q}_k\mathrm{diag}(1/\boldsymbol{g}_{\boldsymbol{Q}_k}))\|_F$$

Further bounding the last term in the norm

$$\frac{\sqrt{nmr}}{\delta}\|\boldsymbol{C}\|_F\|(\boldsymbol{Q}_{k+1}\mathrm{diag}(1/\boldsymbol{g}_{\boldsymbol{Q}_{k+1}}) - \boldsymbol{Q}_k\mathrm{diag}(1/\boldsymbol{g}_{\boldsymbol{Q}_{k+1}})$$

$$+ \boldsymbol{Q}_k\mathrm{diag}(1/\boldsymbol{g}_{\boldsymbol{Q}_{k+1}}) - \boldsymbol{Q}_k\mathrm{diag}(1/\boldsymbol{g}_{\boldsymbol{Q}_k})\|_F$$

$$\leq \frac{\sqrt{nmr}}{\delta}\|\boldsymbol{C}\|_F(\|\mathrm{diag}(1/\boldsymbol{g}_{\boldsymbol{Q}_{k+1}})\|_F\|\boldsymbol{Q}_{k+1} - \boldsymbol{Q}_k\|_F$$

$$+ \|\boldsymbol{Q}_k\|_F\|\mathrm{diag}(1/\boldsymbol{g}_{\boldsymbol{Q}_{k+1}}) - \mathrm{diag}(1/\boldsymbol{g}_{\boldsymbol{Q}_k})\|_F)$$

$$\leq \frac{\sqrt{nmr}}{\delta}\|\boldsymbol{C}\|_F\left(\frac{1}{\delta} + \frac{\sqrt{r}}{\delta^2}\right)\|\boldsymbol{Q}_{k+1} - \boldsymbol{Q}_k\|_F$$

Thus, the final bound on the $\nabla^{(B)}_{\boldsymbol{Q}}$ term is:

$$\leq \frac{\sqrt{n}}{\delta}\|\nabla^{(A)}_{\boldsymbol{Q}_{k+1}} - \nabla^{(A)}_{\boldsymbol{Q}_k}\|_F + \frac{\sqrt{nmr}}{\delta}\|\boldsymbol{C}\|_F\|(\boldsymbol{Q}_{k+1}\mathrm{diag}(1/\boldsymbol{g}_{\boldsymbol{Q}_{k+1}}) - \boldsymbol{Q}_k\mathrm{diag}(1/\boldsymbol{g}_{\boldsymbol{Q}_k})\|_F$$

$$\leq \left(\frac{\|\boldsymbol{C}\|_F\sqrt{nr}}{\delta^4} + \frac{\sqrt{nmr}}{\delta^2}\|\boldsymbol{C}\|_F\left(1 + \frac{\sqrt{r}}{\delta}\right)\right)\|\boldsymbol{Q}_{k+1} - \boldsymbol{Q}_k\|_F$$

The total component-wise smoothness bound on $\boldsymbol{Q}$ is then

$$\|\nabla_{\boldsymbol{Q}}\mathcal{L}_{\mathrm{LC}}(\boldsymbol{Q}_{k+1}, \boldsymbol{R}_k, \boldsymbol{T}_k) - \nabla_{\boldsymbol{Q}}\mathcal{L}_{\mathrm{LC}}(\boldsymbol{Q}_k, \boldsymbol{R}_k, \boldsymbol{T}_k)\|_F$$

$$\leq \frac{\|\boldsymbol{C}\|_F}{\delta^2}\left(\left(\frac{\sqrt{nr}}{\delta^2} + \sqrt{nmr}\left(1 + \frac{\sqrt{r}}{\delta}\right)\right) + \frac{\sqrt{r}}{\delta}\right)\|\boldsymbol{Q}_{k+1} - \boldsymbol{Q}_k\|_F$$

Identical reasoning applies for $\nabla_{\boldsymbol{R}}\mathcal{L}_{\mathrm{LC}}$, where we similarly have the gradient split into two terms:

$$\nabla_{\boldsymbol{R}}\mathcal{L}_{\mathrm{LC}}(\boldsymbol{Q}, \boldsymbol{R}, \boldsymbol{T}) = \nabla^{(A)}_{\boldsymbol{R}}\mathcal{L}_{\mathrm{LC}}\nabla^{(B)}_{\boldsymbol{R}}\mathcal{L}_{\mathrm{LC}}$$

$$= \boldsymbol{C}^{\mathrm{T}}\boldsymbol{Q}\boldsymbol{X} - \boldsymbol{1}_m\mathrm{diag}^{-1}(\mathrm{diag}(1/\boldsymbol{g}_R)\boldsymbol{R}^{\mathrm{T}}\boldsymbol{C}^{\mathrm{T}}\boldsymbol{Q}\boldsymbol{X})^{\mathrm{T}}$$

$$= \nabla^{(A)}_{\boldsymbol{R}}\mathcal{L}_{\mathrm{LC}} - \boldsymbol{1}_m\mathrm{diag}^{-1}(\mathrm{diag}(1/\boldsymbol{g}_R)\boldsymbol{R}^{\mathrm{T}}\nabla^{(A)}_{\boldsymbol{R}}\mathcal{L}_{\mathrm{LC}})^{\mathrm{T}}$$

As before, we may first show smoothness in $\nabla_{\boldsymbol{R}}^{(A)}$ using the same steps as for $\boldsymbol{Q}$

$$\|\nabla_{\boldsymbol{R}}^{(A)}\mathcal{L}_{\mathrm{LC}}(\boldsymbol{Q}_{k+1},\boldsymbol{R}_{k+1},\boldsymbol{T}_k) - \nabla_{\boldsymbol{R}}^{(A)}\mathcal{L}_{\mathrm{LC}}(\boldsymbol{Q}_{k+1},\boldsymbol{R}_k,\boldsymbol{T}_k)\|_F$$

$$= \|\boldsymbol{C}^{\mathrm{T}}\boldsymbol{Q}_{k+1}\mathrm{diag}(1/\boldsymbol{g}_{\boldsymbol{Q}_{k+1}})\boldsymbol{T}_k\mathrm{diag}(1/\boldsymbol{g}_{\boldsymbol{R}_{k+1}}) - \boldsymbol{C}^{\mathrm{T}}\boldsymbol{Q}_{k+1}\mathrm{diag}(1/\boldsymbol{g}_{\boldsymbol{Q}_{k+1}})\boldsymbol{T}_k\mathrm{diag}(1/\boldsymbol{g}_{\boldsymbol{R}_k})\|_F$$

$$\leq \|\boldsymbol{C}\|_F\|\mathrm{diag}(1/\boldsymbol{g}_{\boldsymbol{Q}_{k+1}})\|_F\|\boldsymbol{Q}_{k+1}\|_F\|\boldsymbol{T}_k\|_F\|\mathrm{diag}(1/\boldsymbol{g}_{\boldsymbol{R}_{k+1}}) - \mathrm{diag}(1/\boldsymbol{g}_{\boldsymbol{R}_k})\|_F$$

$$\leq \frac{\|\boldsymbol{C}\|_F}{\delta}\|\mathrm{diag}(1/\boldsymbol{g}_{\boldsymbol{R}_{k+1}}) - \mathrm{diag}(1/\boldsymbol{g}_{\boldsymbol{R}_k})\|_F$$

$$\leq \frac{\|\boldsymbol{C}\|_F}{\delta^3}\|\boldsymbol{g}_{\boldsymbol{R}_{k+1}} - \boldsymbol{g}_{\boldsymbol{R}_k}\|_2$$

$$\leq \frac{\|\boldsymbol{C}\|_F\sqrt{r}}{\delta^3}\|\boldsymbol{R}_{k+1} - \boldsymbol{R}_k\|_F.$$

For $\nabla_{\boldsymbol{R}}^{(B)}$, one may use the same reasoning as before to find:

$$\|\nabla_{\boldsymbol{R}}^{(B)}\mathcal{L}_{\mathrm{LC}}(\boldsymbol{Q}_{k+1},\boldsymbol{R}_{k+1},\boldsymbol{T}_k) - \nabla_{\boldsymbol{R}}^{(B)}\mathcal{L}_{\mathrm{LC}}(\boldsymbol{Q}_{k+1},\boldsymbol{R}_k,\boldsymbol{T}_k)\|_F$$

$$\leq \|\mathbf{1}_m\left[\mathrm{diag}^{-1}(\mathrm{diag}(1/\boldsymbol{g}_{\boldsymbol{R}_{k+1}})\boldsymbol{R}_{k+1}^{\mathrm{T}}\nabla_{\boldsymbol{R}_{k+1}}^{(A)}\mathcal{L}_{\mathrm{LC}}) - \mathrm{diag}^{-1}(\mathrm{diag}(1/\boldsymbol{g}_{\boldsymbol{R}_k})\boldsymbol{R}_k^{\mathrm{T}}\nabla_{\boldsymbol{R}_k}^{(A)}\mathcal{L}_{\mathrm{LC}})\right]^{\mathrm{T}}\|_F$$

$$\leq \sqrt{m}\|\mathrm{diag}(1/\boldsymbol{g}_{\boldsymbol{R}_{k+1}})\boldsymbol{R}_{k+1}^{\mathrm{T}}\nabla_{\boldsymbol{R}_{k+1}}^{(A)}\mathcal{L}_{\mathrm{LC}} - \mathrm{diag}(1/\boldsymbol{g}_{\boldsymbol{R}_k})\boldsymbol{R}_k^{\mathrm{T}}\nabla_{\boldsymbol{R}_k}^{(A)}\mathcal{L}_{\mathrm{LC}}\|_F$$

As before, one may apply three rounds of triangle inequality inside the norm to bound this directly in terms of $\|\nabla_{\boldsymbol{R}_{k+1}}^{(A)} - \nabla_{\boldsymbol{R}_k}^{(A)}\|_F$, $\|\mathrm{diag}(1/\boldsymbol{R}_{k+1}) - \mathrm{diag}(1/\boldsymbol{R}_k)\|_F$, and $\|\boldsymbol{R}_{k+1} - \boldsymbol{R}_k\|_F$. Each of these terms is smooth in $\boldsymbol{R}$ by the lower-bound argument, so that smoothness in $\boldsymbol{R}$ holds analogously to $\boldsymbol{Q}$. The remainder of the proof for smoothness in $\boldsymbol{R}$ thus follows identically to that of $\boldsymbol{Q}$ above.

For $\boldsymbol{T}$, the component-wise bound of

$$\|\nabla_{\boldsymbol{T}_{k+1}}\mathcal{L}_{\mathrm{LC}} - \nabla_{\boldsymbol{T}_k}\mathcal{L}_{\mathrm{LC}}\|_F = \|\boldsymbol{Q}_{k+1}\boldsymbol{C}\boldsymbol{R}_{k+1}^{\mathrm{T}} - \boldsymbol{Q}_{k+1}\boldsymbol{C}\boldsymbol{R}_{k+1}^{\mathrm{T}}\|_F^2 \leq L_T\|\boldsymbol{T}_{k+1} - \boldsymbol{T}_k\|_F$$

holds trivially for any $L_T > 0$ as the gradient is uniquely determined by $\boldsymbol{Q}$ and $\boldsymbol{R}$ alone. Thus there exist $L_{\boldsymbol{Q}}, L_{\boldsymbol{R}}, L_{\boldsymbol{T}} > 0$ as component-wise smoothness constants for $\mathcal{L}_{\mathrm{LC}}(\boldsymbol{Q},\boldsymbol{R},\boldsymbol{T})$. $\qquad\square$

We next prove Proposition 3.4, restated just below for convenience.

**Proposition E.6** (Proposition 3.4). *Consider the FRLC objective* (8). *The FRLC algorithm, Algorithm 4, yields $\beta$-smooth iterates for $\beta = \mathrm{poly}(\|\boldsymbol{C}\|_F, m, r, \delta)$, where $\delta$ denotes the lower-bound on the entries of $\boldsymbol{g}_{\boldsymbol{Q}}, \boldsymbol{g}_{\boldsymbol{R}}$. Consider the convergence metric of 3.3 adapted from Ghadimi et al. (2014), given as:*

$$\Delta_k(\boldsymbol{x}_k, \boldsymbol{x}_{k+1}) = \sum_{i=1}^p \|\mathbf{g}_{\mathcal{X},k,i}\|^2 = \frac{1}{\gamma_k^2}\left[\|\boldsymbol{Q}_k - \boldsymbol{Q}_{k-1}\|_F^2 + \|\boldsymbol{R}_k - \boldsymbol{R}_{k-1}\|_F^2 + \|\boldsymbol{T}_k - \boldsymbol{T}_{k-1}\|_F^2\right]$$

*for $\boldsymbol{x}_k = (\boldsymbol{Q}_k, \boldsymbol{T}_k, \boldsymbol{R}_k)$. Define the gap to the optimal solution $D$ as in (36), and let $L = \sup_i(L_i)$ to be the global smoothness constant across all components. Then for $\gamma_k = \alpha/L$ as defined in 3.3 the FRLC algorithm has the non-asymptotic stationary convergence guarantee that:*

$$\min_{k \in 1,..,N-1} \Delta_k \leq \frac{2D^2L^2}{N\alpha^2}$$

*Proof.* The proof of the non-asymptotic stationary convergence of mirror descent of Ghadimi et al. (2014), adapted for coordinate mirror descent using the block-descent lemma in 3.3, only requires component-wise smoothness in $(\boldsymbol{Q}, \boldsymbol{R}, \boldsymbol{T})$. The proof of this for FRLC is given in **??**, and the guarantee follows directly for this value of $L = \max(L_{\boldsymbol{Q}}, L_{\boldsymbol{R}}, L_{\boldsymbol{T}}) = \mathrm{poly}(\|\boldsymbol{C}\|_F, m, r, \delta)$. $\qquad\square$

**Proposition E.7.** *Low-rank Approximation Error. Let $\mathrm{SR\text{-}W}_r^\star$ denote the optimal rank-$r$ approximation for the semi-relaxed low-rank optimal transport problem, and let $\mathrm{SR\text{-}W}^\star$ denote the optimal solution for the full-rank semi-relaxed optimal transport problem.*

*Additionally, suppose $c_{\boldsymbol{b}} = \sum_{j=1}^m \boldsymbol{b}_j$ denotes the sum of the entries of the second marginal ($c_{\boldsymbol{b}} = 1$ if a probability measure). Then we have the following upper-bound on the objective error:*

$$|\mathrm{SR\text{-}W}_r^\star(\mu_{\boldsymbol{b}}) - \mathrm{SR\text{-}W}^\star(\mu_{\boldsymbol{b}})| \leq c_{\boldsymbol{b}}\left(\max_{p,q}\{\boldsymbol{C}_{pq}\} - \min_{p,q}\{\boldsymbol{C}_{pq}\}\right)\ln\left(\min\{n,m\}/(r-1)\right)$$

*We note that this bound also applies for the standard balanced optimal transport case, giving:*

$$|\mathrm{W}_r^\star(\mu_{\boldsymbol{a}}, \mu_{\boldsymbol{b}}) - \mathrm{W}^\star(\mu_{\boldsymbol{a}}, \mu_{\boldsymbol{b}})| \leq \left( \max_{p,q}\{\boldsymbol{C}_{pq}\} - \min_{p,q}\{\boldsymbol{C}_{pq}\} \right) \ln\left( \min\{n, m\}/(r-1) \right)$$

*and improves the previous bound of* $|\mathrm{W}_r^\star(\mu_{\boldsymbol{a}}, \mu_{\boldsymbol{b}}) - \mathrm{W}^\star(\mu_{\boldsymbol{a}}, \mu_{\boldsymbol{b}})| \leq \|\boldsymbol{C}\|_\infty \ln\left( \min\{n, m\}/(r-1) \right)$ *as the distance matrix* $\boldsymbol{C}$ *contains only non-negative entries.*

*Proof.* We adapt the proof from Scetbon & Cuturi (2022) for the balanced case, which previously gave the bound:

$$|\mathrm{W}_r^\star(\mu_{\boldsymbol{a}}, \mu_{\boldsymbol{b}}) - \mathrm{W}^\star(\mu_{\boldsymbol{a}}, \mu_{\boldsymbol{b}})| \leq \|\boldsymbol{C}\|_\infty \ln\left( \min\{n, m\}/(r-1) \right)$$

In particular, for $z = \min\{m, n\}$, there exists an optimal $\mathrm{rank}_+(\boldsymbol{P}^*) \leq z$ where one may express the optimal solution for the non-negative coupling matrix $\boldsymbol{P}$ as a sum of $z$ rank-one, non-negative outer products $\tilde{\boldsymbol{q}}_k \tilde{\boldsymbol{r}}_k^{\mathrm{T}} \succcurlyeq 0$:

$$\boldsymbol{P}^* = \sum_{k=1}^z \tilde{\boldsymbol{q}}_k \tilde{\boldsymbol{r}}_k^{\mathrm{T}} = \sum_{k=1}^z \lambda_k \boldsymbol{q}_k \boldsymbol{r}_k^{\mathrm{T}}$$

Where we write this sum in terms of normalized vectors $\boldsymbol{q}_k = \tilde{\boldsymbol{q}}_k / \|\tilde{\boldsymbol{q}}_k\|_1$, $\boldsymbol{r}_k = \tilde{\boldsymbol{r}}_k / \|\tilde{\boldsymbol{r}}_k\|_1$, and $\lambda_k = \|\tilde{\boldsymbol{r}}_k\|_1 \|\tilde{\boldsymbol{q}}_k\|_1$. Without any loss of generality, $(\lambda_k)_{k=1}^z$ is ordered in terms of decreasing value such that $\lambda_1 \geq \lambda_2 \geq ... \geq \lambda_z$. Note that we have a fixed constraint for the sum of the entries of $\boldsymbol{P}$ for the semi-relaxed case, assuming $\boldsymbol{b}$ is a general positive measure, as $\mathbf{1}_n^{\mathrm{T}} \boldsymbol{P} \mathbf{1}_m = \left(\boldsymbol{P}^{\mathrm{T}} \mathbf{1}_n\right)^{\mathrm{T}} \mathbf{1}_m = \boldsymbol{b}^{\mathrm{T}} \mathbf{1}_m = \sum_{j=1}^m \boldsymbol{b}_j := c_{\boldsymbol{b}}$ (where $c_{\boldsymbol{b}} = 1$ if $\boldsymbol{b}$ is chosen to be a probability measure, i.e. in the balanced case). Moreover, it is simple to observe for these ordered values that $\lambda_k \leq (c_{\boldsymbol{b}}/k)$. As in Scetbon & Cuturi (2022), define the weighted average of the bottom $z - r + 1$ vectors of the decomposition to be:

$$\boldsymbol{\alpha}_r = \frac{\sum_{i=r}^z \lambda_i \boldsymbol{q}_i}{\sum_{i=r}^z \lambda_i}$$

$$\boldsymbol{\beta}_r = \frac{\sum_{i=r}^z \lambda_i \boldsymbol{r}_i}{\sum_{i=r}^z \lambda_i}$$

And take the rank-$r$ approximation using the optimal $r - 1$ vectors of OPT and this weighted average of the bottom to be:

$$\tilde{\boldsymbol{P}}_r = \sum_{i=1}^{r-1} \lambda_i \boldsymbol{q}_i \boldsymbol{r}_i^{\mathrm{T}} + \left( \sum_{i=r}^z \lambda_i \right) \boldsymbol{\alpha}_r \boldsymbol{\beta}_r^{\mathrm{T}}$$

Where, by the assumption that $\boldsymbol{P}^* \in \Pi_{\boldsymbol{b}}$ is feasible:

$$\tilde{\boldsymbol{P}}_r^{\mathrm{T}} \mathbf{1}_n = \sum_{i=1}^{r-1} \lambda_i \boldsymbol{r}_i \boldsymbol{q}_i^{\mathrm{T}} \mathbf{1}_n + \left( \sum_{i=r}^z \lambda_i \right) \boldsymbol{\beta}_r \boldsymbol{\alpha}_r^{\mathrm{T}} \mathbf{1}_n = \sum_{i=1}^{r-1} \lambda_i \boldsymbol{r}_i + \left( \sum_{i=r}^z \lambda_i \right) \boldsymbol{\beta}_r = \sum_{i=1}^z \lambda_i \boldsymbol{r}_i = \boldsymbol{b}$$

Thus $\tilde{\boldsymbol{P}}_r \in \Pi_{\boldsymbol{b},r}$ is a feasible rank-$r$ solution by feasibility of $\boldsymbol{P}^*$. One can verify that if $\boldsymbol{P}^* \in \Pi_{\boldsymbol{a},\boldsymbol{b}}$, then $\tilde{\boldsymbol{P}}_r \mathbf{1}_m = \boldsymbol{a}$ and the solution is again feasible. From this, we observe that the difference between this solution and $\boldsymbol{P}^*$ is an upper-bound to the difference between $\boldsymbol{P}^*$ and the optimal rank-$r$ solution:

$$|\mathrm{SR\text{-}W}_r^\star(\mu_{\boldsymbol{a}}, \mu_{\boldsymbol{b}}) - \mathrm{SR\text{-}W}^\star(\mu_{\boldsymbol{a}}, \mu_{\boldsymbol{b}})| = |\langle \boldsymbol{P}_r^*, \boldsymbol{C} \rangle_F - \langle \boldsymbol{P}^*, \boldsymbol{C} \rangle_F| \leq \langle \tilde{\boldsymbol{P}}_r, \boldsymbol{C} \rangle_F - \langle \boldsymbol{P}^*, \boldsymbol{C} \rangle_F$$

$$= \left\langle \sum_{i=1}^{r-1} \lambda_i \boldsymbol{q}_i \boldsymbol{r}_i^{\mathrm{T}} + \left( \sum_{i=r}^z \lambda_i \right) \boldsymbol{\alpha}_r \boldsymbol{\beta}_r^{\mathrm{T}} - \sum_{i=1}^z \lambda_i \boldsymbol{q}_i \boldsymbol{r}_i^{\mathrm{T}}, \boldsymbol{C} \right\rangle_F$$

$$= \left\langle \left( \sum_{i=r}^z \lambda_i \right) \boldsymbol{\alpha}_r \boldsymbol{\beta}_r^{\mathrm{T}} - \sum_{i=r}^z \lambda_i \boldsymbol{q}_i \boldsymbol{r}_i^{\mathrm{T}}, \boldsymbol{C} \right\rangle_F$$

Noting that $\boldsymbol{\alpha}_r, \boldsymbol{\beta}_r$ and $\boldsymbol{q}_i, \boldsymbol{r}_i$ are unit normalized positive vectors, the sum of the entries of the outer product $\mathbf{1}_n^{\mathrm{T}} \boldsymbol{q}_i \boldsymbol{r}_i^{\mathrm{T}} \mathbf{1}_m = 1$, and likewise for $\boldsymbol{\alpha}_r \boldsymbol{\beta}_r^{\mathrm{T}}$. Thus, continuing from the above display:

$$
\begin{aligned}
&= (\sum_{i=r}^{z} \lambda_i) \langle \boldsymbol{\alpha}_r \boldsymbol{\beta}_r^{\mathrm{T}}, \boldsymbol{C} \rangle_F - \sum_{i=r}^{z} \lambda_i \langle \boldsymbol{q}_i \boldsymbol{r}_i^{\mathrm{T}}, \boldsymbol{C} \rangle_F \\
&\leq (\sum_{i=r}^{z} \lambda_i) \langle \boldsymbol{\alpha}_r \boldsymbol{\beta}_r^{\mathrm{T}}, \boldsymbol{C} \rangle_F - (\sum_{i=r}^{z} \lambda_i) \min_{p,q} \{\boldsymbol{C}_{pq}\} \\
&\leq \left( \max_{p,q} \{\boldsymbol{C}_{pq}\} - \min_{p,q} \{\boldsymbol{C}_{pq}\} \right) \sum_{i=r}^{z} \lambda_i \\
&\leq c_{\boldsymbol{b}} \left( \max_{p,q} \{\boldsymbol{C}_{pq}\} - \min_{p,q} \{\boldsymbol{C}_{pq}\} \right) \sum_{i=r}^{z} \frac{1}{i} \\
&\leq c_{\boldsymbol{b}} \left( \max_{p,q} \{\boldsymbol{C}_{pq}\} - \min_{p,q} \{\boldsymbol{C}_{pq}\} \right) \ln\left(z/(r-1)\right)
\end{aligned}
$$

Concluding the proof. As discussed, this directly applies to the balanced case (for $c_{\boldsymbol{b}} = 1$). $\qquad\square$

# F   Initialization

We propose a new initialization of the sub-couplings $\boldsymbol{Q}, \boldsymbol{R}, \boldsymbol{T}$ for the LC-factorization. Algorithm 6 generates a random full-rank initial condition in the set of couplings $\Pi_{\boldsymbol{a},\boldsymbol{b}}$ which still satisfies the marginal constraints. It accomplishes this by sampling random matrices which are full-rank and applying the Sinkhorn algorithm to each of them. Scetbon et al. (2021) proposed an initialization which represents an improvement over the rank-1 product measure which is rank-2. Follow-up work proposed initialization using k-means Scetbon & Cuturi (2022). However, this assumes the previous diagonal factorization and is thus not application for generating a latent coupling which may be non-diagonal, non-square, and with two distinct inner marginals. Our initialization is tailored to the LC-factorization, is effective, and has a full-rank guarantee. In particular, higher-rank initializations may exhibit better convergence properties by allowing the gradient to explore a larger set of directions immediately in the optimization. This initialization is given in Algorithm 6.

---
**Algorithm 6**    Initialize-Couplings
___

Input $\boldsymbol{a} \in \Delta_n, \boldsymbol{b} \in \Delta_m, \boldsymbol{g}_Q \in \Delta_r, \boldsymbol{g}_R \in \Delta_r$
$\boldsymbol{C_Q} \sim [0,1]^{n \times r}, \boldsymbol{C_R} \sim [0,1]^{m \times r}, \boldsymbol{C_T} \sim [0,1]^{r \times r}$
$\boldsymbol{K_Q} \leftarrow e^{\boldsymbol{C_Q}}, \boldsymbol{K_R} \leftarrow e^{\boldsymbol{C_R}}, \boldsymbol{K_T} \leftarrow e^{\boldsymbol{C_T}}$
$\boldsymbol{Q} \leftarrow Sinkhorn(\boldsymbol{K_Q}, \boldsymbol{a}, \boldsymbol{g}_Q)$
$\boldsymbol{R} \leftarrow Sinkhorn(\boldsymbol{K_R}, \boldsymbol{b}, \boldsymbol{g}_R)$
$\boldsymbol{T} \leftarrow Sinkhorn(\boldsymbol{K_T}, \boldsymbol{g}_Q = \boldsymbol{Q}^{\mathrm{T}} \mathbf{1}_n, \boldsymbol{g}_R = \boldsymbol{R}^{\mathrm{T}} \mathbf{1}_m)$
Return $(\boldsymbol{Q}, \boldsymbol{R}, \boldsymbol{T})$

---

**Proposition F.1.** *Suppose one samples an initial condition on the optimal transport coupling using Algorithm 6, where we assume $C_{ij} \sim \mathrm{Unif}(0,1)$ such that $\mathbb{P}(\mathrm{rank}(\boldsymbol{C}) < \min\{n,m\}) = 0$. Additionally, suppose that $\boldsymbol{a}, \boldsymbol{b} > \boldsymbol{0}$ holds elementwise for both marginals $\boldsymbol{a} \in \Delta_n, \boldsymbol{b} \in \Delta_r$. Then the elementwise exponential $\exp\{\boldsymbol{C}\}$ (or $\exp\{-\boldsymbol{C}\}$) has full-rank and the return $\mathrm{Sinkhorn}(e^{-\boldsymbol{C}}, \boldsymbol{a}, \boldsymbol{b})$ has full-rank.*

*Proof.* It is established that a random matrix $\boldsymbol{C} \sim [0,1]^{n \times m}$ has full-rank with probability one. For $\boldsymbol{K} = \exp\{\boldsymbol{C}\}$, it holds that the matrix must be entry-wise positive with $\boldsymbol{K}_{ij} \geq 0$. If columns $\boldsymbol{C}_{.,i} \neq \boldsymbol{C}_{.,j}$ then clearly $\boldsymbol{C}_{.,i}^{\odot k} \neq \boldsymbol{C}_{.,j}^{\odot k}$, and if $\boldsymbol{C}_{.,i}, \boldsymbol{C}_{.,j} \succcurlyeq \boldsymbol{0}$ and are independent remain so under element-wise powers. One may easily show this by contrapositive. Suppose there exist constants $c_1, c_2$ such that:

$$
c_1 \boldsymbol{C}_{.,i}^{\odot k} + c_2 \boldsymbol{C}_{.,j}^{\odot k} = 0
$$

As $C_{.,i}, C_{.,j} \succ 0$, without loss of generality one may assume $c_1 > 0$ and $c_2 < 0$. Then:

$$c_1 C_{.,i}^{\odot k} = c_1 \mathbf{1} \odot C_{.,i}^{\odot k} = -c_2 \mathbf{1} \odot C_{.,j}^{\odot k} = -c_2 C_{.,j}^{\odot k} \implies \left(-\frac{c_1}{c_2}\right)^{1/k} \mathbf{1} = c\mathbf{1} = \frac{C_{.,j}}{C_{.,i}}$$

So clearly one has that $C_{.,j} - cC_{.,i} = 0$ for $c > 0$. This implies the columns $C_{.,j}$ and $C_{.,i}$ are dependent. Thus it is clear that elementwise powers of entrywise positive independent vectors preserve independence. The same principle extends trivially to exponentiation of the columns, where if one assumes by contradiction that $c_1 e^{C_{.,i}} + c_2 e^{C_{.,j}} = 0$ for $c_1 > 0$, $c_2 < 0$, one finds $\log\left(-\frac{c_1}{c_2}\right)\mathbf{1} = C_{.,j} - C_{.,i}$. Without loss of generality, assume $0 < -c_2 \le c_1$ and $C_{.,j} > C_{.,i} > 0$, so that $\delta = \log\left(-\frac{c_1}{c_2}\right) \ge 0$. Then, considering constants $q_1, q_2$:

$$q_2 C_{.,j} + q_1 C_{.,i} = (q_1 + q_2)C_{.,i} + \delta q_2 \mathbf{1} = 0$$

Assuming $C_{.,i}$ has greater than one unique entry, which we assume as the entries are sampled densely in $\mathbb{R}$, the two vectors are dependent if and only if $q_1 = -q_2$ and $\delta = 0$, implying $C_{.,i} = C_{.,j}$. Thus, for the set of independent column vectors of $C$, given as $\{C_{.,i}\}_{i=1}^m$, the set $\{e^{C_{.,i}}\}_{i=1}^m$, is also linearly independent. This holds analogously for the row vectors. As $C$ is full-rank and $\text{span}(\{C_{.,i}\}) = \mathbb{R}^{\min\{m,n\}}$, we have that $\text{span}(K) = \mathbb{R}^{\min\{m,n\}}$ as $K = e^C = \sum_{k=0}^{\infty} \frac{C^{\odot k}}{k!}$ (analogously $e^{-C}$) and remains full-rank.

Sinkhorn expresses each variable as

$$X = \text{diag}(u)K\,\text{diag}(v)$$

where $\text{rank}(X) = \text{rank}(\text{diag}(u)K\,\text{diag}(v))$. As Cuturi (2013b) updates the vectors $u \leftarrow a/Kv$ and $v \leftarrow b/K^{\text{T}}u$ from $u_0 = \mathbf{1}_n$ and $v_0 = \mathbf{1}_m$, if $a, b > 0$ holds element-wise, one has that $u, v > 0$ elementwise as well. Then, one has that $\text{null}\,\text{diag}(v) = \{0\}$ and $\text{null}\,\text{diag}(u) = \{0\}$, implying that $\text{rank}(\text{diag}(u)K\,\text{diag}(v)) = \text{rank}(K) = \min\{n, m\}$.

Thus, our initialization returns a random coupling matrix $X \in \Pi_{a,b}$ of full-rank. $\qquad\square$

In the next proposition, we show that one can *analytically* solve for the block-optimal weights $g$ for the factorization of the coupling matrix $P$ as $P = Q\,\text{diag}(1/g)R^{\text{T}}$ Forrow et al. (2019); Scetbon et al. (2021).

**Proposition F.2.** *For the minimization problem expressed as*

$$\min_{g \in \Delta_r} \langle Q\,\text{diag}(1/g)R^{\text{T}}, C\rangle_F$$

*One has the closed-form minimizer of $g^*$ defined entrywise as:*

$$g_i^* = \frac{\sqrt{\omega_i}}{\sum_{j=1}^r \sqrt{\omega_j}}$$

*For $\omega = \text{diag}^{-1}(Q^{\text{T}}CR)$, when $\omega \ge 0$ holds entrywise.*

*Proof.* As $\omega \ge 0$, we consider the simplex condition $\sum_{j=1}^r g_j = 1$ alone. Writing out the Lagrangian associated to our objective, with $\lambda \in \mathbb{R}$ our equality-condition dual variable, we have:

$$\mathcal{L}(g, \lambda) = \langle Q\,\text{diag}(1/g)R^{\text{T}}, C\rangle_F + \lambda(1 - \sum_j g_j)$$

Let us consider a rewriting of the inner product term:

$$\langle Q\,\text{diag}(1/g)R^{\text{T}}, C\rangle_F = \sum_{i=1}^n \sum_{j=1}^m C_{ij} \sum_{k=1}^r Q_{ik}\left(\frac{1}{g_k}\right)R_{kj}^{\text{T}}$$

$$= \sum_{k=1}^r \left(\frac{1}{g_k}\right) \sum_{i=1}^n \sum_{j=1}^m Q_{ki}^{\text{T}} C_{ij} R_{jk}$$

$$= \sum_{k=1}^r \left(\frac{1}{g_k}\right)(Q^{\text{T}}CR)_{k,k}$$

$$= \sum_{k=1}^r \frac{\omega_k}{g_k} = \omega^{\text{T}}(1/g)$$

Where division is interpreted element-wise in the last line. Thus, one can interpret the problem as minimizing the weighted sum of reciprocals of a density. As a result, we can simplify our Lagrangian's Froebenius inner product to a vector dot-product as:

$$\mathcal{L}(\boldsymbol{g}, \lambda) = \boldsymbol{\omega}^{\mathrm{T}} (1/\boldsymbol{g}) - \lambda(1 - \sum_{j=1}^{r} \boldsymbol{g}_j)$$

Thus, the first order condition tells us that the value of the coupling weight $\boldsymbol{g}_j$ is related to $\lambda$ as:

$$\partial_{\boldsymbol{g}_j} \mathcal{L}(\boldsymbol{g}, \lambda) = -\frac{\boldsymbol{\omega}_j}{\boldsymbol{g}_j^2} + \lambda = 0 \implies \boldsymbol{g}_j = \sqrt{\frac{\boldsymbol{\omega}_j}{\lambda}}$$

And by relying on the summation condition on the probability density $\boldsymbol{g}$, yields the Langrange multiplier as

$$\sum_{j=1}^{r} \boldsymbol{g}_j = 1 = \sum_{j=1}^{r} \sqrt{\frac{\boldsymbol{\omega}_j}{\lambda}}$$

so that one finds

$$1 = \frac{1}{\sqrt{\lambda}} \sum_{j=1}^{r} \sqrt{\boldsymbol{\omega}_j} \implies \lambda = \left( \sum_{j=1}^{r} \sqrt{\boldsymbol{\omega}_j} \right)^2$$

Plugging our Lagrange-multiplier into the above expression yields:

$$\boldsymbol{g}_j = \sqrt{\frac{\boldsymbol{\omega}_j}{\lambda}} = \sqrt{\frac{\boldsymbol{\omega}_j}{\left( \sum_{i=1}^{r} \sqrt{\boldsymbol{\omega}_i} \right)^2}} = \frac{\sqrt{\boldsymbol{\omega}_j}}{\sum_{i=1}^{r} \sqrt{\boldsymbol{\omega}_i}}$$

As the Hessian $\nabla_{\boldsymbol{g}}^2 \mathcal{L} = \mathrm{diag}(\frac{\boldsymbol{\omega}}{\boldsymbol{g}^3}) \succ \boldsymbol{0}$, we conclude that this value of $\boldsymbol{g}$ indeed minimizes the loss over $\Delta_r$. □

# G    Alternating updates on the dual variables

For the problem:

$$\inf_{(\boldsymbol{Q}, \boldsymbol{R}_k, \boldsymbol{g}_k) \in \mathcal{C}_1 \cap \mathcal{C}_2} \left( \frac{1}{\gamma_k} \mathrm{KL}(\boldsymbol{Q} \| \boldsymbol{K_Q}) + \tau \mathrm{KL}(\boldsymbol{Q} \boldsymbol{1}_r \| \boldsymbol{a}) - \boldsymbol{\lambda}_1^{\mathrm{T}} \boldsymbol{Q}^{\mathrm{T}} \boldsymbol{1}_n \right) \tag{40}$$

one can find a simple set of semi-relaxed updates for the coupling matrix. We note the primal-dual relationship of Sinkhorn, $\boldsymbol{Q} = \mathrm{diag}(e^{\gamma_k \boldsymbol{f}_1}) \boldsymbol{K_Q} \mathrm{diag}(e^{\gamma_k \boldsymbol{h}_1})$, and consider the entry-wise first-order condition required for the sub-coupling $\boldsymbol{Q}$:

$$0 = \gamma_k^{-1} \log \left( \frac{\boldsymbol{Q}_{ij}}{(\boldsymbol{K_Q})_{ij}} \right) + \tau \log \left( \frac{\langle \boldsymbol{Q}_{i,.}, \boldsymbol{1}_r \rangle}{\boldsymbol{a}_i} \right) - \boldsymbol{\lambda}_{1,i}$$
$$\implies \log \boldsymbol{Q}_{ij} = \tau \gamma_k \log \left( \frac{\boldsymbol{a}_i}{\langle \boldsymbol{Q}_{i,.}, \boldsymbol{1}_r \rangle} \right) + \log (\boldsymbol{K_Q})_{ij} - \boldsymbol{\lambda}_{1j} \gamma_k \tag{41}$$

Thus:

$$\boldsymbol{Q}_{ij} = \left( \frac{\boldsymbol{a}_i}{\boldsymbol{Q}_{i,.}^{\mathrm{T}} \boldsymbol{1}_r} \right)^{\tau \gamma_k} (\boldsymbol{K_Q})_{ij} \, e^{-\boldsymbol{\lambda}_{1j} \gamma_k}$$

And in matrix-form, this yields:

$$\boldsymbol{Q} = \mathrm{diag} \left( \frac{\boldsymbol{a}}{\boldsymbol{Q} \boldsymbol{1}_r} \right)^{\tau \gamma_k} \boldsymbol{K_Q} \, \mathrm{diag}(e^{-\gamma_k \boldsymbol{\lambda}_1})$$
$$= \mathrm{diag}(e^{\gamma_k \boldsymbol{f}_1}) \boldsymbol{K_Q} \, \mathrm{diag}(e^{\gamma_k \boldsymbol{h}_1})$$

And expanding the $\boldsymbol{Q}\boldsymbol{1}_r$ term explicitly, noting that $\boldsymbol{X}\operatorname{diag}(\boldsymbol{v})\boldsymbol{1} = \boldsymbol{X}\boldsymbol{v}$, we have:

$$\operatorname{diag}\left(\frac{\boldsymbol{a}}{\boldsymbol{Q}\boldsymbol{1}_r}\right)^{\tau\gamma_k}\boldsymbol{K_Q}\operatorname{diag}(e^{-\gamma_k\boldsymbol{\lambda}_1})$$

$$= \operatorname{diag}\left(\frac{\boldsymbol{a}}{\operatorname{diag}(e^{\gamma_k\boldsymbol{f}_1})\boldsymbol{K_Q}\operatorname{diag}(e^{\gamma_k\boldsymbol{h}_1})\boldsymbol{1}_r}\right)^{\tau\gamma_k}\boldsymbol{K_Q}\operatorname{diag}(e^{-\gamma_k\boldsymbol{\lambda}_1})$$

$$= \operatorname{diag}\left(\frac{\boldsymbol{a}}{e^{\gamma_k\boldsymbol{f}_1}\odot\boldsymbol{K_Q}e^{\gamma_k\boldsymbol{h}_1}}\right)^{\tau\gamma_k}\boldsymbol{K_Q}\operatorname{diag}(e^{-\gamma_k\boldsymbol{\lambda}_1})$$

Thus, we identify $e^{\gamma_k\boldsymbol{h}_1} = e^{-\gamma_k\boldsymbol{\lambda}_1}$ as the right dual vector, and identify the following relationship in terms of the left dual vector:

$$\left(\frac{\boldsymbol{a}}{e^{\gamma_k\boldsymbol{f}_1}\odot\boldsymbol{K_Q}e^{\gamma_k\boldsymbol{h}_1}}\right)^{\tau\gamma_k} = e^{\gamma_k\boldsymbol{f}_1} \implies e^{\gamma_k\boldsymbol{f}_1} = \left(\frac{\boldsymbol{a}}{\boldsymbol{K_Q}e^{\gamma_k\boldsymbol{h}_1}}\right)^{\frac{\tau}{\tau+1/\gamma_k}}$$

From 40, the condition that $(\boldsymbol{Q}, \boldsymbol{R}_k, \boldsymbol{g}_k) \in \mathcal{C}_1 \cap \mathcal{C}_2$ implies that $\boldsymbol{Q}^{\mathrm{T}}\boldsymbol{1}_m = \boldsymbol{g}_k := \boldsymbol{g}$. As such, we find that:

$$\boldsymbol{Q}^{\mathrm{T}}\boldsymbol{1}_m = \operatorname{diag}(e^{\gamma_k\boldsymbol{h}_1})\boldsymbol{K_Q^{\mathrm{T}}}\operatorname{diag}(e^{\gamma_k\boldsymbol{f}_1})\boldsymbol{1}_m = \operatorname{diag}(e^{\gamma_k\boldsymbol{h}_1})\boldsymbol{K_Q^{\mathrm{T}}}e^{\gamma_k\boldsymbol{f}_1} = \boldsymbol{g}$$

Implying an update for $e^{\gamma_k\boldsymbol{h}_1}$ in the form:

$$e^{\gamma_k\boldsymbol{h}_1} = \left(\frac{\boldsymbol{g}}{\boldsymbol{K_Q^{\mathrm{T}}}e^{\gamma_k\boldsymbol{f}_1}}\right)$$

Analogous reasoning applies for a relaxation of the other marginal, yielding the $\mathrm{SR^R}$-projection and $\mathrm{SR^L}$-projection (i.e. semi-relaxed OT).

---

**Algorithm 7**    $\mathrm{SR^L}$-projection    *(semi-relaxed OT, left marginal relaxed)*

---

Input $\boldsymbol{K}, \gamma, \tau, \boldsymbol{a}, \boldsymbol{b}, \delta$
$\boldsymbol{u} \leftarrow \boldsymbol{1}_n$
$\boldsymbol{v} \leftarrow \boldsymbol{1}_r$
**repeat**
     $\tilde{\boldsymbol{u}} \leftarrow \boldsymbol{u}$
     $\tilde{\boldsymbol{v}} \leftarrow \boldsymbol{v}$
     $\boldsymbol{u} \leftarrow (\boldsymbol{a}/\boldsymbol{K}\boldsymbol{v})^{\tau/(\tau+\gamma^{-1})}$
     $\boldsymbol{v} \leftarrow (\boldsymbol{b}/\boldsymbol{K}^{\mathrm{T}}\boldsymbol{u})$
**until** $\gamma^{-1}\max\{\|\log\tilde{\boldsymbol{u}}/\boldsymbol{u}\|_\infty, \|\log\tilde{\boldsymbol{v}}/\boldsymbol{v}\|_\infty\} < \delta$
return $\operatorname{diag}(\boldsymbol{u})\boldsymbol{K}\operatorname{diag}(\boldsymbol{v})$

---

# H    Discussion of Complexity

For $(\boldsymbol{Q}, \boldsymbol{R}, \boldsymbol{T}) \in \mathbb{R}_+^{n\times r_1} \times \mathbb{R}_+^{m\times r_2} \times \mathbb{R}_+^{r_1\times r_2}$, the space complexity $O(nr_1 + r_1r_2 + mr_2)$ is linear if the ranks $r_1, r_2 = o(1)$ are taken to be small constants. The time-complexity of Algorithm 4 is $O(BLr^2(n+m))$ for $B$ the number of inner Sinkhorn iterations, $L$ the number of mirror-descent steps, $n, m$ the number of samples in the first and second dataset, and $r = \max\{r_1, r_2, d\}$ for $r_1, r_2$ the ranks of the latent coupling and $d$ the rank of the factorized distance matrix $\boldsymbol{C}$ (generally chosen to be a constant near $r_1, r_2$). Each matrix-multiplication is of max order $(n+m)r^2$, which happens a constant number of times in the computation of each gradient $\nabla_i$, and for the respective Sinkhorn matrix-vector multiplications $\boldsymbol{K}\boldsymbol{v}$ and $\boldsymbol{K}^{\mathrm{T}}\boldsymbol{u}$. The $L$ outer steps follow from the mirror-descent convergence rate and the number of iterations $B$ required for each projection follow from the convergence of Sinkhorn. In particular, for $\varepsilon$ a fixed error tolerance and $\eta$ the entropy constant, one finds a $\pm\varepsilon D$ approximation for $D$ the diameter of the data in $B = \operatorname{poly}(1/\eta\varepsilon)$ iterations using the Sinkhorn algorithm Charikar et al. (2023); Cuturi (2013a).

# I Review of Background Material

## I.1 Low-Rank Approximation of Pairwise Distance Matrices

As mentioned previously, works such as Charikar et al. (2023) have developed algorithms with linear $O((n+m)^{1+o(1)}\text{poly}(1/\epsilon))$ time-complexity and $O((n+m)d)$ space-complexity for sketching the optimal transport cost *value*. Recent works on low-rank factorization of the optimal transport coupling matrix $\boldsymbol{P}$ (the matrix associated to the coupling $\gamma \in \Pi(\mu, \nu)$) Scetbon & Cuturi (2022); Scetbon et al. (2023, 2021) have achieved per-iteration time-complexities of $O(T(n+m)dr)$ for some constant non-negative rank $r \geq 1$, $d$ the dimension of the metric space, and $T$ the number of iterations. By the JL-lemma one can simply embed the points in dimension $d = O(\log(nm)/\epsilon^2)$ while preserving pairwise distances, however, currently no proofs exist which offer the number of iterations $T$ until convergence to some tolerance $\varepsilon$. This is partially due to how recent these works are, and also to the non-convexity of the objective which is sensitive to initial conditions Scetbon et al. (2021). However, the space complexity of the algorithm is $O((n+m)dr)$, which is noteworthy for being *linear* in the number of points and avoids storing the potentially intractable $O(nm)$ coupling matrix $\boldsymbol{P}$. To accomplish this, however, these works rely on a low-rank approximation of the pairwise distance matrix. A number of works by Indyk and Woodruff have concerned algorithms for finding low-rank approximations for such distance matrices. A seminal work Bakshi & Woodruff (2018) developed an algorithm which, given two point sets $\{\boldsymbol{x}_i\}_{i=1}^n$ and $\{\boldsymbol{y}_j\}_{j=1}^m$ in some metric space $\mathcal{X}$, finds a rank $r$ approximation in $O((n+m)^{1+\gamma}\text{poly}(r, 1/\varepsilon))$ for $\gamma > 0$ an arbitrarily small constant and $\varepsilon > 0$ an error parameter. A more recent work Indyk et al. (2019) improves on this one, by reading a sample-optimal $O((n+m)r/\varepsilon)$ entries of the input matrix with a run-time which removes dependence on $\gamma$ that is merely $O((n+m)\text{poly}(r, 1/\varepsilon))$. This algorithm is used by all of the low-rank optimal transport works. These works, by finding a low-rank approximation to the coupling matrix $\boldsymbol{P} \approx \boldsymbol{A}\boldsymbol{B}^{\mathrm{T}} \in \mathbb{R}_+^{n \times m}$ due to space limitations on the coupling, necessarily cannot store the full distance matrix $\boldsymbol{C} \in \mathbb{R}_+^{n \times m}$ of the same size in memory either. As such, it must also be approximated as $\boldsymbol{C} \approx \boldsymbol{V}\boldsymbol{U}$ before input to the algorithm, where we necessarily require very effective approximations of $\boldsymbol{U}$ and $\boldsymbol{V}$ to tolerate the additional source of error from coarse-graining the distance matrix to be low-rank. As such, we present some of the details and algorithm of Indyk et al. (2019) as an essential component of the existing low-rank optimal transport solvers. We begin by summarizing the main theorems in Indyk et al. (2019), which provide an algorithm (upper-bound) on the low-rank distance-matrix approximation problem and a lower-bound on the number of entries which must be read.

**Theorem I.1.** *Indyk et al. (2019) There is a randomized algorithm that, given a distance matrix $\boldsymbol{C} \in \mathbb{R}^{n \times m}$, reads $O((n+m)r/\varepsilon)$ entries of $\boldsymbol{C}$, runs in time $\tilde{O}(n+m) \cdot \text{poly}(r, 1/\varepsilon)^1$ and computes low-rank factors $\boldsymbol{V} \in \mathbb{R}^{n \times r}$, $\boldsymbol{U} \in \mathbb{R}^{r \times m}$ that with probability $0.99$ satisfy:*

$$\|\boldsymbol{C} - \boldsymbol{V}\boldsymbol{U}\|_F^2 \leq \|\boldsymbol{C} - \boldsymbol{C}_r\|_F^2 + \varepsilon\|\boldsymbol{C}\|_F^2 \tag{42}$$

*For $\boldsymbol{C}_r$ the optimal rank-$r$ approximation of $\boldsymbol{C}$.*

This is a remarkable result, especially in light of the next theorem.

**Theorem I.2.** *Indyk et al. (2019) Let $r \leq m \leq n$ and $\varepsilon > 0$ such that $r/\varepsilon = O(\min\{m, n^{1/3}\})$. Any randomized and possibly adaptive algorithm that given a distance matrix $\boldsymbol{C} \in \mathbb{R}^{n \times m}$ computes $\boldsymbol{V} \in \mathbb{R}^{n \times r}$, $\boldsymbol{U} \in \mathbb{R}^{r \times m}$ satisfying $\|\boldsymbol{C} - \boldsymbol{V}\boldsymbol{U}\|_F^2 \leq \|\boldsymbol{C} - \boldsymbol{C}_r\|_F^2 + \varepsilon\|\boldsymbol{C}\|_F^2$ must read $\Omega((n+m)r/\varepsilon)$ entries of $\boldsymbol{C}$ in expectation. This lower bound also holds for symmetric distance matrices $\boldsymbol{C} \in \mathcal{S}_n$.*

The lower-bound follows from a difficult argument which involves constructing a hard distribution over distance matrices, involving the use of random matrix theory. The upper-bound, however, follows relatively straightforwardly from a number of previous algorithms and their associated guarantees, along with the algorithm presented below. We introduce the algorithm and also offer a proof of I.1 for completeness.

---

[1] Where $\tilde{O}(\cdot)$ hides poly-log factors.

**Algorithm 8**   Low-Rank approximation for distance matrix $\boldsymbol{C}$

---

Input point sets $\{\boldsymbol{x}_i\}_{i=1}^n$, $\{\boldsymbol{y}_j\}_{j=1}^M$ in metric space $\mathcal{X}$ and metric $d$
Pick indices $i^* \in [n], j^* \in [m]$ uniformly at random
**for** $i = 1$ to $n$ **do**
    Update sample probability $p_i = d(\boldsymbol{x}_i, \boldsymbol{y}_{j^*})^2 + d(\boldsymbol{x}_{i^*}, \boldsymbol{y}_{j^*})^2 + \frac{1}{m} \sum_{j=1}^m d(\boldsymbol{x}_{i^*}, \boldsymbol{y}_j)^2$
**end for**
Sample $O(r/\varepsilon)$ rows $\boldsymbol{C}_{i,\cdot} \sim Categorical\left(\frac{p_i}{\sum_i p_i}\right)$
Compute $\boldsymbol{U}$ using Frieze et al. (2004)
Compute $\boldsymbol{V}$ using Chen & Price (2017)
return $\boldsymbol{V}, \boldsymbol{U}$

---

This algorithm relies on two previous works, whose main results we summarize here.

**Theorem I.3.** *Frieze et al. (2004) Let $\boldsymbol{C} \in \mathbb{R}^{n \times m}$. For a sample of $O(r/\varepsilon)$ rows according to a distribution $\boldsymbol{p} \in \Delta_n$ which satisfies $p_i \geq \Omega(1)\|\boldsymbol{C}_{i,\cdot}\|_2^2/\|\boldsymbol{C}\|_F^2$ for $i \in [n]$. Then in $O(mr/\epsilon + poly(r, 1/\varepsilon))$ time one may compute a matrix $\boldsymbol{U} \in \mathbb{R}^{r \times m}$ from this sample which satisfies:*

$$\|\boldsymbol{C} - \boldsymbol{C}\boldsymbol{U}^{\mathrm{T}}\boldsymbol{U}\|_F^2 \leq \|\boldsymbol{C} - \boldsymbol{C}_k\|_F^2 + \varepsilon\|\boldsymbol{C}\|_F^2$$

*With probability 0.99.*

Thus, to compute the first low-rank factor $\boldsymbol{U}$, we need to ensure the $p_i$ generated from the algorithm satisfies this requirement and offer the (short) proof below.

*Proof.* First, it is helpful to note $d(x, y)^2 \leq (d(x, z) + d(z, y))^2 = d(x, z)^2 + 2d(x, z)d(y, z) + d(y, z)^2 \leq 2(d(x, z)^2 + d(y, z)^2)$. Where in the last step one uses AM-GM where $\prod_i a_i^{1/n} \leq \frac{\sum_i a_i}{n}$. Rewriting the norm of row $i$ we have:

$$
\begin{aligned}
\|\boldsymbol{C}_{i,\cdot}\|_2^2 &= \sum_{j=1}^m d(\boldsymbol{x}_i, \boldsymbol{y}_j)^2 \leq 2\sum_{j=1}^m d(\boldsymbol{x}_i, \boldsymbol{y}_{j^*})^2 + d(\boldsymbol{y}_{j^*}, \boldsymbol{y}_j)^2 \\
&= 2md(\boldsymbol{x}_i, \boldsymbol{y}_{j^*})^2 + 2\sum_{j=1}^m d(\boldsymbol{y}_{j^*}, \boldsymbol{y}_j)^2 \\
&\leq 2md(\boldsymbol{x}_i, \boldsymbol{y}_{j^*})^2 + 4\sum_{j=1}^m d(\boldsymbol{y}_{j^*}, \boldsymbol{x}_{i^*})^2 + d(\boldsymbol{y}_j, \boldsymbol{x}_{i^*})^2 \\
&= 2md(\boldsymbol{x}_i, \boldsymbol{y}_{j^*})^2 + 4md(\boldsymbol{y}_{j^*}, \boldsymbol{x}_{i^*})^2 + 4\sum_{j=1}^m d(\boldsymbol{y}_j, \boldsymbol{x}_{i^*})^2 \\
&= 4m\left(\frac{1}{2}d(\boldsymbol{x}_i, \boldsymbol{y}_{j^*})^2 + d(\boldsymbol{y}_{j^*}, \boldsymbol{x}_{i^*})^2 + \frac{1}{m}\sum_{j=1}^m d(\boldsymbol{y}_j, \boldsymbol{x}_{i^*})^2\right) \leq 4mp_i
\end{aligned}
$$

As we have the re-normalization $p_i \leftarrow \frac{p_i}{\sum_i p_i}$ before sampling, we need to consider the value of the expectation $\mathbb{E}[\sum_i p_i]$ to conclude.

$$\mathbb{E}\left[\sum_{i=1}^n p_i\right] = \sum_{i=1}^n \mathbb{E}\left[d(\boldsymbol{x}_i, \boldsymbol{y}_{j^*})^2 + d(\boldsymbol{x}_{i^*}, \boldsymbol{y}_{j^*})^2 + \frac{1}{m}\sum_{j=1}^m d(\boldsymbol{x}_{i^*}, \boldsymbol{y}_j)^2\right]$$

$$= \sum_{i=1}^n \mathbb{E}_{j^*\sim[m]}\left[d(\boldsymbol{x}_i, \boldsymbol{y}_{j^*})^2\right] + \mathbb{E}_{i^*\sim[n],j^*\sim[m]}\left[d(\boldsymbol{x}_{i^*}, \boldsymbol{y}_{j^*})^2\right]$$

$$+ \frac{1}{m}\sum_{j=1}^m \mathbb{E}_{i^*\sim[n]}\left[d(\boldsymbol{x}_{i^*}, \boldsymbol{y}_j)^2\right]$$

$$= \sum_{i=1}^n \frac{1}{m}\sum_{j=1}^m d(\boldsymbol{x}_i, \boldsymbol{y}_j) + n\sum_{i=1}^n\sum_{j=1}^m \frac{1}{nm}d(\boldsymbol{x}_i, \boldsymbol{y}_j) + \frac{n}{m}\sum_{j=1}^m \frac{1}{n}\sum_{i=1}^n d(\boldsymbol{x}_i, \boldsymbol{y}_j)$$

$$= \frac{3}{m}\sum_{i=1}^n\sum_{j=1}^m d(\boldsymbol{x}_i, \boldsymbol{y}_j) = \frac{3}{m}\|\boldsymbol{C}\|_F^2$$

By an application of Markov's inequality we have that:

$$P\left[\left(\sum_{i=1}^n p_i\right)^{-1} \geq \left(\frac{3\|\boldsymbol{C}\|_F^2}{m\delta}\right)^{-1}\right] = P\left[\sum_{i=1}^n p_i \leq \frac{3\|\boldsymbol{C}\|_F^2}{m\delta}\right] \geq 1 - \frac{\mathbb{E}[\sum_{i=1}^n p_i]}{\frac{3\|\boldsymbol{C}\|_F^2}{m\delta}} = 1 - \delta$$

Thus with probability $1 - \delta$ we have:

$$\frac{p_i}{\sum_{i'=1}^n p_{i'}} \geq \frac{mp_i\delta}{3\|\boldsymbol{C}\|_F^2} = \frac{4mp_i\delta}{12\|\boldsymbol{C}\|_F^2} \geq \frac{\|\boldsymbol{C}_{i,.}\|_2^2\delta}{12\|\boldsymbol{C}\|_F^2} = \Omega(\delta)\frac{\|\boldsymbol{C}_{i,.}\|_2^2}{\|\boldsymbol{C}\|_F^2}$$

This indicates the algorithm presented has probabilities with an appropriate bound for using the algorithm of Frieze et al. (2004) to sample the $O(r/\varepsilon)$ rows of $\boldsymbol{C}$ and generate a rank-r factor $\boldsymbol{U}$. □

To conclude the result of Indyk et al. (2019) requires reference to an additional work which solves a regression problem for $\boldsymbol{V}$ given $\boldsymbol{C}$ and $\boldsymbol{U}$.

**Theorem I.4.** *Chen & Price (2017) There is a randomized algorithm $\mathcal{A}$, given matrices $\boldsymbol{C} \in \mathbb{R}^{n\times m}$, $\boldsymbol{U} \in \mathbb{R}^{r\times m}$ reads only $O(r/\varepsilon)$ columns of $\boldsymbol{C}$ with time-complexity $O(mr) + poly(r/\varepsilon)$ and returns $\boldsymbol{V} \in \mathbb{R}^{n\times r}$ which satisfies*

$$\|\boldsymbol{C} - \boldsymbol{V}\boldsymbol{U}\|_F^2 \leq (1+\varepsilon)\min_{\boldsymbol{Z}\in\mathbb{R}^{n\times r}}\|\boldsymbol{C} - \boldsymbol{Z}\boldsymbol{U}\|_F^2$$

*with probability 0.99.*

Thus, using the result of Chen and Price Chen & Price (2017), one may find a satisfying $\boldsymbol{V}$ easily for a fixed $\boldsymbol{U}, \boldsymbol{C}$. In particular, Indyk et al. (2019) concludes by tying together the low-rank distance matrix algorithm and the guarantees of the algorithms from Frieze et al. (2004) Chen & Price (2017) as follows

$$\|\boldsymbol{C} - \boldsymbol{V}\boldsymbol{U}\|_F^2 \leq (1+\varepsilon)\min_{\boldsymbol{Z}}\|\boldsymbol{C} - \boldsymbol{Z}\boldsymbol{U}\|_F^2$$

$$\leq (1+\varepsilon)\|\boldsymbol{C} - \boldsymbol{C}\boldsymbol{U}^{\mathrm{T}}\boldsymbol{U}\|_F^2$$

$$\leq (1+\varepsilon)(\|\boldsymbol{C} - \boldsymbol{C}_r\|_F^2 + \varepsilon\|\boldsymbol{C}\|_F^2)$$

$$= \|\boldsymbol{C} - \boldsymbol{C}_r\|_F^2 + \varepsilon\|\boldsymbol{C} - \boldsymbol{C}_r\|_F^2 + (1+\varepsilon)\varepsilon\|\boldsymbol{C}\|_F^2$$

$$\leq \|\boldsymbol{C} - \boldsymbol{C}_r\|_F^2 + (1+2\varepsilon)\varepsilon\|\boldsymbol{C}\|_F^2 \leq \|\boldsymbol{C} - \boldsymbol{C}_r\|_F^2 + \tilde{\varepsilon}\|\boldsymbol{C}\|_F^2$$

Which achieves the bound up to a constant scaling of $\varepsilon$ and shows the result of Indyk et al. (2019). We next investigate low-rank optimal transport solvers, which assume the result and algorithm of Indyk et al. (2019) to tractably scale to massive datasets.

# J  Connection of Optimal Transport to Projection Problems

Before discussing works which have address the problem of finding low-rank decompositions of the coupling matrix $\boldsymbol{P}$, we discuss a few relevant preliminaries. The Bregman-divergence of some function $F(\boldsymbol{x})$, defined by the first-order Taylor expansion of $F$:

$$D_F(\boldsymbol{x}, \boldsymbol{y}) = F(\boldsymbol{x}) - F(\boldsymbol{y}) + \nabla F(\boldsymbol{y})^{\mathrm{T}} (\boldsymbol{x} - \boldsymbol{y})$$

For the negative entropy function $-\mathrm{H}(\boldsymbol{\xi})$, this corresponds to the KL-divergence $\mathrm{KL}(\boldsymbol{\zeta} \mid \boldsymbol{\xi})$. We introduce the iterative-Bregman projection algorithm in the context of the KL-divergence owing to the direct connection with entropically-regularized optimal transport.

**Definition J.1.** Iterative Bregman ProjectionsBregman (1967)

Suppose $\mathcal{C} = \cap_{l=1}^{L} \mathcal{C}_l$ is an intersection of closed convex sets $\{\mathcal{C}_l\}_{l=1}^{L}$. For $n > L$, let the indexing be $L$-periodic as $\mathcal{C}_n := \mathcal{C}_{n \bmod L}$. Suppose we want to find a minimizer $\boldsymbol{\zeta}$ of the KL-divergence with some positive vector $\boldsymbol{\xi} \in \mathbb{R}_+^{n \times m}$ such that $\boldsymbol{\zeta} \in \mathcal{C}$. This is to say, we hope to solve the problem of minimizing a distance subject to the condition that one remains in this intersection of convex sets:

$$\min_{\boldsymbol{\zeta} \in \mathcal{C}} \mathrm{KL}(\boldsymbol{\zeta} \| \boldsymbol{\xi})$$

Where the projection of $\boldsymbol{\xi}$ onto the set $\mathcal{C}$ is denoted by

$$\boldsymbol{\zeta}^* = \arg\min_{\boldsymbol{\zeta} \in \mathcal{C}} \mathrm{KL}(\boldsymbol{\zeta} \| \boldsymbol{\xi}) := \mathcal{P}_{\mathcal{C}}^{KL}(\boldsymbol{\xi})$$

Supposing that each $\mathcal{C}_l$ forms a quotient space $\mathcal{C}_l = V/U$ for a subspace $U \subset V$, defined as $V/U = \{\boldsymbol{v} + U \mid \boldsymbol{v} \in V\}$[2], the iterative Bregman projection algorithm alternates projections onto each set $\mathcal{C}_n$ as

$$\boldsymbol{\zeta}^{(n)} \leftarrow \mathcal{P}_{\mathcal{C}_n}^{KL}(\boldsymbol{\zeta}^{(n-1)}) \tag{43}$$

starting from $\boldsymbol{\zeta}^{(0)} = \boldsymbol{\xi}$.

One may show Bregman (1967) the convergence of Bregman projections to the unique minimizer in $\mathcal{C}, \boldsymbol{\zeta}^*$, where we have the guarantee that $\boldsymbol{\zeta}^{(n)} \to \boldsymbol{\zeta}^*$ as $n \to \infty$. These iterative projections only have convergence guarantees when the constraint sets are quotient spaces–this is clearly not the case for the constraints of the optimal transport LP, and a few more notions are required.

**Definition J.2.** Dykstra's AlgorithmDykstra (1983)

Given a point $\boldsymbol{x}_0 \in E$ for $E$ a Euclidean space [3], to find the unique point in $\mathcal{C} = \cap_{l=1}^{L} \mathcal{C}_l$ for closed, convex sets $\mathcal{C}_l$ that minimize the distance to $\boldsymbol{x}_0$ as

$$\boldsymbol{x}^* = \arg\min_{\boldsymbol{x} \in \mathcal{C}} \|\boldsymbol{x} - \boldsymbol{x}_0\|_2$$

One may initialize the residuals $\boldsymbol{q}_{-(L-1)} = \ldots = \boldsymbol{q}_0 = 0$ and apply the algorithm:

$$\boldsymbol{x}_n = \mathcal{P}_{\mathcal{C}_n}(\boldsymbol{x}_{n-1} + \boldsymbol{q}_{n-1})$$
$$\boldsymbol{q}_n = (\boldsymbol{x}_{n-1} - \boldsymbol{x}_n) + \boldsymbol{q}_{n-L}$$

Where $\mathcal{C}_n := \mathcal{C}_{n \bmod L}$ as before and $\mathcal{P}_{\mathcal{C}_n}$ denotes the projection operator onto the convex set $\mathcal{C}_n$.

One can note that Dykstra's algorithm for projections onto intersections of convex sets no longer relies on the assumption that the set is a quotient space, and applies in the case that it is merely closed under convex-combinations. To generalize Dykstra's for more general functions that than the $\ell_2$-norm, one may define the projection with respect to the Bregman divergence of a cost function $F$ and define the projection by the minimization: $\mathcal{P}_{\mathcal{C}_n} := \arg\min_{\boldsymbol{x}} D_F(\boldsymbol{x}, \boldsymbol{y})$. It was proven in Bauschke & Lewis (2000) that the generalized form of Dykstra's iterations are given as:

$$\boldsymbol{x}_n = \mathcal{P}_{\mathcal{C}_n} (\nabla F^*(\nabla F(\boldsymbol{x}_{n-1}) + \boldsymbol{q}_{n-1}))$$
$$\boldsymbol{q}_n = (\nabla F(\boldsymbol{x}_{n-1}) - \nabla F(\boldsymbol{x}_n)) + \boldsymbol{q}_{n-L}$$

For $F^*$ denoting the Fenchel-conjugate of $F$, which we define later in connection to the low-rank dual problem. Notably, Bauschke & Lewis (2000) also provided guarantees of convergence to the optimal solution which extend to the case that $F$ is the negative entropy and $D_F$ the KL-divergence. These constitute Dykstra's algorithm with cyclic Bregman projections, and project a point to the closest point in the intersection of convex sets $\mathcal{C} = \cap_{l=1}^{L} \mathcal{C}_l$ for an arbitrary cost function $F$ and its associated Bregman-divergence $D_F$.

---

[2]This is to say, $\boldsymbol{x}, \boldsymbol{y} \in \mathcal{C}_l \implies c_1 \boldsymbol{x} + c_2 \boldsymbol{y} \in V/U$ for any $c_1, c_2 \in \mathbb{R}$.
[3]e.g. $\mathbb{R}, \mathbb{R}^d, \mathbb{R}^{n \times m}$, etc.

## J.1 Dykstra's algorithm with cyclic Bregman projections

As such, we can see the updates for $F$ being the negative entropy and $D_F$ the KL-divergence. Without proof, the conjugate of the negative entropy is simply given as $F^*(\boldsymbol{\zeta}) = \exp\{\boldsymbol{\zeta} - \mathbf{1}\}$. Thus, for the minimization problem:

$$\boldsymbol{\zeta}^* = \arg\min_{\boldsymbol{\zeta} \in \mathcal{C}} \mathrm{KL}(\boldsymbol{\zeta}\|\boldsymbol{\xi}) := \mathcal{P}_{\mathcal{C}}^{\mathrm{KL}}(\boldsymbol{\xi})$$

Letting $\log \boldsymbol{q}_{-(L-1)} = ... = \log \boldsymbol{q}_0 = 0$, one may combine the two algorithms above to solve this minimization using generalized Dykstra's iterations. In particular, we have:

$$
\begin{aligned}
\boldsymbol{\zeta}^{(n)} &= \mathcal{P}_{\mathcal{C}_n}\left(\nabla F^*(\nabla F(\boldsymbol{\zeta}^{(n-1)}) + \log \boldsymbol{q}_{n-1})\right) \\
&= \mathcal{P}_{\mathcal{C}_n}\left(\exp\left(\nabla F(\boldsymbol{\zeta}^{(n-1)}) + \log \boldsymbol{q}_{n-1} - \mathbf{1}\right)\right) \\
&= \mathcal{P}_{\mathcal{C}_n}\left(\exp\left(\log \boldsymbol{\zeta}^{(n-1)} + \mathbf{1} + \log \boldsymbol{q}_{n-1} - \mathbf{1}\right)\right) \\
&= \mathcal{P}_{\mathcal{C}_n}\left(\boldsymbol{\zeta}^{(n-1)} \odot \boldsymbol{q}_{n-1}\right)
\end{aligned}
$$

And:

$$
\begin{aligned}
\log \boldsymbol{q}_n &= (\nabla F(\boldsymbol{\zeta}^{(n-1)}) - \nabla F(\boldsymbol{\zeta}^{(n)})) + \log \boldsymbol{q}_{n-N} \\
&= \left(\log \boldsymbol{\zeta}^{(n-1)} + \mathbf{1} - (\log \boldsymbol{\zeta}^{(n)} + \mathbf{1}) + \log \boldsymbol{q}_{n-L}\right) \\
&= \log\left(\boldsymbol{q}_{n-L} \odot \frac{\boldsymbol{\zeta}^{(n-1)}}{\boldsymbol{\zeta}^{(n)}}\right)
\end{aligned}
$$

So that:

$$\boldsymbol{\zeta}^{(n)} \leftarrow \mathcal{P}_{\mathcal{C}_n}\left(\boldsymbol{\zeta}^{(n-1)} \odot \boldsymbol{q}_{n-1}\right) \tag{44}$$

$$\boldsymbol{q}_n \leftarrow \boldsymbol{q}_{n-L} \odot \frac{\boldsymbol{\zeta}^{(n-1)}}{\boldsymbol{\zeta}^{(n)}} \tag{45}$$

Where division is interpreted to be element-wise, $\odot$ refers to the Hadamard product, and the logarithm is applied elementwise.

## J.2 Connection to Sinkhorn distances

Interestingly, the Sinkhorn algorithm described in Algorithm 5 can be alternatively derived in the context of Bregman iterations Benamou et al. (2015) as a minimization of the form:

$$\mathrm{W}_\epsilon(\mu, \nu) = \epsilon \min_{\boldsymbol{P} \in \Pi(\mu,\nu)} \mathrm{KL}(\boldsymbol{P}\|\boldsymbol{\xi})$$

Where $\boldsymbol{\xi}$ is the kernel $\boldsymbol{\xi} = e^{-\boldsymbol{C}/\epsilon}$ and $\Pi(\mu,\nu) = \mathcal{C}_1 \cap \mathcal{C}_2$ for the convex constraint sets $\mathcal{C}_1 = \{\boldsymbol{P} : \boldsymbol{P}\mathbf{1}_m = \boldsymbol{a}\}$ and $\mathcal{C}_2 = \{\boldsymbol{P} : \boldsymbol{P}^{\mathrm{T}}\mathbf{1}_n = \boldsymbol{b}\}$. To cast this into the Bregman-projection framework, one alternates between the two updates:

$$
\begin{aligned}
\boldsymbol{P}^{(l)} &= \mathcal{P}_{\mathcal{C}_1}(\boldsymbol{P}^{(l-1)}) \\
\boldsymbol{P}^{(l+1)} &= \mathcal{P}_{\mathcal{C}_2}(\boldsymbol{P}^{(l)})
\end{aligned}
$$

Where for the first projection one has the following first-order KKT condition:

$$
\begin{aligned}
\nabla\left(\epsilon\mathrm{KL}(\boldsymbol{P}\|\boldsymbol{P}^{(l-1)}) + \boldsymbol{\lambda}^{\mathrm{T}}(\boldsymbol{P}\mathbf{1}_m - \boldsymbol{a})\right) &= \epsilon\log\left(\frac{\boldsymbol{P}}{\boldsymbol{P}^{(l-1)}}\right) + \boldsymbol{\lambda}\mathbf{1}_m^{\mathrm{T}} = \mathbf{0} \\
&\implies \boldsymbol{P} = \mathrm{diag}(e^{-\boldsymbol{\lambda}/\epsilon})\boldsymbol{P}^{(l-1)}
\end{aligned}
$$

With the constraint of $\mathcal{C}_1$ that $\boldsymbol{P}\mathbf{1}_m = \boldsymbol{a}$, this implies $\mathrm{diag}(\boldsymbol{a}/\boldsymbol{P}^{(l-1)}\mathbf{1}_m) = \mathrm{diag}(e^{-\boldsymbol{\lambda}/\epsilon})$ and recovers the first update of Sinkhorn $\boldsymbol{P}^{(l)} \leftarrow \mathrm{diag}(\boldsymbol{a}/\boldsymbol{P}^{(l-1)}\mathbf{1}_m)\boldsymbol{P}^{(l-1)}$. An analogous argument gives the second, where all iterates satisfy $\boldsymbol{P}^{(l)} = \mathrm{diag}(\boldsymbol{u}^{(l)})e^{-\boldsymbol{C}/\epsilon}\mathrm{diag}(\boldsymbol{v}^{(l)})$ for $\boldsymbol{u}, \boldsymbol{v}$ as defined in Algorithm 5.

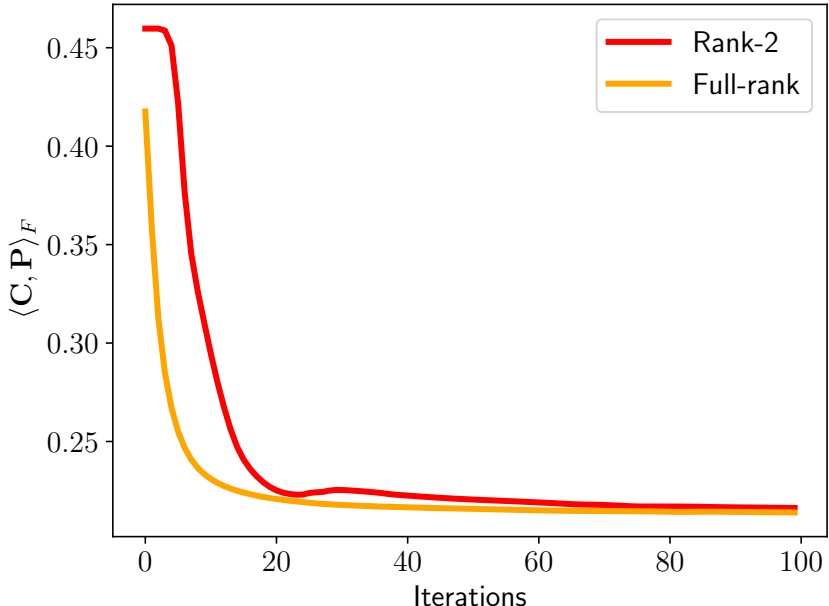

Figure 5: Transport cost $\langle \boldsymbol{C}, \boldsymbol{P} \rangle_F$ against number of iterations for FRLC with rank 200 on the synthetic dataset of two moons and eight Gaussians. Smooth convergence is observed for both rank-2 and full-rank random initialization.

## K    Additional Simulations

We tested on two additional synthetic datasets, both used as benchmarking datasets in Scetbon et al. (2021). We follow exactly the parameters provided in Scetbon et al. (2021) to simulate these datasets. For the first one, we simulated $n = m = 10,000$ points from two Gaussian mixtures in 2D (Fig. 6). The first Gaussian mixture is a mixture of three Gaussian distributions with means $(0,0), (0,1), (1,1)$ respectively. The mixture proportion is $\frac{1}{3}$ and the covariance is $0.05 \times$ identity for each Gaussian. The second Gaussian mixture is a mixture of two Gaussian distributions with means $(0.5, 0.5), (-0.5, 0.5)$ respectively. The mixture proportions is $\frac{1}{2}$ and the covariance is $0.05 \times$ identity for each Gaussian.

For the second dataset, we simulated $n = m = 5,000$ points from two Gaussian mixtures in 10D. The first Gaussian mixture is a mixture of three Gaussian distributions with means $(0,0,0,\cdots,0), (0,1,0,\cdots,0), (1,1,0,\cdots,0)$ respectively. The mixture proportions is $\frac{1}{3}$ and the covariance is $0.05 \times$ identity for each Gaussian. The second Gaussian mixture is a mixture of two Gaussian distributions with means $(0.5, 0.5, 0, \cdots, 0), (-0.5, 0.5, 0, \cdots, 0)$ respectively. The mixture proportions is $\frac{1}{2}$ and the covariance is $0.05 \times$ identity for each Gaussian.

For each dataset, we repeat the same procedure as in § 4.1, running FRLC and LOT with Euclidean distance as cost to find a low-rank coupling matrix between the two Gaussian mixtures with rank between 50 and 200. We observe the same pattern as Fig. 2b. FRLC obtains lower transport cost with increasing rank, and achieves lower cost for each rank than LOT under all initializations (Fig. 23c, Fig. 7).

## L    Graph Partitioning

### L.1    Evaluation on a Graph Partitioning Task

We next evaluate FRLC on an unsupervised graph partitioning (node clustering) problem described by Chowdhury & Needham (2021). Specifically, given a graph $G = (V, E)$ of $n$ nodes, we represent the graph as $G = (\boldsymbol{A}, \boldsymbol{h})$, where $\boldsymbol{A} \in \mathbb{R}^{n \times n}$ encodes the intra-graph node relationship (e.g. adjacency matrix) and $\boldsymbol{h} \in \Delta_n$ is a uniform measure. We cluster the nodes of $G$ by estimating a GW coupling

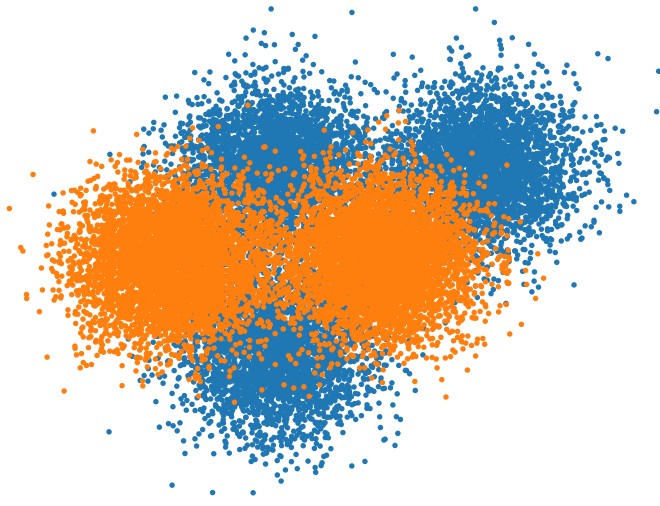

Figure 6: Plot of the two simulated mixtures of Gaussians in 2D, following the same parameters as Scetbon et al. (2021).

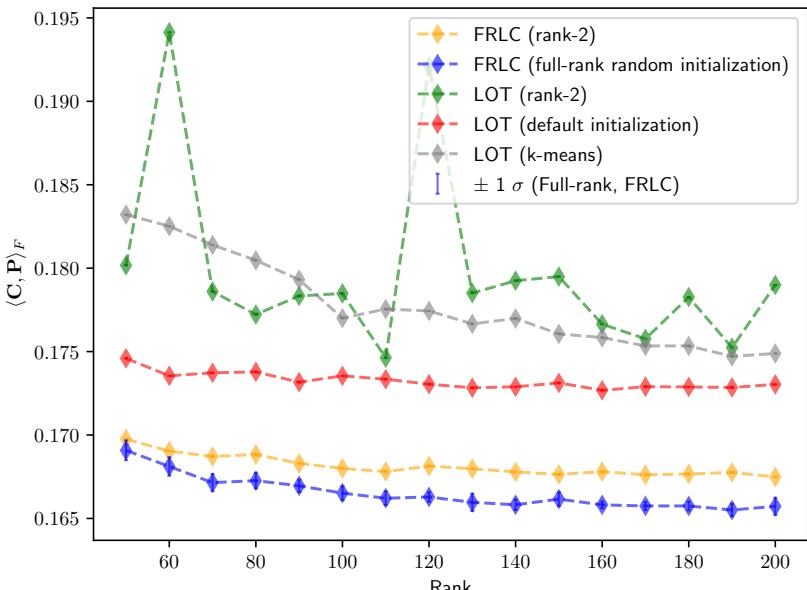

Figure 7: Transport cost $\langle \boldsymbol{C}, \boldsymbol{P} \rangle_F$ achieved by LOT Scetbon et al. (2021) and FRLC across different ranks and different initializations on the Wasserstein problem on the synthetic dataset of two mixtures of Gaussians in 2D.

| Method | Dataset | Time in seconds (CPU) | Cost Value $\langle \boldsymbol{C}, \boldsymbol{P} \rangle_F$ | $\|\boldsymbol{P}\mathbf{1}_n - \boldsymbol{a}\|_2$ | $\|\boldsymbol{P}^T\mathbf{1}_m - \boldsymbol{b}\|_2$ |
|---|---|---|---|---|---|
| FRLC | 2-moons, | **0.379** | **0.207** | 1.56E-5 | 6.1E-18 |
| LOT | 8-Gaussians | 0.751 | 0.210 | 1.90E-5 | 1.89E-5 |
| FRLC | Gaussian mixture | **0.354** | **0.178** | 1.05E-5 | 7.0E-18 |
| LOT | (2D) | 0.735 | 0.181 | 1.65E-5 | 1.70E-5 |
| FRLC | Gaussian mixture | **0.323** | **0.294** | 2.48E-6 | 8.9E-18 |
| LOT | (10D) | 0.677 | 0.307 | 1.39E-5 | 1.46E-5 |

Table 2: Runtime of FRLC and LOT on the synthetic datasets of 1000 samples, as well as cost value $\langle \boldsymbol{C}, \boldsymbol{P} \rangle_F$ and marginal tightness for context. This was done with the FRLC setting `max_inneriters_balanced=1000`, `max_inneriters_relaxed=50`, `min_iter=7` and rank $r = 100$. This time excludes the extra time incurred for `ott-jax` problem setup and includes the setup time for FRLC.

between $G$ and a smaller graph $\overline{G} = (\overline{\boldsymbol{B}}, \overline{\boldsymbol{h}})$, where $\overline{\boldsymbol{B}} \in \mathbb{R}^{m \times m}$ represents the relationship between each of the $m$ clusters and $\overline{\boldsymbol{h}} \in \Delta_m$ is the proportion of nodes of $G$ in each cluster. Without a priori knowledge, $\overline{\boldsymbol{h}}$ is set to be uniform and $\overline{\boldsymbol{B}}$ is set as the identity matrix. Vincent-Cuaz et al. (2022) notes that instead of solving a balanced GW problem, semi-relaxed GW with the right marginal $\overline{\boldsymbol{h}}$ relaxed learns $\overline{\boldsymbol{h}}$ from data and leads to more accurate node clustering.

| Dataset | | GWL | FRLC SR-GW | SpecGWL | SpecFRLC SR-GW |
|---|---|---|---|---|---|
| Wikipedia | sym, raw | 0.314 | 0.387 | 0.372 | **0.444** |
| (1998 nodes, 2700 edges) | sym, noisy | 0.250 | 0.361 | 0.293 | **0.400** |
| | asym, raw | 0.263 | 0.276 | 0.194 | **0.304** |
| | asym, noisy | **0.208** | 0.201 | 0.141 | 0.177 |
| EU-email | sym, raw | 0.434 | **0.464** | 0.009 | 0.040 |
| (1005 nodes, 25571 edges) | sym, noisy | 0.392 | **0.422** | 0.009 | 0.014 |
| | asym, raw | 0.388 | **0.398** | 0.012 | 0.028 |
| | asym, noisy | **0.385** | 0.348 | 0.008 | 0.012 |
| Amazon | raw | 0.322 | 0.338 | **0.505** | 0.479 |
| (1501 nodes, 4626 edges) | noisy | 0.274 | 0.257 | 0.438 | **0.453** |
| Village | raw | 0.531 | **0.710** | 0.553 | 0.579 |
| (1991 nodes, 8423 edges) | noisy | 0.413 | 0.536 | 0.397 | **0.829** |

Table 3: Performance (measured using Adjusted Mutual Information (AMI)) in graph partitioning for *full-rank* OT algorithms GWL, SpecGWL and full-rank semi-relaxed FRLC. The top performing method for each dataset is highlighted in bold.

We benchmark FRLC for semi-relaxed GW on four real-world graphs: a Wikipedia hyperlink network with 15 webpage categories Yang & Leskovec (2012); an email interaction network within a European institute with 42 departments Yin et al. (2017); an Amazon product network with 12 product categories Yang & Leskovec (2012); and a network of interactions between 12 Indian villages Banerjee et al. (2013). We also test on the symmetric and noisy versions of each graph provided by Chowdhury & Needham (2021). We compare with two OT-based methods: (1) GWL Xu et al. (2019), which solves a balanced GW problem between $G$ and $\overline{G}$ with the adjacency matrix of $G$ as the intra-domain cost matrix $\boldsymbol{A}$; (2) SpecGWL Chowdhury & Needham (2021) which uses the heat kernel on the graph Laplacian as $\boldsymbol{A}$. We similarly run our FRLC algorithm using with both the adjacency matrix (denoted FRLC-SR-GW) and heat kernel (denoted SpecFRLC-SR-GW). Since the number of clusters in each dataset is not large, we compute the full-rank coupling matrix in each case.

Overall, FRLC achieves the best clustering performance on 9 out of 12 datasets (Table 3). When using the adjacency matrix, our semi-relaxed algorithm achieves better clustering result than GWL on 9 out of 12 datasets. When using the heat kernel, our semi-relaxed algorithm achieves better

clustering result than SpecGWL on 11 out of 12 datasets. These results show the importance of semi-relaxed OT on real-world problems, as well as the accuracy of FRLC.

## L.2    Problem Statement

As discussed in Vincent-Cuaz et al. (2022); Chowdhury & Needham (2021), it is possible to achieve unsupervised graph partitioning (node clustering) through Gromov-Wasserstein (GW) OT. Given a graph $(V, E)$ of $n$ nodes, we encode it as $G = (\boldsymbol{A}, \boldsymbol{h})$, where $\boldsymbol{A} \in \mathbb{R}^{n \times n}$ encodes the intra-graph node relationship (e.g. adjacency matrix, Laplacian) and $\boldsymbol{h} \in \Delta n$ is a uniform measure over the nodes. If we want to cluster the nodes of $G$ into $m$ clusters, we can define a new graph $\overline{G} = (\overline{\boldsymbol{B}}, \overline{\boldsymbol{h}})$, where $\overline{\boldsymbol{B}} \in \mathbb{R}^{m \times m}$ is a diagonal matrix representing the cluster's connections and $\overline{\boldsymbol{h}}$ is a distribution over clusters estimating the proportion of nodes in $G$ in each cluster. Since we don't know the density of clusters a priori, we can set $\overline{\boldsymbol{h}}$ to be uniform. Usually $\overline{\boldsymbol{B}}$ is set as the identity matrix.

To cluster the nodes in $G$, we can solve a GW problem matching nodes in $G$ with nodes in $\overline{G}$, with intra-domain cost matrices $\boldsymbol{A}$ and $\overline{\boldsymbol{B}}$:

$$
\begin{aligned}
\min \quad & \sum_{ij'kl'} (\boldsymbol{A}_{ik} - \overline{\boldsymbol{B}}_{j'l'})^2 \boldsymbol{P}_{ij'} \boldsymbol{P}_{kl'} \\
s.t. \quad & \boldsymbol{P} \in \Pi_{\boldsymbol{h}, \overline{\boldsymbol{h}}}
\end{aligned}
\tag{46}
$$

The cluster assignment of each node in $G$ can then be recovered from $\boldsymbol{P}$ by finding the node in $\overline{G}$ mapped to it with the maximum weight.

However, since the proportion of nodes in $G$ in each cluster is not known a priori, solving a balanced GW problem fixing the marginal of $\boldsymbol{P}$ on $\overline{G}$ to be a uniform $\overline{\boldsymbol{h}}$ significantly constrains the expressivity of the algorithm. Therefore, as proposed in Vincent-Cuaz et al. (2022), we can instead solve a semi-relaxed GW problem with the right marginal relaxed, fixing the marginal on $G$ to be $\boldsymbol{h}$ but allowing the marginal on $\overline{G}$ to deviate from $\overline{\boldsymbol{h}}$:

$$
\begin{aligned}
\min \quad & \sum_{ij'kl'} (\boldsymbol{A}_{ik} - \overline{\boldsymbol{B}}_{j'l'})^2 \boldsymbol{P}_{ij'} \boldsymbol{P}_{kl'} + \tau \mathrm{KL}(\boldsymbol{P}^{\mathrm{T}} \mathbf{1}_n \mid \overline{\boldsymbol{h}}) \\
s.t. \quad & \boldsymbol{P} \in \Pi_{\boldsymbol{h}, \cdot}
\end{aligned}
\tag{47}
$$

The learned $\overline{\boldsymbol{h}}$ from semi-relaxed GW estimates the posterior proportion of nodes in $G$ in each of the $m$ clusters.

## L.3    Dataset and Preprocessing

We run our semi-relaxed FRLC algorithm on four real-world graph datasets: a Wikipedia hyperlink network with 1998 nodes and 15 clusters Yang & Leskovec (2012), a directed graph of email interactions in a European research institute with 1005 nodes and 42 clusters Yin et al. (2017), an Amazon product network with 1501 nodes and 12 clusters Yang & Leskovec (2012), and a network of interactions between Indian villages with 1991 nodes and 12 clusters Banerjee et al. (2013). The Wikipedia and EU-email graphs are directed, so we also use undirected versions of them. We also use noisy version of each graph by adding up to 10% additional edges following Chowdhury & Needham (2021), leading to a total of 12 different graphs to cluster.

## L.4    Experiment Settings

We compare our algorithm with two baseline methods, GWL and SpecGWL. GWL Xu et al. (2019) solves the entropy-regularized version of the balanced GW problem of (46) with the adjacency matrix of $G$ as $\boldsymbol{A}$. We set $\boldsymbol{h}$ such that the density of each node is proportional to its degree. We set $\overline{\boldsymbol{h}}$ to be a distribution estimated by sorting the weights of $\boldsymbol{h}$ and sampling $m$ values via linear interpolation, following Chowdhury & Needham (2021). We set $\overline{\boldsymbol{B}} = diag(\overline{\boldsymbol{h}})$. We set the entropy regularization

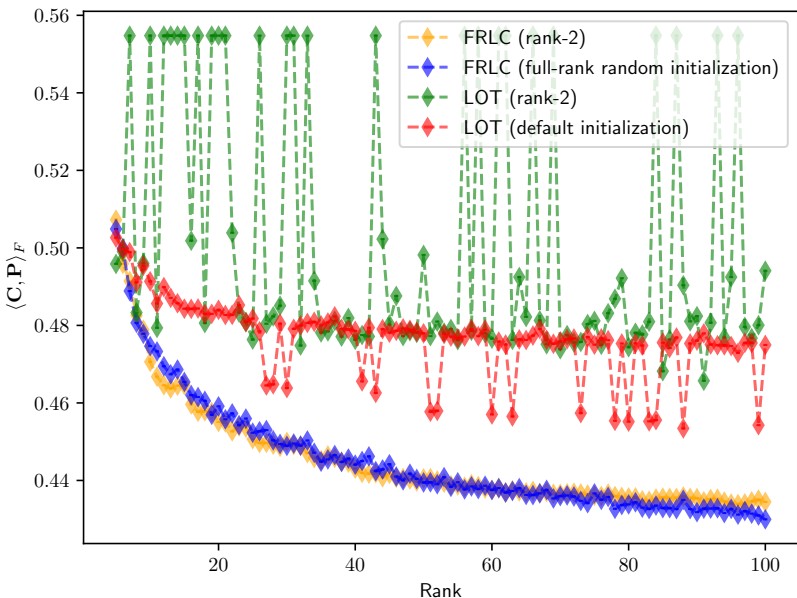

Figure 8: FRLC achieves lower primal cost $\langle \boldsymbol{C}, \boldsymbol{P} \rangle_F$ for $\boldsymbol{P} \in \Pi_{\boldsymbol{a},\boldsymbol{b}}$ than Scetbon et al. (2021) on a spatial-transcriptomics dataset of mouse embryonic development Chen et al. (2022). FRLC demonstrates a more robust trend of improved performance with higher rank.

$\eta = 10^{-6}$. SpecGWL Chowdhury & Needham (2021) solves the same problem as GWL, but instead of using the adjacency matrix as $\boldsymbol{A}$, it uses the heat kernel on the normalized graph Laplacians Chung (2005). We set the heat parameter $t = 10$.

For our method, we solve the semi-relaxed GW problem of (47) with full rank solution. We use both adjacency matrix and heat kernel as $\boldsymbol{C}$ and label the result of the two representations as FRLC-SR-GW and SpecFRLC-SR-GW. We set $\tau = 0.01$ as to minimize the conformation to the right marginal. Since our method depends on random initialization, we run our method 10 times on each dataset and report the mean performance. We evaluate the resulting clusterings of all methods by computing the Adjusted Mutual Information (AMI) between the computed clustering and the ground truth clustering. This experiment, and the experiments on mouse embryo spatial transcriptomics, were conducted on cluster GPUs.

## M  Spatial Transcriptomics Alignment

### M.1  Problem Statement

In this problem, we use FRLC to find a low-rank alignment matrix between cells from two spatial transcriptomics (ST) Ståhl et al. (2016) slices, collected at two timepoints, then use the computed alignment matrix for two downstream prediction tasks. An ST experiment on a 2D tissue slice yields a pair $(X, Z)$. $X \in \mathbb{R}^{n \times p}$ is the gene expression matrix, where $n$ is the number of cells on the slice and $p$ is the number of genes measured. $X_{ij} \in \mathbb{R}$ is the gene expression level of gene $j$ in cell $i$, where a higher number indicates stronger expression. $Z \in \mathbb{R}^{n \times 2}$ is the spatial coordinate matrix, where each row $i$ stores the x-y coordinate of cell $i$ on the slice. Therefore, each cell on the slice has a gene expression vector of length $p$, which encodes the feature of the cell, as well as a coordinate vector of length two, which encodes the geometrical information of the cell on the slice.

Our input data is a pair of ST slices $\mathcal{S}_0 = (X^0, Z^0), \mathcal{S}_1 = (X^1, Z^1)$, with $n$ and $m$ cells, of the same tissue region. We assume $\mathcal{S}_0$ is collected at timepoint $t = 0$ and $\mathcal{S}_1$ is collected at timepoint $t = 1$, hence the transition from $\mathcal{S}_0$ to $\mathcal{S}_1$ reflects the biological development of the tissue during the time period. We would like to find the ancestor-descendant relationship between cells from $\mathcal{S}_0$ and $\mathcal{S}_1$ by computing an optimal transport coupling matrix between cells from $\mathcal{S}_0$ and cells from $\mathcal{S}_1$. The state-of-the-art spatial transcriptomics alignment method moscot Klein et al. (2023) claims that

unbalanced OT is the most appropriate setup for this problem. Specifically, given discrete uniform measure $\boldsymbol{a}$ and $\boldsymbol{b}$ over cells from $\mathcal{S}_0$ and $\mathcal{S}_1$, we solve the unbalanced Wasserstein problem

$$\min_{\boldsymbol{P} \in \mathbb{R}_+^{n \times m}} \langle \boldsymbol{C}, \boldsymbol{P} \rangle + \tau \mathrm{KL}(\boldsymbol{P}\mathbf{1}_m \| \boldsymbol{a}) + \tau \mathrm{KL}(\boldsymbol{P}^{\mathrm{T}}\mathbf{1}_n \| \boldsymbol{b}) \tag{48}$$

$\boldsymbol{C} \in \mathbb{R}_+^{n \times m}$ has entries $\boldsymbol{C}_{ij} = c(X_i^0, X_j^1)$, where $c$ is the Euclidean distance between the features of cell $i$ from $\mathcal{S}_0$ and cell $j$ from $\mathcal{S}_1$.

The above formulation only considers the feature information of the two slices, but not the geometrical information. Therefore, we can also solve the unbalanced GW problem

$$\min_{\boldsymbol{P} \in \mathbb{R}_+^{n \times m}} \sum_{i,j',k,l'} (\boldsymbol{A}_{ik} - \boldsymbol{B}_{j'l'})^2 \boldsymbol{P}_{ij'} \boldsymbol{P}_{kl'} + \tau \mathrm{KL}(\boldsymbol{P}\mathbf{1}_m \| \boldsymbol{a}) + \tau \mathrm{KL}(\boldsymbol{P}^{\mathrm{T}}\mathbf{1}_n \| \boldsymbol{b}) \tag{49}$$

where $\boldsymbol{A} \in \mathbb{R}^{n \times n}$ is the Euclidean distance matrix between the spatial location of cells within $\mathcal{S}_0$, and $\boldsymbol{B} \in \mathbb{R}^{m \times m}$ is the Euclidean distance matrix between the spatial location of cells within $\mathcal{S}_1$. Similarly, we can combine the above two formulations to solve the unbalanced FGW problem.

Notice the inherent asymmetry stemming from the temporal nature of the problem: all cells on $\mathcal{S}_1$ should have an ancestor from $\mathcal{S}_1$, but not all cells from $\mathcal{S}_1$ need to have a descendent in $\mathcal{S}_0$ because of cell death. Therefore, the most natural OT task for this problem is semi-relaxed OT with the left marginal (the marginal on the first/ancestor slice) relaxed. Specifically, we can also solve the semi-relaxed Wasserstein problem

$$\min_{\boldsymbol{P} \in \Pi_{\cdot, \boldsymbol{b}}} \langle \boldsymbol{C}, \boldsymbol{P} \rangle + \tau \mathrm{KL}(\boldsymbol{P}\mathbf{1}_m \| \boldsymbol{a}) \tag{50}$$

as well as semi-relaxed GW and FGW problem.

**Gene expression prediction task**    Given the alignment matrix $\boldsymbol{P}$ linking cells from $\mathcal{S}_0$ to $\mathcal{S}_1$, we can predict properties of cells in $\mathcal{S}_1$ from properties of cells in $\mathcal{S}_0$. Let the expression of a gene $j$ in $\mathcal{S}_0$ be a vector $\boldsymbol{f}_j \in \mathbb{R}^n$, such that $\boldsymbol{f}_{ji}$ is the expression level of gene $j$ in cell $i$, we can predict the expression of gene $j$ in $\mathcal{S}_1$ as $\widetilde{\boldsymbol{f}}_j = m \times \boldsymbol{P}^{\mathrm{T}} \times \boldsymbol{f}_j \in \mathbb{R}^m$. The accuracy of the prediction can be measured by the Spearman correlation between the predicted expression and the ground truth expression $\overline{\boldsymbol{f}}_j$ of gene $j$ in $\mathcal{S}_1$: $\rho(\widetilde{\boldsymbol{f}}_j, \overline{\boldsymbol{f}}_j)$. In this work, we test the prediction accuracy on 10 test genes: Tubb2b, Pantr1, Actc1, Tnni1, Afp, Hbb-bh1, Fez1, Crabp1, Crabp2, Col3a1, which are markers genes for various cell types in mouse embryo.

**Cell type prediction task**    We can also use the cell type labels of cells in $\mathcal{S}_0$ to predict the cell type labels of cells in $\mathcal{S}_1$. Specifically, for each cell $j$ in $\mathcal{S}_1$, we can assign it the type of the cell $\operatorname{argmax}_i \boldsymbol{P}_{ij}$ in $\mathcal{S}_0$. We can measure the accuracy of the cell type prediction by computing the Adjusted Rand Index (ARI) and Adjusted Mutual Information (AMI) between the predicted clustering of cells in $\mathcal{S}_1$ and the ground truth clustering.

## M.2    Dataset and Preprocessing

We use the large-scale real-world dataset of mouse embryo development Chen et al. (2022), consisting of eight timepoints of ST slices during the whole process of mouse embryo development. In this work, we align the pair of adjacent timepoints of E11.5 and E12.5 embryos, consisting of 30,124 cells and 51,365 cells, respectively. We preprocess the dataset using the standard SCANPY Wolf et al. (2018) pipeline. We first filter the two slices to have the same set of genes, resulting in 26,436 genes for all cells from both slices. We then log-normalize the gene expression of all cells from the two slices, and apply Principle Component Analysis (PCA) to reduce the dimensionality of gene expressions to 30. We take the Euclidean distance between the gene expression in the PCA space as the cost matrix $\boldsymbol{C}$. We take the Euclidean distance between the 2D coordinate of each cell within each slice as the intra-domain cost matrices $\boldsymbol{A}$ and $\boldsymbol{B}$.

Fig. 9 visualizes the two slices in this dataset, with each cell annotated with a cell type from the original publication.

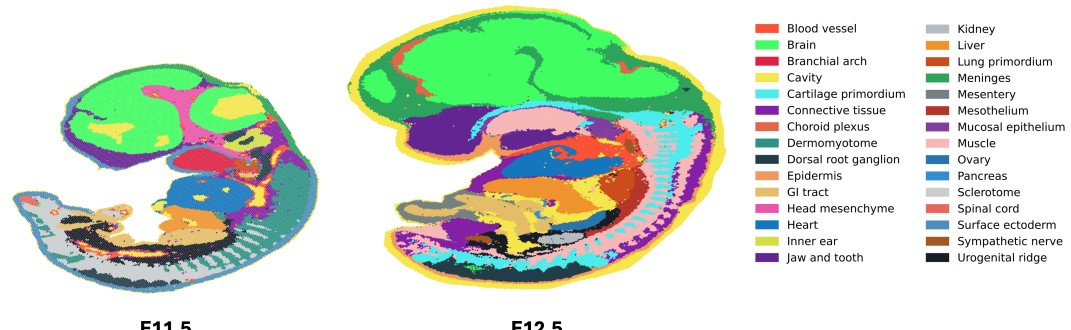

**E11.5**                                    **E12.5**

Figure 9: Visualization of the E11.5 and E12.5 mouse embryos, with each cell colored by the cell type annotated by Chen et al. (2022).

## M.3    Experiment Settings

We compare with the unbalanced low-rank optimal transport algorithm of Scetbon et al. (2023), the backbone of the spatial transcriptomics alignment method moscot Klein et al. (2023), which was shown by Scetbon et al. (2023); Klein et al. (2023) to achieve state-of-the art performance on spatial transcriptomics alignment. We use Scetbon et al. (2023) to solve three unbalanced problems for spatial transcriptomics alignment: the unbalanced Wasserstein problem of (48) (result denoted as LOT-U-W), the unbalanced GW problem of (49) (LOT-U-GW), and the unbalanced FGW problem with a convex combination of the previous two costs (LOT-U-FGW). We use FRLC to solve the same three unbalanced problems (results denoted as FRLC-U-W, FRLC-U-GW, FRLC-U-FGW). Since the two slices contain $> 30,000$ cells and $> 50,000$ cells respectively, a full-rank solution is not feasible, hence we solve for low-rank solutions with the rank validated. We also solve semi-relaxed versions of Wasserstein (result denoted as FRLC-SR-W), GW (FRLC-SR-GW), FGW (FRLC-SR-FGW) problems using our FRLC algorithm, as well as using a particular setting of LOT-U that is equivalent to semi-relaxed solver (results denoted as LOT-SR-W, LOT-SR-GW, LOT-SR-FGW).

We perform extensive grid search to find the best hyperparameter combinations for each method and each problem. The grid of hyperparameters searched for each method is reported in Table. 6. The best performing hyperparameter combination for each method is reported in Table. 7 along with the performance on the validation genes. We pick the best hyperparameters using the Spearman correlation on the gene expression prediction task for 10 validation genes: `Ckb`, `Fabp7`, `Myl4`, `Tnnt2`, `Apoa2`, `Hba-x`, `Tubb3`, `Epha7`, `Ldha`, `Col1a2`, which are marker genes of various cell types as well. We report the performance of the alignment computed by each method using the Spearman correlation on the gene expression prediction task for 10 test genes, as well as the ARI and AMI on the cell type prediction task. Fig. 10 visualizes the ground truth cell type classification versus the classification predicted by our method FRLC-SR-W.

## M.4    Runtime

We report the runtime and OT cost of FRLC and LOT Scetbon et al. (2021) on this dataset (mouse embryo E11.5-12.5) as well as two other datasets (E9.5-10.5, E10.5-11.5) from Chen et al. (2022) in Table 4. For all three datasets, FRLC achieves a better OT cost in a shorter time.

| Dataset | LOT (seconds) | FRLC (seconds) | LOT (OT Cost) | FRLC (OT Cost) |
|---|---|---|---|---|
| Mouse embryo (E9.5–10.5) | 2.545 | 1.112 | 0.440 | 0.385 |
| Mouse embryo (E10.5–11.5) | 4.209 | 1.190 | 0.371 | 0.344 |
| Mouse embryo (E11.5–12.5) | 8.667 | 1.889 | 0.478 | 0.439 |

Table 4: Comparison of methods on Stereo-Seq mouse embryo spatial transcriptomics datasets using GPU and default settings: `min_iter=10`, `max_iter=100`, rank $r = 50$.

# N  $n^{\text{th}}$-roots of unity

## N.1   Problem Statement

To test if FRLC can effectively coarse-grain transport between two datasets with obvious cluster structure, we generate a pair of two-dimensional datasets $\mathbf{Z}^{(1)}$ and $\mathbf{Z}^{(2)}$, with ten and five clusters respectively. We run FRLC with $\boldsymbol{T} \in \mathbb{R}_+^{10 \times 5}$ to see if the barycentric projections for the LC factorization (Definition 4.1) can recover the cluster structure. These projections induce 10 and 5 barycenters for the first and second dataset, and are defined by

$$\mathbf{Y}^{(1)} := \operatorname{diag}(1/\boldsymbol{g}_Q)\boldsymbol{Q}^{\mathrm{T}}\mathbf{Z}^{(1)}, \quad \mathbf{Y}^{(2)} := \operatorname{diag}(1/\boldsymbol{g}_R)\boldsymbol{R}^{\mathrm{T}}\mathbf{Z}^{(2)}.$$

We examine, visually, whether these barycenters are good representatives of the clusters in each dataset, and whether the latent coupling depicts a reasonable transfer of mass between barycenters. We also run FRLC with $\boldsymbol{T} \in \mathbb{R}_+^{10 \times 10}$ and plot the barycenters from the resulting factorization for comparison. As discussed in § 4.2, the barycentric projections defined above, and in Definition 4.1 can be applied to factored couplings Forrow et al. (2019); Scetbon et al. (2021), yielding projections of the form:

$$\mathbf{Y}^{(1)} := \operatorname{diag}(1/\boldsymbol{g})\boldsymbol{Q}^{\mathrm{T}}\mathbf{Z}^{(1)}, \quad \mathbf{Y}^{(2)} := \operatorname{diag}(1/\boldsymbol{g})\boldsymbol{R}^{\mathrm{T}}\mathbf{Z}^{(2)}.$$

Thus, we also ran the method of Scetbon et al. (2021) on this data, called LOT throughout the experiment, with $\boldsymbol{g} \in (\mathbb{R}_+^*)^5$ and $\boldsymbol{g} \in (\mathbb{R}_+^*)^{10}$, plotting its barycenters in each case along with the diagonal latent coupling $\operatorname{diag}(\boldsymbol{g})$.

## N.2   Dataset and Preprocessing

We instantiate a pair of two-dimensional datasets $\mathbf{Z}^{(1)}$ and $\mathbf{Z}^{(2)}$ as follows. Let $\mathcal{U}_n$ denote the $n^{\text{th}}$-roots of unity:

$$\mathcal{U}_n := \{e^{2\pi ik/n} : k = 0, \ldots, n\}.$$

These complex numbers can be equivalently expressed as ordered pairs on the unit circle:

$$\mathsf{c}_k = \begin{pmatrix} \operatorname{Re}(e^{2\pi ik/n}) \\ \operatorname{Im}(e^{2\pi ik/n}) \end{pmatrix} = \begin{pmatrix} \cos(2\pi k/n)) \\ \sin(2\pi k/n) \end{pmatrix}.$$

We consider $n$ uniformly weighted mixtures of Gaussians, where for each sample $X$, we first sample a root of unity uniformly,

$$k \sim \operatorname{Uniform}(n),$$

and conditionally on $k$, we sample $X$ from an isotropic Gaussian centered at this root of unity

$$X \sim \mathcal{N}(\mathsf{c}_k, \sigma^2 \operatorname{Id}_2),$$

using a standard deviation $\sigma = 0.1$. Samples are generated with the `make_blobs` function from `sklearn.datasets`. We generate two datasets in this way, $\mathbf{Z}^{(1)}$ for $n = 10$, and $\mathbf{Z}^{(2)}$ for $n = 5$. To generate dataset $\mathbf{Z}^{(1)}$, we first homogeneously scale the roots of unity to lie on a circle of radius 3 before sampling, so that the two datasets do not overlap but still use the same standard deviation. We do not scale the centers used for $\boldsymbol{Z}^{(2)}$, so they all lie on the unit circle. We generate $\mathbf{Z}^{(1)} = \{\mathbf{z}_1^{(1)}, \ldots, \mathbf{z}_{1000}^{(1)}\}$ and $\mathbf{Z}^{(2)} = \{\mathbf{z}_1^{(2)}, \ldots, \mathbf{z}_{1000}^{(2)}\}$ using 1000 samples each and form the empirical measures

$$\mu := \sum_{i=1}^{1000} \frac{1}{1000}\delta_{\mathbf{z}_i^{(1)}}, \quad \nu := \sum_{j=1}^{1000} \frac{1}{1000}\delta_{\mathbf{z}_j^{(2)}}.$$

supported on $\mathbf{Z}^{(1)}$ and $\mathbf{Z}^{(2)}$, corresponding to uniform probability vectors $\boldsymbol{a}, \boldsymbol{b} \in \Delta_{1000}$.

## N.3   Experiment Settings

We used default hyperparameter settings for both methods. In particular, FRLC sets $\tau = 75$ and $\gamma = 90$, with a maximum of 200 iterations and a minimum of 25 iterations, subject to the stopping criterion $\Delta$ given in (10). The LOT default settings are $\gamma = 10$ (there is no hyperparameter analogous to $\tau$ in LOT). Both methods use a random initialization and were run on CPU.

# O  Discussion of differences between FRLC and existing low-rank optimal transport algorithms

We provide an extensive comparison of FRLC against the parametrization and objective of Scetbon et al. (2021) in Appendix A. As Latent-OT of Lin et al. (2021) is an extension of the $k$-Wasserstein barycenter problem, it has a distinct objective and thus performs worse on primal OT cost (Table 5), making a direct experimental comparison on primal cost minimization only appropriate relative to the works of Scetbon et al. (2021, 2023, 2022). This stated, we still list a number of distinctions between FRLC and Latent-OT – noting that most differences between FRLC and Forrow et al. (2019) transfer from this discussion since Lin et al. (2021) extends the $k$-Barycenter problem of Forrow et al. (2019). (1) Lin et al. optimize two sets of variables: sub-couplings $(Q, R, T)$ and anchor points $(z^x, z^y)$ on which the sub-couplings depend. FRLC only has $(Q, R, T)$ as variables of the optimization. (2) Cost matrices used in Lin et al. (2021) are built from distances between each dataset and its representative anchor points for $Q, R$, or the distances between the two sets of anchor points for $T$. In contrast, ground costs used in FRLC to update $(Q, R, T)$ are always derived from the distance matrix $C$ in the Wasserstein objective $\langle C, P \rangle_F$. Specifically, the cost matrices used by Lin et al. are:

$$[C_Q]_{ik} = \|x_i - z_k^x\|_2^2, \quad [C_R]_{j\ell} = \|y_j - z_\ell^y\|_2^2, \quad [C_T]_{k\ell} = \|z_k^x - z_\ell^y\|_2^2,$$

optionally using a Wasserstein distance for the entries of $C_T$. (3) FRLC costs are given in the exponents of the Gibbs kernels written above and below Equation 9. These are derived directly from the rank-$r$ Wasserstein problem $\min_{P \in \Pi_r(a,b)} \langle C, P \rangle_F$ and differ substantially from those of the proxy objective in Lin et al. (2021); Forrow et al. (2019). (4) The different objectives and variables lead to very different algorithms: Lin et al. alternate updates to the sub-couplings $(Q, R, T)$ using Dykstra, with updates to the latent anchor points $(z^x, z^y)$ using first-order conditions. In contrast, FRLC alternates semi-relaxed OT to update $(Q, R)$ and balanced OT to update $T$. (5) Because FRLC does not require anchor points to define costs, FRLC can handle cost matrices which are not simple functions of distance. For example, if $C_{ij}$ is the price of transporting good $i$ to warehouse $j$ one may not be able to re-evaluate a price $c(x_i, z_k^x)$ between $x_i$ and latent anchor $z_k^x$. In such situations, while finding a low-rank plan may make sense (e.g. to approximate an assignment for a massive dataset), an "anchor" may not have clear definition in the setting of general cost matrices. (6) The Lin et al. objective is only a proxy for a Wasserstein-type loss, and Lin et al. do not explore extensions to Gromov-Wasserstein (GW), or Fused GW, which FRLC readily generalizes to. A summary of the existing low-rank OT algorithms and key distinctions between them is given in Table 1.

For completeness, we offer a compare against the work Latent OT Lin et al. (2021), which solves a variation of the $k$-Wasserstein barycenter problem. As discussed, while their factorization is similar, their problem is distinct from FRLC as it does not solve the primal OT problem for general cost. We report the cost obtained by FRLC and by Latent OT on various simulated datasets in Table 5.

| Dataset | OT-cost (FRLC) | OT-cost (Lin et al.) |
|---|---|---|
| 5th and 10th roots of unity (rank $r_1, r_2 = 5, 10$) | **1.174** | 2.124 |
| Two-moons and 8-Gaussians (rank $r = 20$) | **2.716** | 4.291 |
| 2D Gaussian mixture (rank $r = 20$) | **0.552** | 0.922 |
| 10D Gaussian mixture (rank $r = 20$) | **1.038** | 1.298 |

Table 5: Comparison against Lin et al. (2021) in primal OT-cost $\langle C, P \rangle_F$.

# P  Limitations

Our method introduces an additional hyperparameter $\tau$ relative to previous approaches Scetbon et al. (2021), controlling the strength of the KL penalty on the inner marginals when updating $Q$ and $R$.

Empirically, we found FRLC to be robust to different choices of $\tau$, but applying the method optimally requires this additional hyperparamter in any grid-search.

We also note that the non-asymptotic criterion $\Delta(\cdot, \cdot)$ is weak relative to stronger notions of convergence, and that often users might prefer to simply run the method up to some number of maximal iterations by setting the parameter for whether $\Delta(\cdot, \cdot)$ is used to `False`. The W optimization empirically converges to minima smoothly, so for the most part there is not much of a need for $\Delta$ except for early stopping. We recommend that users plot the loss over iterations and use it to set the tolerance parameter `tol`, and the minimum and maximum iteration parameters for the time-being. The needs for these parameters might vary widely–the minimum number of iterations should be very low (around 5) for simple datasets and substantially higher for high-dimensional, structured ones.

Although we demonstrate strong performance already, there is massive room for improvement as our implementation is preliminary and not at the level of a high-performance library like `ott-jax`. We use lightweight vanilla implementations of Sinkhorn as a sub-routine, not taking advantage of the momentum-based techniques which could accelerate it massively. Thus, one can imagine that the potential scalability and speed of this method could be much higher than reported in this document.

## Q  Broader impacts

FRLC is general enough to be used modularly within any ML algorithm using OT as a subroutine to help with scalability. We also note that the LC factorization is similar to a PCA in the context of OT, yielding an optimal low-rank coupling with an interpretable latent coupling factor.

| Hyperparameter | Values |
|---|---|
| rank (Both) | 50, 100, 200 |
| $\tau$ (Ours) | 30, 50, 100 |
| $\tau$ (LOT-U) | 0.99, 0.9, 0.7 |
| $\epsilon$ (LOT-U) | 0.001, 0.01, 0.1 |

Table 6: Hyperparameter grid considered in hyperparameter search for validation. Scetbon et al. (2023); Klein et al. (2023) scales $\tau' = \frac{\tau}{1-\tau}$ and their $\tau'$ are in the same range.

| Solver | Rank | $\tau$ (Ours) | $\tau$ (UL) | $\epsilon$ (UL) | Spearman $\rho$ (Validation) |
|---|---|---|---|---|---|
| FRLC-SR-W (Ours) | 200 | 30 | - | - | 0.465 |
| FRLC-SR-GW (Ours) | 100 | 100 | - | - | 0.288 |
| FRLC-SR-FGW (Ours) | 200 | 50 | - | - | 0.465 |
| FRLC-U-W (Ours) | 200 | 100 | - | - | 0.471 |
| FRLC-U-GW (Ours) | 50 | 30 | - | - | 0.282 |
| FRLC-U-FGW (Ours) | 200 | 30 | - | - | 0.353 |
| LOT-U-W | 200 | - | 0.9 | 0.001 | 0.394 |
| LOT-U-GW | 100 | - | 0.99 | 0.01 | 0.001 |
| LOT-U-FGW | 200 | - | 0.7 | 0.001 | 0.393 |
| LOT-SR-W | 200 | - | 0.7 | 0.001 | 0.394 |
| LOT-SR-GW | 200 | - | 0.7 | 0.1 | 0.003 |
| LOT-SR-FGW | 200 | - | 0.99 | 0.001 | 0.399 |

Table 7: The best performing hyperparameters for each solver and the performance on the validation genes.

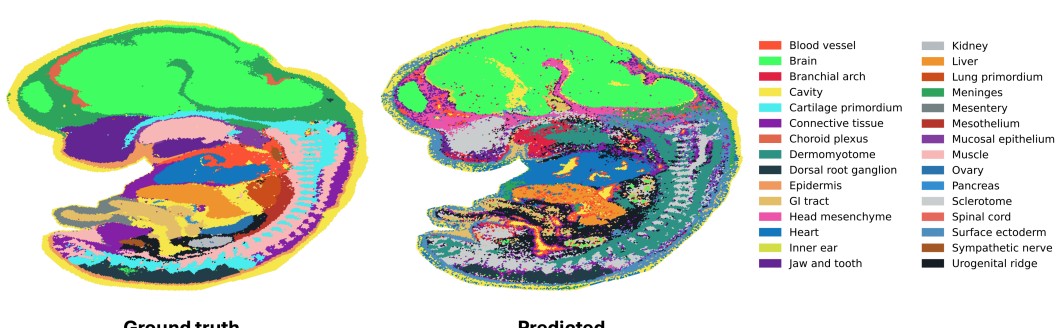

Figure 10: Ground truth and the predicted cell type classification of the E12.5 embryo.

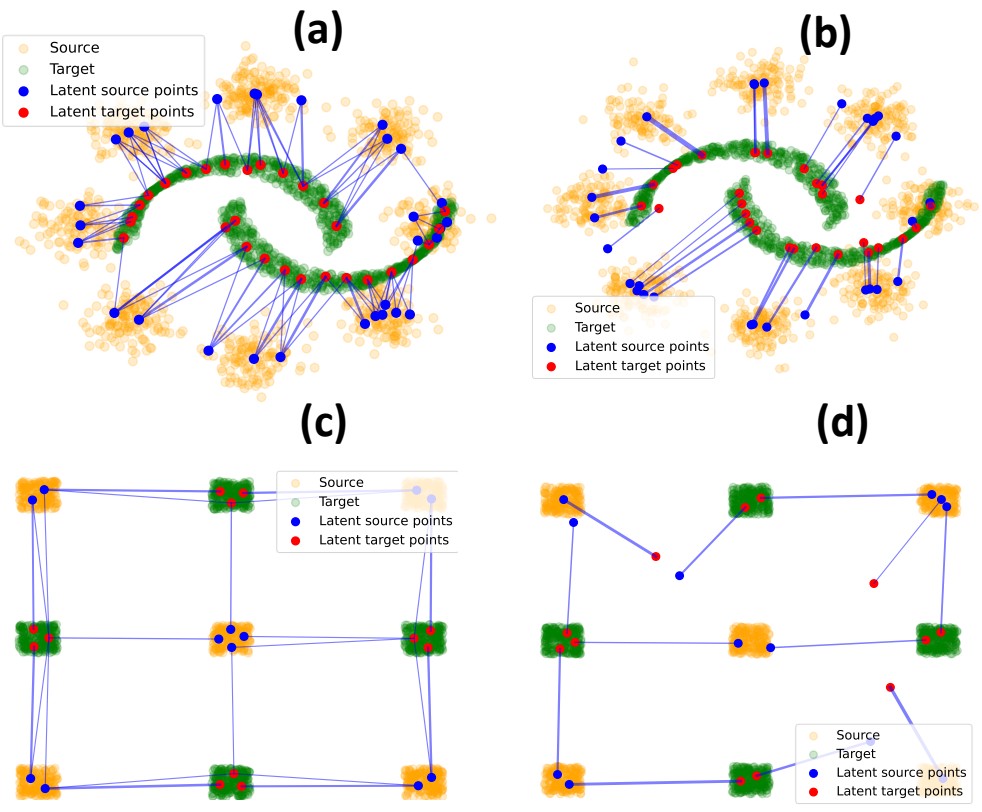

Figure 11: (a) LC-projection barycenters aligned with FRLC latent-coupling $\boldsymbol{T}$ on (a) two moons and eight Gaussians ($r = 30$), (b) LC-projection barycenters aligned with $\mathrm{diag}(\boldsymbol{g})$ from Scetbon et al. (2021) ($r = 30$), (c) the checkerboard dataset with FRLC latent coupling aligned barycenters ($r = 12$), and (d) with diagonal alignment Scetbon et al. (2021) ($r = 12$). We show in A.1 that the output of FRLC can be diagonalized to the factorization of Forrow et al. (2019) (Figure 12).

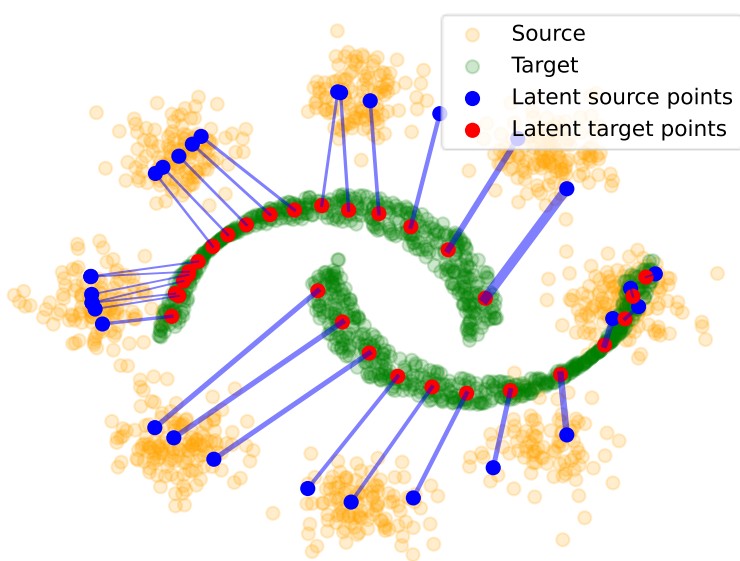

Figure 12: As discussed in A.1, one may recover the factorization of Forrow et al. (2019) as a sub-case of the LC-factorization. Shown is the factorization found by diagonalizing the output of FRLC from $(\boldsymbol{Q}, \boldsymbol{R}, \boldsymbol{T}) \mapsto (\boldsymbol{Q}', \boldsymbol{R}, \mathrm{diag}(\boldsymbol{g}_R))$.

