# OpenReview forum: "Low-Rank Optimal Transport through Factor Relaxation with Latent Coupling"
_NeurIPS.cc/2024/Conference — NeurIPS 2024 poster_

### Official Review · Reviewer_1xw8 · 2024-07-12

**Soundness:** 4
**Presentation:** 4
**Contribution:** 4
**Rating:** 9
**Confidence:** 4

**Summary:**

As the paper clearly outlines in the introduction, Optimal Transportation (OT) is widely used in various fields of machine learning, however, the cost for computing OT is computationaly expensive (quadratic scaling), even after Sinkhorn algorithm employing entropy regularization significantly alleviated the cost. The purpose of Low-rank Optimal Transport (LOT) is to reduce the complexity by constraining the transportation plan to possess a certain low-rank structure.

The key contribution of the paper is (1) new parametrization of LOT using a Latent Coupling (LC) factorization, (2) The eponymous "Balance FRLC" algorithm for resolving the optimization procedure for computing LOT with respect to various objectives associated with (LOT), (3) Theoretical guarantees for the smoothness of the regularized objectives, as well as upper bounds for the convergence criteria.  The algorithm is applied to illustrative toy examples and then to a more realistic example (spatial transcriptomics alignment) and the results indicate improvements over previous LOT.

**Strengths:**

The contributions of the paper is significant. The writing is excellent. The LC factorization, adapted to the LOT framework from a previous work, is well-motivated. The Balanced FRLC algorithm appears a sound and non-trivial extension of previous approaches for LOT (Sinkhorn, semi-relaxed OT). The algorithm is particularly simple and shown to be effective. The experimental results are clear and illustrative, improvements are convincing.

**Weaknesses:**

What this reviewer found missing is discussion of the case the optimal transport map is itself not low-rank. It is well-known that the optimal transportation plan is not necessarily low-rank (as indicated e.g. by their connection to differential equations with non-regular solutions), so LOT may converge very slowly to the full-rank optimal transportation plan. In section 4.2, the authors provide a simple illustrative example, but this is when the transportation plan is known to be low-rank - in this case the transportation plan seems to stabilize after some threshold rank - however, what happens when the full-rank plan does not have an inherently low-rank structure?

**Questions:**

- line 103: Would it be good to say the what complexity "massively improved" to?
- Can the authors comment on how to pick the rank for the LOT? (This of course ties in to the point above the rank of the transportation plan.)
- Perhaps a more discussion of of how different the LC factorization is to, say the previous factored coupling with (the diagonal version $P = Q \text{diag}(1/g) R^T$). This relation seems important when considering "more" LC factorizations: If this LC factorization improves the result while simultaneously allowing a simple optimization algorithm, would it then be natural to discuss further factorizations that are nested this way, with additional LC factorizations for $T$ and so on?

**Limitations:**

This work builds on the LOT framework - so perhaps the authors comment on some scenarios where the LC factorization can potentially yield no better results, when compared to previous LOT?

---

> ### Author Rebuttal · Authors · 2024-08-07
>
> We thank reviewer 1xw8 for their careful reading of our work, and for their feedback.
>
> > What this reviewer found missing is discussion of the case the optimal transport map is itself not low-rank. It is well-known that the optimal transportation plan is not necessarily low-rank (as indicated e.g. by their connection to differential equations with non-regular solutions), so LOT may converge very slowly to the full-rank optimal transportation plan. In section 4.2, the authors provide a simple illustrative example, but this is when the transportation plan is known to be low-rank - in this case the transportation plan seems to stabilize after some threshold rank - however, what happens when the full-rank plan does not have an inherently low-rank structure?
>
> This is a great question. We are actively investigating this, but do not have a complete answer during this short rebuttal period. We will include this point in the Conclusion of our main text:
>
> _"Another direction for further investigation is to better understand what structure LC factorizations capture when the optimal plan is known to have full rank, e.g. when the Monge map exists as has been explored by works such as [Liu et al. (2021)](https://arxiv.org/abs/2111.06546)."_
>
> > line 103: Would it be good to say the what complexity "massively improved" to?
>
> Yes, thank you for the suggestion. For a fixed error tolerance $\varepsilon$ and entropy constant $\eta$, FRLC finds a $\pm \varepsilon D$ approximation for $D$ the diameter of the dataset in $B = \mathrm{poly}(1/(\eta \varepsilon))$ iterations which is a major improvement over the previous cubic solvers.
>
> > Can the authors comment on how to pick the rank for the LOT? (This of course ties in to the point above the rank of the transportation plan.)
>
> Thank you for your question. The best we can say regarding the rank is that the choice is problem-specific, and is much like choosing $k$ in k-means or choosing the rank $r$ for a compact SVD (or PCA). We will add the following sentence to the Conclusion (Section 5) to directly address this limitation:
>
> _"Two key limitations of our work are: (1) selecting values of the latent coupling ranks and the hyperparameter $\tau$ controlling the smoothness of the trajectory; and (2) strengthening the convergence criterion. These and other limitations are discussed in Section N of the Appendix."_
>
> > Perhaps a more discussion of of how different the LC factorization is to, say the previous factored coupling with (the diagonal version $P=Q\mathrm{diag}(1/g)R^\mathrm{T}$). This relation seems important when considering "more" LC factorizations: If this LC factorization improves the result while simultaneously allowing a simple optimization algorithm, would it then be natural to discuss further factorizations that are nested this way, with additional LC factorizations for  and so on?
>
> We agree that one strength of the LC factorization is that it can be applied again to decompose a latent coupling, leading to a multi-scale family of optimal transport problems similar to [Gerber et al. (2017)](https://dl.acm.org/doi/abs/10.5555/3122009.3176816). We leave a thorough exploration of this to future work, but will will highlight the possibility of nesting LC factorizations in the main text -- thank you.
>
> > This work builds on the LOT framework - so perhaps the authors comment on some scenarios where the LC factorization can potentially yield no better results, when compared to previous LOT?
>
> FRLC generally has better performance, especially on large and structured datasets such as spatial transcriptomics, but the difference in performance is insignificant for ultra low rank (e.g. $r _ 1=2$) on synthetic datasets. There are also cases where one might correctly assume that transitions are diagonal between clusters -- in these cases, it might be better to use the previous factorization, which implicitly clusters the two datasets with a common set of labels. We note that we can recover the previous LOT factorization: by taking ${Q}' \gets {Q}\mathrm{diag}(1/{g} _ {Q}){T}$ one may diagonalize the return as $({Q},{R},{T}) \to ({Q}',{R}, \mathrm{diag}({g} _ {R}))$, with no change to the OT cost.

---

> > ### Comment · Reviewer_1xw8 · 2024-08-12
> >
> > I thank the authors for the detailed response. I will keep the score.

---

### Official Review · Reviewer_CGKE · 2024-07-15

**Soundness:** 3
**Presentation:** 3
**Contribution:** 3
**Rating:** 6
**Confidence:** 3

**Summary:**

This work introduces a new low rank formulation of optimal transport based on the latent coupling decomposition introduced in https://arxiv.org/pdf/2012.11589 . Compared to previous formulations of low rank OT, this formulation allows easier extensions to unbalanced and Gromov Wasserstein settings.

**Strengths:**

I found that the presented FRLC method is a quite elegant approach to low rank OT. It decouples optimization into entropy regularized semi relaxed OT problems for $Q$ and $R$, and $T$ is updated solving entropy regularized  OT of size $r$. This decoupling is enabled by the form of the inner coupling which is no longer assumed to be diagonal. This construction alleviates the need to use the Dykstra machinery as in (Scetbon 2021).
With the proposed approach, $Q$ and $R$ are not required to have the same inner marginal. From what I understand, the optimization over inner marginals is handled by the semi relaxed steps and does not require a separate step as in previous approaches.

Another strength of the approach is that it generalizes smoothly to the unbalanced setting.

Other strengths include:
* I carefully looked at the presented method and the appendix and the algorithmic approach made sense.
* non asymptoptic convergence bounds are derived for the method.
* the appendix includes a complete review of the necessary technical background presented in a clear way, easy to follow.
* limitations are clearly stated (section N in appendix).
* a new initialization scheme for LC decomposition is proposed.
* there is a gain in interpretability compared to (Scetbon 2021) as the inner coupling captures the coupling between clusters in the data.

**Weaknesses:**

* from section 3.1 to 3.2, the new aspects of the approach compared to https://arxiv.org/pdf/2012.11589 should be stated more clearly. The contributions of the paper should be separated more clearly from previous works. I had to read the other paper to better understand each contribution.
* In algorithm 1, it would be interesting to have the computational complexity of each step (presented in the algorithm directly).

**Questions:**

* How to choose $r$ in practical scenarios ?
* Would it make sense to remove the entropic penalties (at the expense of smoothness) to recover results about hard clustering of the points via the decomposition of the OT coupling ? Put differently, what is the effect of entropic smoothing on the interpretability of the coupling in terms of clustering ?

**Limitations:**

/

---

> ### Author Rebuttal · Authors · 2024-08-07
>
> We thank reviewer CGKE for their careful reading of our work, and for their feedback.
>
> > from section 3.1 to 3.2, the new aspects of the approach compared to [https://arxiv.org/pdf/2012.11589](https://arxiv.org/pdf/2012.11589) should be stated more clearly. The contributions of the paper should be separated more clearly from previous works. I had to read the other paper to better understand each contribution.
>
> We agree that we could have been more clear about how our method is distinct from the approach of Lin et al. (2021). We greatly appreciate your comment, and we refer you to the bulleted list in our general response highlighting the differences between our work and that of Lin et al. (2021). We will add a section in our Appendix contrasting the approaches in more detail.
>
> > In algorithm 1, it would be interesting to have the computational complexity of each step (presented in the algorithm directly).
>
> We thank you for your comment, and will make sure to include a discussion of time-complexity in the appendix and in-line with our algorithm in the main text. We refer you to the time-complexity analysis of FRLC included in our general response above.
>
> > How to choose $r$ in practical scenarios ?
>
> This is a great question. The best we can say regarding the rank is that the choice is problem-specific, and is much like choosing $k$ in k-means or choosing the rank $r$ for a compact SVD (or PCA). We added the following sentence to the Conclusion (section 5) to directly address this limitation:
>
> _"Two key limitations of our work are: (1) selecting values of the latent coupling ranks and the hyperparameter $\tau$ controlling the smoothness of the trajectory; and (2) strengthening the convergence criterion. These and other limitations are discussed in Section N of the Appendix."_
>
> > Would it make sense to remove the entropic penalties (at the expense of smoothness) to recover results about hard clustering of the points via the decomposition of the OT coupling? Put differently, what is the effect of entropic smoothing on the interpretability of the coupling in terms of clustering?
>
> This is a fantastic question.
>
> Scetbon et al. [Scetbon et al. (2022a)](https://proceedings.neurips.cc/paper_files/paper/2022/hash/2d69e771d9f274f7c624198ea74f5b98-Abstract-Conference.html) show in their Proposition 9 that removing the entropic penalties and setting the second marginal to equal the first generalizes the $k$-means objective. [Lin et al. (2021)](https://proceedings.mlr.press/v139/lin21a.html) show that setting the distance between the anchors ($C _ {z}$) to zero also recovers $k$-means.
> However, because FRLC does not use Scetbon et al.'s factorization, and also does not evaluate a distance $C _ {z}$ between anchors, we do not know if removing the entropic penalties in FRLC would recover a known hard clustering (or co-clustering) method. We will add the following sentence to our Conclusion section discussing future work:
>
> _"Another interesting question is whether a relationship exists between the minimization of the Wasserstein cost over LC-factorizations without entropic regularization and calculation of a hard clustering or co-clustering between data points, analogous to Proposition 9, [Scetbon et al. (2022a)](https://proceedings.neurips.cc/paper_files/paper/2022/hash/2d69e771d9f274f7c624198ea74f5b98-Abstract-Conference.html)."_

---

> > ### Comment · Reviewer_CGKE · 2024-08-09
> >
> > Thank you very much for your answer.
> > I would like to keep my score.

---

### Official Review · Reviewer_pnTm · 2024-07-16

**Soundness:** 3
**Presentation:** 3
**Contribution:** 3
**Rating:** 6
**Confidence:** 4

**Summary:**

The paper introduces a novel framework called Factor Relaxation with Latent Coupling (FRLC) which is based on coordinate mirror descent to compute the low-rank LC factorization.The algorithm decouples the optimization into three sub-problems, offering greater flexibility, interpretability, and linear space complexity. FRLC is applicable to various OT objectives (Wasserstein, Gromov-Wasserstein, Fused Gromov-Wasserstein) and marginal constraints (balanced, unbalanced, and semi-relaxed). Theoretical results support its effectiveness, and empirical tests demonstrate superior performance in applications like graph clustering and spatial transcriptomics.

**Strengths:**

The introduction of the LC factorization and its integration into the FRLC algorithm represents an innovation in reducing the complexity of optimal transport problems.

The FRLC framework is versatile, handling multiple OT objectives and marginal constraints, making it applicable to a wide range of practical problems.

The latent coupling provides a interpretable description of the transport plan, which is beneficial for understanding and visualizing the results.

**Weaknesses:**

While the empirical results are good, the paper could benefit from more extensive comparisons with additional baseline methods to solidify the claims of superiority.

The theoretical results are limited in that, even though $\Delta_k(x_k, x_{k+1})$ is small, it cannot be guaranteed that the iteration points converge to the optimal solution or even a stationary point.

**Questions:**

I suggest that Propositions E.3, E.5, E.6 should be stated in the main text to make the theoretical part easier to follow.

Table 1 should be put in the main text to show the efficiency of FRLC in terms of runtime. It is hard to say that the method shows significant improvement because the runtime of LOT is already very short (less than one second). I suggest the authors conduct experiments on more challenging and diverse examples.

**Limitations:**

N.A.

---

> ### Author Rebuttal · Authors · 2024-08-07
>
> We thank reviewer pnTm for their careful reading of our work, and for their feedback.
>
> > While the empirical results are good, the paper could benefit from more extensive comparisons with additional baseline methods to solidify the claims of superiority.
>
> The only OT methods solving for a low-rank plan that we are aware of are Factored Coupling [Forrow et al. (2019)](http://proceedings.mlr.press/v89/forrow19a), LOT ([Scetbon et al. (2021)](http://proceedings.mlr.press/v139/scetbon21a.html), [Scetbon et al. (2022b)](https://proceedings.mlr.press/v162/scetbon22b), [Scetbon et al. (2023)](https://openreview.net/forum?id=d2WsCmoITF)), Latent OT [Lin et al. (2021)](https://proceedings.mlr.press/v139/lin21a.html). We did not compare to Forrow et al. (2019) because a prior work of Scetbon et al. (2021) established LOT as the current state of the art, and we did not compare to Lin et al. because they do not minimize the primal OT cost directly. However, we now added a comparison to Lin on the balanced Wasserstein objective cost, and we will add Table 1 in our above general response to our Appendix.
>
> Importantly, Lin et al. solve what they term the latent OT problem, which is a different objective than the low-rank Wasserstein objective computed by FRLC.  Thus we did not feel it was appropriate to include this comparison with the others in the main text, but will include as part of a new addition to clarify the differences betweeen FRLC and Lin et al.
>
> In Section 4.4 (additional experiments) we refer to Section K of our Appendix where we also evaluated against full-rank OT solvers (from `ott-jax`) on downstream metrics for 12 graph partitioning datasets.
>
> > The theoretical results are limited in that, even though $\Delta _ k(x _ k, x _ {k-1})$ is small, it cannot be guaranteed that the iteration points converge to the optimal solution or even a stationary point.
>
> We agree. In our convergence results, we adapted the criterion of [Ghadimi et al. (2014)](https://link.springer.com/article/10.1007/s10107-014-0846-1) to coordinate-mirror descent to show non-asymptotic stationary convergence, as in other works on low-rank optimal transport [Scetbon et al. (2021)](http://proceedings.mlr.press/v139/scetbon21a.html), [Scetbon et al. (2022b)](https://proceedings.mlr.press/v162/scetbon22b). Such a result is the current baseline for convergence in the literature, but we acknowledge that the convergence criterion and guarantee is limited. A more thorough theoretical analysis with stronger guarantees might be difficult as the objective is non-convex, but would be valuable to analyze and understand more comprehensively in future work.
> We mentioned this limitation in the Appendix, but in the revision we will add the following sentence to the Conclusion (Section 5) to more clearly indicate the limitations:
>
> _"Two key limitations of our work are: (1) selecting values of the latent coupling ranks and the hyperparameter $\tau$ controlling the smoothness of the trajectory; and (2) strengthening the convergence criterion. These and other limitations are discussed in Section N of the Appendix."_
>
> > I suggest that Propositions E.3, E.5, E.6 should be stated in the main text to make the theoretical part easier to follow.
>
> Absolutely, we had removed them from the main text due to the page limit, but will include them in the updated version of the manuscript.
>
> > Table 1 should be put in the main text to show the efficiency of FRLC in terms of runtime. It is hard to say that the method shows significant improvement because the runtime of LOT is already very short (less than one second). I suggest the authors conduct experiments on more challenging and diverse examples.
>
> We will move this table to the main text. We also performed additional comparisons on the large spatial transcriptomics datasets of [Chen et al. (2022)](https://www.sciencedirect.com/science/article/pii/S0092867422003993), which we include in Table 2 of our general response above.

---

> > ### Comment · Reviewer_pnTm · 2024-08-12
> >
> > Thank for your replies. I will increase my score from 5 to 6.

---

### Official Review · Reviewer_jx3E · 2024-07-26

**Soundness:** 3
**Presentation:** 3
**Contribution:** 3
**Rating:** 7
**Confidence:** 3

**Summary:**

The paper presents an approach for low-rank optimal transport (OT) leveraging a latent coupling (LC) factorization and solving it with mirror descent. This approach offers a new parameterization of the low-rank OT problem, providing advantages such as decoupling the problem into three OT problems and enhancing flexibility and interpretability. The authors introduce the Factor Relaxation with Latent Coupling (FRLC) algorithm, which utilizes coordinate mirror descent to compute the LC factorization. FRLC accommodates various OT objectives and marginal constraints. The paper includes theoretical results and demonstrates the performance on selected tasks like graph clustering and spatial transcriptomics.

**Strengths:**

- Generality: handles different OT objective costs and relaxations of the marginal constraints.
- Significance: the proposed method, and improving OT using factorizations, has the potential to address the scalability issue of OT with large datasets, making it highly relevant for applications in machine learning and data science.

**Weaknesses:**

**Presentation and logical organization of the ideas**

I found it difficult to discern the paper's particular contribution, especially concerning previous art like Latent Optimal Transport (Lin et al., 2021), Forrow et al. (2019), and Scetbon et al. (2021). For example, the main contribution is said to be "compute a minimum cost low-rank transport plan using the LC factorization," where LC factorization is precisely the factorizations proposed by Lin et al. (2021), with the further contribution being the computation of this factorization using mirror-descent. I wondered how much of the description also applies to Lin et al. (2021) and which part is particular to the novel method.

Furthermore, according to Lin et al. (2021), there is an algorithm for computing low-rank factorized plans based on factorized costs using a projection method and an algorithm to factorize the costs and select anchor points without the need to have a similar number of anchor points at source and target latent space. How does your approach compare with the approach?

**Comparison and empirical validation**
The first empirical comparison analyzes LOT and FRLC, capturing how both methods behave as the factorization rank changes. This is a good way of understanding the effect of the rank on each case. Nevertheless, in the second setup, the comparison with LOT is not present anymore, which begs the question of how LOT performs in the second empirical evaluation. Furthermore, an extensive empirical study with the structured approach of Lin et al. (2021) would help clarify the particular contributions of the type of factorization and factoring algorithm used.

I found some other issues lacking in a more in-depth discussion of the computational complexity, possible implementation difficulties, and overall limitations in the proposal.

**Questions:**

- Is the main contribution here actually the mirror descent formulation of the OT problem with the factorization with a different set of latent source and target points proposed by Lin et al. (2021)?
- Furthermore, is the main difference between the fact that Lin et al. (2021) use a two-stage process and the novel mirror-descent method solves the OT problem in a more straightforward unified process?
- What are the theoretical complexity of the algorithms, and what practical considerations are relevant when implementing the method?

One presentation improvement that would change my evaluation is a summary table locating the differences in the type of factorization, whether the cost or transport is being factorized, the algorithm used for computing the factorization, etc.

**Limitations:**

I found it necessary to discuss in more depth the limitations and the tradeoffs present in the proposal compared with previous art.

---

> ### Author Rebuttal · Authors · 2024-08-07
>
> We thank reviewer jx3E for their careful reading of our work, and for their feedback. We have abbreviated your questions due to space limitations for our responses.
>
> > [I found ... novel method.]
>
> Thank you for this comment indicating an area where we can strengthen our presentation. Our FRLC algorithm differs from [Lin et al. (2021)](https://proceedings.mlr.press/v139/lin21a.html) in several respects. We omit them here due to space, but ask you to refer to the bulleted list of differences in our general response. We will include these points with additional details in our Appendix to clarify the differences between the approaches.
>
> > [Furthermore ... the approach?]
>
> We hope our general response addresses most of this question.
>
> Additionally, we want to emphasize that Lin et al. (2021) explicitly compute anchor points in each iteration of their algorithm, whereas FRLC has no anchor points. In our evaluation in Section 4.2, we computed LC projections from the output of FRLC on each dataset in order to demonstrate the interpretability of the latent coupling. These LC projections may be interpreted as the analogs of the anchor points used in Lin et al. (2021), but importantly the two concepts are not identical. Even if we specialize to using Euclidean distance, we are unaware of a way to write FRLC cost matrices as functions of the anchor-like points obtained from LC projection.
>
> > [The first ... empirical evaluation.]
>
> First, to clarify, in our experiments section (Section 4), LOT/UL/SRL all refer to the low-rank methods of Scetbon et al. ([Scetbon et al. (2021)](http://proceedings.mlr.press/v139/scetbon21a.html), [Scetbon et al. (2022b)](https://proceedings.mlr.press/v162/scetbon22b), [Scetbon et al. (2023)](https://openreview.net/forum?id=d2WsCmoITF)), and not to the Lin et al. (2021) method which coincidentally is also named LOT. UL and SRL denote the unbalanced and semi-relaxed versions (from Scetbon et al. (2023)) of Scetbon's LOT method Scetbon et al. (2021), respectively. We understand how this terminology can cause confusion, and we will clearly distinguish the method names in our revision. To avoid further confusion in this response (as in our paper), we use LOT to reference the work of Scetbon et al., and write "Lin et al." to refer to Lin et al.'s Latent OT.
>
> We assume that "second setup" refers to Section 4.3, where we compare against UL and SRL rather than LOT. As noted above, UL and SRL are the unbalanced and semi-relaxed variants of LOT, respectively. We note that we also included a comparison between the balanced versions of FRLC and LOT for this dataset (Figure 8 in our Appendix). Thus, all of our comparisons -- including Section 4.1 (Figure 2) and Section 4.2 (Figure 3) -- are directly head-to-head with LOT (with the exception of Appendix K where we compare against the full-rank solvers of `ott-jax`).
>
> > [Furthermore ... algorithm used.]
>
> We ran the Lin et al. (2021) method on several synthetic datasets and report the OT-cost (low-rank Wasserstein cost) found by this method and FRLC in Table 1 of our general response.
>
> > [I found ... the proposal.]
>
> To address the first point, we offer a brief complexity analysis in the general response above.
> Regarding implementation difficulties and limitations, we included a discussion on limitations at the end of our Appendix, and  will add the sentences indicated in our general response to our Conclusion.
>
> > [Is the main ...  Lin et al. (2021)?]
>
> Not quite -- as we noted above the FRLC algorithm does not have latent points (anchor points) at all, but rather optimizes directly over the sub-couplings $Q$, $R$, and $T$. This has several advantages as noted above, including that the steps in our algorithm are themselves OT sub-problems, unlike Forrow et al. (2019), Scetbon et al. (2021), and Lin et al. (2021) which use Dykstra projections. Moreover, FRLC is the first algorithm which solves this factorization for general costs $C$ and other objectives (GW, FGW). We refer to the bullet points above in the general response for more discussion on the differences and contributions.
>
> > [Furthermore ... unified process?]
>
> Yes, we agree that FRLC solves the OT problem in a more straightforward and unified process: we only optimize the sub-couplings, while Lin et al. (2021) alternate updates on the anchor points and the sub-couplings. However, this is only one of multiple differences between the methods.
>
> > [What are ... method?]
>
> We give a brief analysis of the time complexity of FRLC in our general response above, and will include this in our main text and Appendix. Regarding practical considerations in implementing FRLC, in practice taking the hyper-parameter $\tau$ to be close to $0$ approaches a theoretically optimal step on the sub-couplings while large $\tau$ regularizes their trajectory. Thus, it is an open question how to choose this hyperparameter, as discussed in the limitations component of our general response, and likewise with the choice of dimensions for the latent coupling.
>
> > [One presentation ... etc.]
>
> This is a great suggestion -- we include a table here, and will add this to our Appendix.
>
> | Method | Factorization  | Cost | Variables | Algorithm  | Subroutine for sub-couplings |
> |-|-|-|-|-|-|
> | Factored Coupling  (Forrow et al. (2019)) | Diagonal | $k$-Wasserstein barycenter| Anchors \& sub-couplings| Lloyd-type | Dykstra |
> | Latent OT (Lin et al. (2021)) | Non-diagonal | Extension of $k$-Wasserstein barycenter| Anchors \& sub-couplings| Lloyd-type | Dykstra |
> | LOT (Scetbon et al. (2021)) | Diagonal | Primal OT cost | Sub-couplings | Mirror-descent | Dykstra                |
> | FRLC (our work) | Non-diagonal | Primal OT cost | Sub-couplings | Coordinate mirror-descent | OT                       |
>
> **Table:** Comparison of latent and low-rank OT methods.
>
> > [I found ... previous art.]
>
> We included a discussion on limitations in Appendix N but will now add the sentence given above regarding limitations to the main text.

---

> > ### Comment · Reviewer_jx3E · 2024-08-08
> >
> > Thank you for carefully engaging with the main points of my review. I believe the shared concern with other reviewers regarding the presentation of the idea and how it differs from previous art is fundamental to improving the quality of this paper. I believe the table with the comparisons could be in the main text; those technical differences and how they map to papers and methods could be immediate for those working directly with these problems, but it can easily be lost for the larger community, although being relevant.
> >
> > I am satisfied with the complexity analysis and extended empirical discussions.

---

> > > ### Author Response · Authors · 2024-08-09
> > >
> > > Dear reviewer jx3E, we thank you for your comments and are glad to see that this response helped clarify a few of your questions. We agree that placing this work in context is essential, and will add the table to the main text (rather than the appendix) following your suggestion. If you have any further questions, we are more than happy to offer any clarifications during the remainder of the discussion period.

---

### Author Rebuttal · Authors · 2024-08-07

Here we include information for all reviewers. We thank each reviewer for their helpful feedback and questions.

> Regarding time complexity raised by reviewers jx3E, CGKE:

The time complexity of FRLC $O(BLr^{2}(n+m))$, where $B$ is the number of Sinkhorn iterations, $L$ the number of mirror descent iterations, $n$ is the number of samples in the first dataset, $m$ is the number of samples in the second dataset, and $r$ is both the rank of the latent coupling and the rank of the distance matrix $C$. The time complexity is thus linear in $n+m$ per iteration. The number $L$ of outer iterations follows from the convergence rate of coordinate mirror descent. The number of iterations required for each projection follow from the convergence of Sinkhorn where, for $\varepsilon$ a fixed error tolerance and $\eta$ the entropy constant, one finds a $\pm \varepsilon D$ approximation for $D$ the diameter of the data in $B = \mathrm{poly}(1/(\eta \varepsilon))$ iterations. Each step of the projection is $O((n+m)r^{2})$ assuming a rank-$r$ factorization of $C$, allowing every matrix multiplication to have one dimension on the order of the constant rank $r$. Therefore, the runtime is $O(B L r^2 (n+m))$.

> Regarding comparisons to Lin et al. (2021) brought up by jx3E, CGKE, we benchmarked FRLC against their method on the Wasserstein objective across four synthetic datasets:

| Dataset | OT-cost (FRLC) | OT-cost (Lin et al.) |
|-|-|-|
| 5th and 10th roots of unity (rank $r _ {1},r _ {2}=5,10$) | 1.174 | 2.124 |
| Two-moons and 8-Gaussians (rank $r=20$) | 2.716 | 4.291 |
| 2D Gaussian mixture (rank $r=20$)  | 0.552 | 0.922 |
| 10D Gaussian mixture (rank $r=20$) | 1.038 | 1.298 |

**Table 1:** Comparison against Lin et al. (2021) in primal OT-cost $\langle {C}, {P} \rangle _ {F}$

The table shows that FRLC achieves a lower cost on all datasets. Importantly, Lin et al. solve what they term the latent OT problem, which is a different objective than the low-rank Wasserstein objective minimized by FRLC. Thus we did not feel it was appropriate to include this comparison in the main text, but will include as part of a new addition to our Appendix to clarify the differences between FRLC and Lin et al.

> We also report our runtime on more challenging datasets, large spatial transcriptomics datasets of [Chen et al. (2022)](https://www.sciencedirect.com/science/article/pii/S0092867422003993), as requested by reviewer pnTm:

| Dataset | LOT (seconds) | FRLC (seconds) | OT cost (LOT) | OT Cost (FRLC) |
|-|-|-|-|-|
| Mouse embryo (E9.5-10.5) | 2.545 | 1.112 | 0.4396 | 0.385 |
| Mouse embryo (E10.5-11.5) | 4.209 | 1.190 | 0.3714 | 0.344 |
| Mouse embryo (E11.5-12.5) | 8.667 | 1.889 | 0.478 | 0.439 |

**Table 2:** Comparison of methods on MOSTA Stereo-Seq mouse embryo spatial transcriptomics datasets (GPU). This is using default settings with `min_iter=10`, `max_iter=100`, rank $r=50$. LOT denotes the low-rank method of Scetbon et al. (2021), as a point of comparison.

> In response to several reviewer questions about hyperparameter selection and our convergence criterion, we will add the following sentence to our Conclusion section:

_"Two key limitations of our work are: (1) selecting values of the latent coupling ranks and the hyperparameter $\tau$ controlling the smoothness of the trajectory; and (2) strengthening the convergence criterion. These and other limitations are discussed in Section N of the Appendix."_

> Regarding questions raised by reviewers jx3E, CGKE on how FRLC differs from the method of Lin et al. (2021), here is a short summary of the differences:

* Lin et al. (2021) optimize two sets of variables: sub-couplings $(Q, R, T)$ and anchor points $(z^x, z^y)$. FRLC only has $(Q,R,T)$ as variables of the optimization.
* Cost matrices used in Lin et al. (2021) are built from distances between each dataset and its representative anchor points for $Q, R$, or the distances between the two sets of anchor points for $T$. In contrast, ground costs used in FRLC to update $(Q,R,T)$ are always derived from the distance matrix $C$ in the Wasserstein objective $\langle C, P\rangle _ F$.
    * Specifically, the cost matrices used by Lin et al. (2021) are:
    $$
    [C _ Q^{\mathrm{Lin}}]_{ik} = \Vert x _ i - z _ k^x \Vert _ 2^2, \quad [C _ R^{\mathrm{Lin}}] _ {j\ell} = \Vert y _ j - z _ \ell^y \Vert _ 2^2, \quad [C _ T^{\mathrm{Lin}}] _ {k\ell} = \Vert z _ k^x - z _ \ell^y \Vert _ 2^2,
    $$
    optionally using a Wasserstein distance for the entries of $C_T^{\mathrm{Lin}}$
    * FRLC costs are given in the exponents of the Gibbs kernels written above and below equation (9) in our paper. These are derived directly from the rank-$r$ Wasserstein problem $\min _ {P \in \Pi _ r(a,b)} \langle C, P \rangle _ {F}$ and differ substantially from those of the proxy objective used in Lin et al. (2021).
* The different objectives and variables lead to different algorithms: Lin et al. (2021) alternate updates to the sub-couplings $(Q,R,T)$ using Dykstra, with updates to the latent anchor points $(z^{x}, z^{y})$ using first-order conditions. In contrast, FRLC alternates semi-relaxed OT to update $(Q,R)$ and balanced OT to update $T$.
* Moreover, because FRLC does not require anchor points to define costs, FRLC can handle cost matrices which are not simple functions of distance. For example, if $C _ {ij}$ is the price of transporting good $i$ to warehouse $j$ one may not be able to re-evaluate a price $c(x _ {i}, z _ {k}^x)$ between $x _ i$ and latent anchor $z _ {k}^x$. In such situations, while finding a low-rank plan may make sense (e.g. to approximate an assignment for a massive dataset), an "anchor" may not have clear definition in the setting of general cost matrices.
* The Lin et al. (2021) objective is only a proxy for a Wasserstein-type loss, and Lin et al. do not explore extensions to Gromov-Wasserstein (GW), or Fused GW, which FRLC readily generalizes to.

---

### Decision · Program_Chairs · 2024-09-25

**Decision:**

Accept (poster)

**Comment:**

The paper presents a new numerical approach for low-rank optimal transport, introducing a novel mirror descent algorithm. The numerical results demonstrate the method's ability to address scalability issues with large datasets in optimal transport. The authors made a significant effort in the rebuttal, providing additional comparisons to previous works. However, the exposition could be improved to more clearly highlight the key novelties compared to existing low-rank optimal transport methods, particularly those presented by Scetbon et al. (2021). Additionally, the authors suggest that the factorization using (Q, R, g) variables was introduced in Forrow et al. (2019), but this formulation was introduced in Scetbon et al. (2021) (the formulation in Forrow et al. relies on an intermediate barycenter measure of vectors, and works with squared Euclidean costs); this should be clarified in the final version. I recommend acceptance.